# On the Complexity of Adversarial Decision Making

**Dylan J. Foster**
dylanfoster@microsoft.com

**Alexander Rakhlin**
rakhlin@mit.edu

**Ayush Sekhari**
sekhari@mit.edu

**Karthik Sridharan**
ks999@cornell.edu

## Abstract

A central problem in online learning and decision making—from bandits to reinforcement learning—is to understand what modeling assumptions lead to sample-efficient learning guarantees. We consider a general *adversarial decision making* framework that encompasses (structured) bandit problems with adversarial rewards and reinforcement learning problems with adversarial dynamics. Our main result is to show—via new upper and lower bounds—that the Decision-Estimation Coefficient, a complexity measure introduced by Foster et al. [17] in the stochastic counterpart to our setting, is necessary and sufficient to obtain low regret for adversarial decision making. However, compared to the stochastic setting, one must apply the Decision-Estimation Coefficient to the *convex hull* of the class of models (or, hypotheses) under consideration. This establishes that the price of accommodating adversarial rewards or dynamics is governed by the behavior of the model class under convexification, and recovers a number of existing results—both positive and negative. En route to obtaining these guarantees, we provide new structural results that connect the Decision-Estimation Coefficient to variants of other well-known complexity measures, including the Information Ratio of Russo and Van Roy [47] and the Exploration-by-Optimization objective of Lattimore and György [32].

## 1 Introduction

To reliably deploy data-driven decision making methods in real-world systems where safety is critical, such methods should satisfy two desiderata: (i) provable robustness in the face of dynamic or even adversarial environments, and (ii) ability to effectively take advantage of problem structure as modeled by the practitioner. In high-dimensional problems, this entails efficiently generalizing across states and actions while delicately exploring new decisions.

For decision making in static, stochastic environments, recent years have seen extensive investigation into optimal sample complexity and algorithm design principles, and the foundations are beginning to take shape. With an emphasis on reinforcement learning, a burgeoning body of research identifies specific modeling assumptions under which sample-efficient interactive decision making is possible [12, 54, 20, 39, 5, 27, 14, 36, 13, 56], as well as general structural conditions that aim to unify these assumptions [45, 19, 51, 53, 15, 21, 17]. For dynamic or adversarial settings, however, comparatively little is known outside of (i) positive results for special cases such as adversarial bandit problems [4, 3, 18, 10, 1, 7, 26, 16, 8, 29], and (ii) a handful of negative results suggesting that online reinforcement learning in agnostic or adversarial settings can actually be statistically intractable [48, 37]. These developments raise the following questions: (a) what are the underlying phenomena that govern the statistical complexity of decision making in adversarial settings? (b) what are the corresponding algorithmic design principles that attain optimal statistical complexity?

**Contributions.** We consider an adversarial variant of the *Decision Making with Structured Observations* (DMSO) framework introduced in Foster et al. [17], where a learner or decision-maker interacts with a sequence of *models* (reward distributions in the case of bandits, or MDPs in the case of reinforcement learning) chosen by an adaptive adversary, and aims to minimize regret against the

36th Conference on Neural Information Processing Systems (NeurIPS 2022).

best decision in hindsight. Models are assumed to belong to a known *model class*, which reflects the learner's prior knowledge about the problem. The main question we investigate is: *How does the structure of the model class determine the minimax regret for adversarial decision making?* We show:

1. For *any* model class, one can obtain high-probability regret bounds that scale with a *convexified* version of the *Decision-Estimation Coefficient* (DEC), a complexity measure introduced by Foster et al. [17].

2. For any algorithm with "reasonable" tail behavior, the optimal regret for adversarial decision making is lower bounded by (a suitably localized version of) the convexified DEC.

In the process of obtaining these results, we draw new connections to several existing complexity measures.

## 1.1 Problem Setting

We adopt an adversarial variant of the DMSO framework of Foster et al. [17] consisting of $T$ rounds, where at each round $t = 1, \ldots, T$:

1. The learner selects a *decision* $\pi^{(t)} \in \Pi$, where $\Pi$ is the *decision space*.

2. Nature selects a *model* $M^{(t)} \in \mathcal{M}$, where $\mathcal{M}$ is a *model class*.

3. The learner receives a reward $r^{(t)} \in \mathcal{R} \subseteq \mathbb{R}$ and observation $o^{(t)} \in \mathcal{O}$ sampled via $(r^{(t)}, o^{(t)}) \sim M^{(t)}(\pi^{(t)})$, where $\mathcal{O}$ is the *observation space*. We abbreviate $z^{(t)} := (r^{(t)}, o^{(t)})$ and $\mathcal{Z} := \mathcal{R} \times \mathcal{O}$.

Here, each model $M = M(\cdot, \cdot \mid \cdot) \in \mathcal{M}$ is a conditional distribution $M : \Pi \to \Delta(\mathcal{R} \times \mathcal{O})$ that maps the learner's decision to a distribution over rewards and observations. This setting subsumes (adversarial) bandit problems, where models correspond to reward functions (or distributions), as well as adversarial reinforcement learning, where models correspond to Markov decision processes (MDPs). In both cases, the model class $\mathcal{M}$ encodes prior knowledge about the decision making problem, such as structure of rewards or dynamics (e.g., linearity or convexity). The model class might be parameterized by linear models, neural networks, or other rich function approximators depending on the problem domain.

For a model $M \in \mathcal{M}$, $\mathbb{E}^{M,\pi}[\cdot]$ denotes expectation under the process $(r, o) \sim M(\pi)$. We define $f^M(\pi) := \mathbb{E}^{M,\pi}[r]$ as the mean reward function and $\pi_M := \arg\max_{\pi \in \Pi} f^M(\pi)$ as the decision with greatest reward for $M$. We let $\mathcal{F}_{\mathcal{M}} = \{f^M \mid M \in \mathcal{M}\}$ denote the induced class of reward functions. We measure performance via *regret* to the best fixed decision in hindsight:[1]

$$\mathbf{Reg}_{\mathsf{DM}} := \sup_{\pi^\star \in \Pi} \sum_{t=1}^{T} \mathbb{E}_{\pi^{(t)} \sim p^{(t)}} \left[ f^{M^{(t)}}(\pi^\star) - f^{M^{(t)}}(\pi^{(t)}) \right]. \tag{1}$$

This formulation—in which models are selected by a potentially adaptive adversary—generalizes Foster et al. [17], who considered a *stochastic* setting where $M^{(t)} = M^\star$ is fixed across all rounds. Examples include:

- **Adversarial bandits.** With no observations ($\mathcal{O} = \{\varnothing\}$), the adversarial DMSO framework is equivalent to the *adversarial bandit* problem with structured rewards. In this context, $\pi^{(t)}$ is typically referred to as an *action* or *arm* and $\Pi$ is referred to as the *action space*. The most basic example here is the adversarial finite-armed bandit problem with $A$ actions [4, 3, 18], where $\Pi = \{1, \ldots, A\}$ and $\mathcal{F}_{\mathcal{M}} = \mathbb{R}^A$. Other well-studied examples include adversarial linear bandits [10, 1, 7], bandit convex optimization [26, 16, 8, 29], and nonparametric bandits [26, 6, 38].[2]

- **Reinforcement learning.** The adversarial DMSO framework encompasses finite-horizon, episodic online reinforcement learning, with each round $t$ corresponding to a single episode:

---

[1]The results in this paper immediately extend to the regret $\sup_{\pi^\star \in \Pi} \sum_{t=1}^{T} r^{(t)}(\pi^\star) - r^{(t)}(\pi^{(t)})$ through standard tail bounds.

[2]Typically, these examples are formulated with deterministic rewards, which we encompass by restricting models in $\mathcal{M}$ to be deterministic. Our formulation is more general and allows for, e.g., semi-stochastic adversaries.

$\pi^{(t)}$ is a *policy* (a mapping from state to actions) to play in the episode, $r^{(t)}$ is the cumulative reward in the episode, and the observation $o^{(t)}$ is the episode's trajectory (sequence of observed states, actions, and rewards). Online reinforcement learning in the stochastic setting where $M^{(t)} = M^\star$ is fixed has received extensive attention [19, 51, 20, 53, 15, 21, 17], but the adversarial setting we study has received less investigation. Examples include the adversarial MDP problem where an adversary chooses a sequence of tabular MDPs, which is known to be intractable [37], and the easier problem in which there is a fixed (known) MDP but rewards are adversarial [40, 57, 41, 22]. See Appendix D for more details.

We refer to Appendix B for additional measure-theoretic details and background, and to Foster et al. [17] for further examples and detailed discussion.[3]

Understanding statistical complexity (i.e., minimax regret) for the DMSO setting at this level of generality is a challenging problem. Even if one restricts only to bandit-type problems with no observations, any complexity measure must capture the role of structural assumptions such as convexity or smoothness in determining the optimal rates. To go beyond bandit problems and handle the general setting, one must accommodate problems with rich, structured feedback such as reinforcement learning, where observations (as well as subtle features of the noise distribution) can reveal information about the underlying model.

## 1.2 Overview of Results

For a model class $\mathcal{M}$, reference model $\overline{M} \in \mathcal{M}$, and scale parameter $\gamma > 0$, the Decision-Estimation Coefficient [17] is defined via

$$\mathsf{dec}_\gamma(\mathcal{M}, \overline{M}) = \inf_{p \in \Delta(\Pi)} \sup_{M \in \mathcal{M}} \mathbb{E}_{\pi \sim p}\big[ f^M(\pi_M) - f^M(\pi) - \gamma \cdot D_{\mathsf{H}}^2\big(M(\pi), \overline{M}(\pi)\big)\big], \qquad (2)$$

where we recall that for probability measures $\mathbb{P}$ and $\mathbb{Q}$ with a common dominating measure $\nu$, (squared) Hellinger distance is given by

$$D_{\mathsf{H}}^2(\mathbb{P}, \mathbb{Q}) = \int \left( \sqrt{\frac{d\mathbb{P}}{d\nu}} - \sqrt{\frac{d\mathbb{Q}}{d\nu}} \right)^2. \qquad (3)$$

We define $\mathsf{dec}_\gamma(\mathcal{M}) = \sup_{\overline{M} \in \mathcal{M}} \mathsf{dec}_\gamma(\mathcal{M}, \overline{M})$, and let $\mathrm{co}(\mathcal{M})$ denote the convex hull of $\mathcal{M}$, which can be viewed as the set of all mixtures of models in $\mathcal{M}$. Our main results show that the *convexified Decision-Estimation Coefficient*,

$$\mathsf{dec}_\gamma(\mathrm{co}(\mathcal{M})),$$

leads to upper and lower bounds on the optimal regret for adversarial decision making.

**Theorem (informal).** *For any model class $\mathcal{M}$, Algorithm 1 ensures that with high probability,*

$$\mathbf{Reg}_{\mathsf{DM}} \lesssim \mathsf{dec}_\gamma(\mathrm{co}(\mathcal{M})) \cdot T, \qquad (4)$$

*where $\gamma$ satisfies the balance $\mathsf{dec}_\gamma(\mathrm{co}(\mathcal{M})) \propto \frac{\gamma}{T} \log|\Pi|$. Moreover, for any algorithm with "reasonable" tail behavior (Section 2.2), regret must scale with a localized version of the same quantity.*

*As a consequence, there exists an algorithm for which $\mathbb{E}[\mathbf{Reg}_{\mathsf{DM}}] \leq \tilde{o}(T)$ if and only if $\mathsf{dec}_\gamma(\mathrm{co}(\mathcal{M})) \propto \gamma^{-\rho}$ for some $\rho > 0$.*

For the stochastic version of our setting, Foster et al. [17] give upper and lower bounds that scale with $\mathsf{dec}_\gamma(\mathcal{M})$, without convexifying (under appropriate technical assumptions; cf. Section 2.3). Hence, our results show that in general, the gap in optimal regret for stochastic and adversarial decision making (or, "price of adversarial outcomes") is governed by the behavior of the DEC under convexification. For example, multi-armed bandits, linear bandits, and convex bandits correspond to convex model classes (where $\mathrm{co}(\mathcal{M}) = \mathcal{M}$), which gives a post-hoc explanation for why these problems are tractable in the adversarial setting. Finite state/action Markov decision processes do not correspond to a convex model class, and have $\mathsf{dec}_\gamma(\mathrm{co}(\mathcal{M}))$ exponentially large compared to $\mathsf{dec}_\gamma(\mathcal{M})$; in this case, our results recover lower bounds of Liu et al. [37].

Beyond these results, we prove that the convexified Decision-Estimation Coefficient is equivalent to:

---

[3]We mention in passing that the upper bounds in this paper encompass the more general setting where rewards are not observed by the learner (i.e., $z^{(t)}$ does not contain the reward), thus subsuming the partial monitoring problem. Our lower bounds, however, require that rewards are observed. See Appendix A.

1. a "parameterized" variant of the generalized Information Ratio of Lattimore and György [32].

2. a novel high-probability variant of the *Exploration-by-Optimization* objective of Lattimore and Szepesvári [35], Lattimore and György [32].

**Our techniques.** On the lower bound side, we strengthen the approach from Foster et al. [17] with an improved change-of-measure argument (leading to improved results even in the stochastic setting), and combine this with the simple idea of constructing adversaries based on static mixture models. On the upper bound side, we extend the powerful Exploration-by-Optimization machinery of Lattimore and György [32] to the DMSO setting, and give a novel high-probability variant of the technique which leads to regret bounds for adaptive adversaries. We show that the performance of this method is controlled by a complexity measure whose value is equivalent to the convexified DEC, as well as parameterized variant of the Information Ratio (we present results in terms of the former to draw comparison to the stochastic setting).

Overall, our results heavily draw on the work of Foster et al. [17] and Lattimore and György [32], but we believe they play a valuable role in bridging these lines of research and formalizing connections.

**Organization.** Section 2 presents our main results, including upper and lower bounds on regret and a characterization of learnability. In Section 3, we provide new structural results connecting the DEC to Exploration-by-Optimization and the Information Ratio. We close with discussion of future directions (Section 4). Additional comparison to related work is deferred to Appendix A. The appendix also contains proofs and additional results, including examples (Appendix D) and further structural results (Appendix E).

## 2 Main Results

We now present our main results. First, using a new high-probability variant of the Exploration-by-Optimization technique [35, 32], we provide an upper bound on regret based on the (convexified) Decision-Estimation Coefficient (Section 2.1). Next, we present a lower bound that scales with a localized version of the same quantity (Section 2.2). Finally, we use these results to give a characterization for learnability (Section 2.3), and discuss the gap between stochastic and adversarial decision making.

To keep presentation as simple as possible, we make the following assumption.

**Assumption 2.1.** *The decision space $\Pi$ has $|\Pi| < \infty$, and we have $\mathcal{R} = [0, 1]$.*

This assumption only serves to facilitate the use of the minimax theorem, and we expect that our results can be generalized (e.g., with covering numbers as in Section 3.4 of Foster et al. [17]).

### 2.1 Upper Bound

In this section we give regret bounds for adversarial decision making based on the (convexified) Decision-Estimation Coefficient. A-priori, it is not obvious why the DEC should bear any relevance to the adversarial setting we consider: The algorithms and regret bounds based on the DEC that Foster et al. [17] introduce for the stochastic setting heavily rely on the ability to estimate a static underlying model, yet in the adversarial setting, the learner may only interact with each model a single time. This renders any sort of global estimation (e.g., for dynamics of an MDP) impossible. In spite of this difficulty, we show that regret bounds can be achieved by building on the *Exploration-by-Optimization* technique of Lattimore and Szepesvári [35], Lattimore and György [32], which provides an elegant approach to estimating rewards that exploits the structure of the model class under consideration.

Exploration-by-Optimization—introduced by Lattimore and Szepesvári [35] and substantially expanded in Lattimore and György [32]—can be thought of as a generalization of the classical EXP3 algorithm [4] for finite-action bandits, which applies the exponential weights method for full-information online learning to a sequence of unbiased importance-weighted estimators for rewards. While EXP3 is near-optimal for bandits, it is unsuitable for general model classes because the reward estimators the algorithm uses do not exploit the structure of the decision space. Consequently, the regret scales linearly with $|\Pi|$ rather than with, e.g., dimension, as one might hope for problems like linear bandits. The idea behind Exploration-by-Optimization is to solve an optimization problem at each round to search for a (potentially biased) reward estimator and modified sampling distribution that better exploit the structure of the model class $\mathcal{M}$, leading to information sharing and improved regret. Lattimore and György [32] showed that for a general partial monitoring setting (cf. Appendix A),

---

**Algorithm 1** High-Probability Exploration-by-Optimization ($\mathsf{ExO}^+$)

---

1: **parameters**: Learning rate $\eta > 0$.

2: **for** $t = 1, 2, \cdots, T$ **do**

3:     Define $q^{(t)} \in \Delta(\Pi)$ via exponential weights update:

$$q^{(t)}(\pi) = \frac{\exp\left(\eta \sum_{i=1}^{t-1} \widehat{f}^{(i)}(\pi)\right)}{\sum_{\pi' \in \Pi} \exp\left(\eta \sum_{i=1}^{t-1} \widehat{f}^{(i)}(\pi')\right)}. \tag{5}$$

4:     Solve *high-probability exploration-by-optimization* objective:          `// See Eq. (8)`

$$(p^{(t)}, g^{(t)}) \leftarrow \underset{p \in \Delta(\Pi), g \in \mathcal{G}}{\arg\min} \sup_{M \in \mathcal{M}, \pi^\star \in \Pi} \Gamma_{q^{(t)}, \eta}(p, g\,; \pi^\star, M). \tag{6}$$

5:     Sample decision $\pi^{(t)} \sim p^{(t)}$ and observe $z^{(t)} = (r^{(t)}, o^{(t)})$.

6:     Form reward estimator:

$$\widehat{f}^{(t)}(\pi) = \frac{g^{(t)}\big(\pi; \pi^{(t)}, z^{(t)}\big)}{p^{(t)}\big(\pi^{(t)}\big)}. \tag{7}$$

---

the expected regret for this method—and for more general family of algorithms based on Bregman divergences—is bounded by a generalization of the Information Ratio of Russo and Van Roy [46, 47].

Our development builds on that of Lattimore and György [32], but we pursue *high-probability* guarantees rather than in-expectation guarantees. This allows us to provide regret bounds that hold for *adaptive adversaries*, rather than oblivious adversaries as considered in prior work.[4] Beyond this basic motivation, our interest in high-probability guarantees comes from the lower bound in the sequel (Section 2.2), which shows that the convexified Decision-Estimation Coefficient lower bounds regret for algorithms with "reasonable" tail behavior. To develop high-probability regret bounds and complement this lower bound, we use a novel variant of the Exploration-by-Optimization objective and a specialized analysis that goes beyond the Bregman divergence framework.

Our algorithm, $\mathsf{ExO}^+$, is displayed in Algorithm 1. At each round $t$, the algorithm computes a *reference distribution* $q^{(t)} \in \Delta(\Pi)$ by applying the standard exponential weights update (with learning rate $\eta > 0$) to a sequence of reward estimators $\widehat{f}^{(1)}, \ldots, \widehat{f}^{(t-1)}$ from previous rounds (Line 3). For the main step (Line 4), the algorithm obtains a *sampling distribution* $p^{(t)} \in \Delta(\Pi)$ and an *estimation function* $g^{(t)} \in \mathcal{G} := (\Pi \times \Pi \times \mathcal{Z} \to \mathbb{R})$ by solving a minimax optimization problem based on a new objective we term *high-probability exploration-by-optimization*: Defining

$$\Gamma_{q, \eta}(p, g\,; \pi^\star, M) := \mathbb{E}_{\pi \sim p}[f^M(\pi^\star) - f^M(\pi)] \tag{8}$$

$$+ \frac{1}{\eta} \cdot \mathbb{E}_{\pi \sim p, z \sim M(\pi)} \mathbb{E}_{\pi' \sim q}\left[\exp\left(\frac{\eta}{p(\pi)}(g(\pi'; \pi, z) - g(\pi^\star; \pi, z))\right) - 1\right],$$

we solve

$$(p^{(t)}, g^{(t)}) \leftarrow \underset{p \in \Delta(\Pi), g \in \mathcal{G}}{\arg\min} \sup_{M \in \mathcal{M}, \pi^\star \in \Pi} \Gamma_{q^{(t)}, \eta}(p, g\,; \pi^\star, M). \tag{9}$$

Finally (Lines 5 and 6), the algorithm samples $\pi^{(t)} \sim p^{(t)}$, observes $z^{(t)} = (r^{(t)}, o^{(t)})$, and then forms an importance-weighted reward estimator via $\widehat{f}^{(t)}(\pi) := g^{(t)}(\pi; \pi^{(t)}, z^{(t)})\big/p^{(t)}(\pi^{(t)})$.

The interpretation of the high-probability Exploration-by-Optimization objective (8) is as follows: For a given round $t$, the model $M \in \mathcal{M}$ and decision $\pi^\star \in \Pi$ should be thought of as a proxy for the true model $M^{(t)}$ and optimal decision, respectively. By solving the minimax problem in (9), the min-player aims to—in the face of an unknown, worst-case model—find a sampling distribution that minimizes instantaneous regret, yet ensures good tail behavior for the importance-weighted estimator $g(\cdot; \pi, z)/p(\pi)$. Tail behavior is captured by the moment generating function-like term in (8), which penalizes the learner for over-estimating rewards under the reference distribution $q$ or under-estimating rewards under $\pi^\star$.

---

[4]In general, in-expectation regret bounds do not imply high-probability bounds. For example, in adversarial bandits, the EXP3 algorithm can experience linear regret with constant probability [34].

We show that this approach leads to a bound on regret that scales with the convexified DEC.

**Theorem 2.1** (Main upper bound). *For any choice of $\eta > 0$, Algorithm 1 ensures that for all $\delta > 0$, with probability at least $1 - \delta$,*

$$\mathbf{Reg_{DM}} \leq \mathsf{dec}_{1/8\eta}(\mathrm{co}(\mathcal{M})) \cdot T + \frac{2}{\eta} \cdot \log(|\Pi|/\delta). \tag{10}$$

*In particular, for any $\delta > 0$, with appropriate $\eta$, the algorithm ensures that with probability at least $1 - \delta$,*

$$\mathbf{Reg_{DM}} \leq O(1) \cdot \inf_{\gamma > 0}\{\mathsf{dec}_{\gamma}(\mathrm{co}(\mathcal{M})) \cdot T + \gamma \cdot \log(|\Pi|/\delta)\}. \tag{11}$$

This regret bound holds for arbitrary, potentially adaptive adversaries. The result should be compared to the upper bound for the stochastic setting in Foster et al. [17] (e.g., Theorem 3.3), which takes a similar form, but scales with the weaker quantity $\sup_{\overline{M} \in \mathrm{co}(\mathcal{M})} \mathsf{dec}_{\gamma}(\mathcal{M}, \overline{M})$.[5] See Appendix A for comparison to Lattimore and Szepesvári [35], Lattimore and György [32].

**Equivalence of Exploration-by-Optimization and Decision-Estimation Coefficient.** We now discuss a deeper connection between Exploration-by-Optimization and the DEC. Define the minimax value of the high-probability Exploration-by-Optimization objective via

$$\mathsf{exo}_{\eta}(\mathcal{M}, q) := \inf_{p \in \Delta(\Pi), g \in \mathcal{G}} \sup_{M \in \mathcal{M}, \pi^{\star} \in \Pi} \Gamma_{q,\eta}(p, g \, ; \pi^{\star}, M), \tag{12}$$

and let $\mathsf{exo}_{\eta}(\mathcal{M}) := \sup_{q \in \Delta(\Pi)} \mathsf{exo}_{\eta}(\mathcal{M}, q)$. This quantity can be interpreted as a complexity measure for $\mathcal{M}$ whose, value reflects the difficulty of exploration. The following structural result (Corollary 3.1 in Section 3), which is critical to the proof of Theorem 2.1, shows that this complexity measure is equivalent to the convexified Decision-Estimation Coefficient:

$$\mathsf{dec}_{(4\eta)^{-1}}(\mathrm{co}(\mathcal{M})) \leq \mathsf{exo}_{\eta}(\mathcal{M}) \leq \mathsf{dec}_{(8\eta)^{-1}}(\mathrm{co}(\mathcal{M})), \quad \forall \eta > 0. \tag{13}$$

As we show, the regret of Algorithm 1 is controlled by the value of $\mathsf{exo}_{\eta}(\mathcal{M})$, and thus Theorem 2.1 follows. In the process of proving (13), we also establish equivalence of the Exploration-by-Optimization objective and a *parameterized* version of the Information Ratio, which is of independent interest (cf. Section 3). Both results build on, but go beyond the Bregman divergence-based framework in Lattimore and György [32], and exploit a somewhat obscure connection between Hellinger distance and the moment generating function (MGF) for the logarithmic loss. In particular, we use a technical lemma (proven in Appendix C), which shows that up to constants, the Hellinger distance between two probability distributions can be expressed as variational problem based on the associated MGFs.

**Lemma 2.1.** *Let $\mathbb{P}$ and $\mathbb{Q}$ be probability distributions over a measurable space $(\mathcal{X}, \mathscr{F})$. Then*

$$\frac{1}{2}D_{\mathsf{H}}^2(\mathbb{P}, \mathbb{Q}) \leq \sup_{g:\mathcal{X} \to \mathbb{R}}\left\{1 - \mathbb{E}_{\mathbb{P}}[e^g] \cdot \mathbb{E}_{\mathbb{Q}}[e^{-g}]\right\} \leq D_{\mathsf{H}}^2(\mathbb{P}, \mathbb{Q}). \tag{14}$$

The lower inequality in Lemma 2.1 is proven using a trick similar to one used by Zhang [55] to prove high-probability bounds for maximum likelihood estimation based on Hellinger distance. To prove the upper bound in (13), we apply the lower inequality in (14) with the test function $g$ taking the role of the estimation function in the Exploration-by-Optimization objective.

**Further remarks.** The main focus of this work is statistical complexity (in particular, minimax regret), and the runtime and memory requirements of Algorithm 1, which are linear in $|\Pi|$, are not practical for large decision spaces. Improving the computational efficiency is an interesting question for future work. We mention in passing that Theorem 2.1 answers a question raised by Foster et al. [17] of obtaining in the frequentist setting a regret bound matching the Bayesian regret bound in their Theorem 3.6.

---

[5]If a proper estimation algorithm (i.e., an algorithm producing estimators that lie in $\mathcal{M}$) is available, Foster et al. [17] (Theorem 4.1) gives tighter bounds scaling with $\mathsf{dec}_{\gamma}(\mathcal{M})$.

## 2.2 Lower Bound

We now complement the regret bound in the prequel with a lower bound based on the convexified DEC. Our most general result shows that for any algorithm, either the expected regret or its (one-sided) second moment must scale with a localized version of the convexified DEC.

To state the result, we define the *localized model class* around a model $\overline{M}$ via

$$\mathcal{M}_\varepsilon(\overline{M}) = \big\{ M \in \mathcal{M} : f^{\overline{M}}(\pi_{\overline{M}}) \geq f^M(\pi_M) - \varepsilon \big\},$$

and define $\mathsf{dec}_{\gamma,\varepsilon}(\mathcal{M}) := \sup_{\overline{M} \in \mathcal{M}} \mathsf{dec}_\gamma(\mathcal{M}_\varepsilon(\overline{M}), \overline{M})$ as the *localized Decision-Estimation Coefficient*. We let $(x)_+ := \max\{x, 0\}$ and define $V(\mathcal{M}) := \sup_{M,M' \in \mathcal{M}} \sup_{\pi \in \Pi} \sup_{A \in \mathcal{R} \otimes \mathcal{O}} \big\{ \frac{M(A|\pi)}{M'(A|\pi)} \big\} \vee e$;[6] finiteness of $V(\mathcal{M})$ is not necessary, but removes a $\log(T)$ factor from Theorem 2.2.

**Theorem 2.2** (Main lower bound). *Let $C(T) := c \cdot \log(T \wedge V(\mathcal{M}))$ for a sufficiently large numerical constant $c > 0$. Set $\varepsilon_\gamma := \frac{\gamma}{4C(T)T}$. For any algorithm, there exists an oblivious adversary for which*

$$\mathbb{E}[\mathbf{Reg}_{\mathsf{DM}}] + \sqrt{\mathbb{E}(\mathbf{Reg}_{\mathsf{DM}})_+^2} \geq \Omega(1) \cdot \sup_{\gamma > \sqrt{2C(T)T}} \mathsf{dec}_{\gamma,\varepsilon_\gamma}(\mathrm{co}(\mathcal{M})) \cdot T - O(T^{1/2}). \tag{15}$$

Theorem 2.2 implies that for any algorithm (such as Algorithm 1) with tail behavior beyond what is granted by control of the first moment, the regret in Theorem 2.1 cannot be substantially improved. In more detail, consider the notion of a *sub-Chebychev* algorithm.

**Definition 2.1** (Sub-Chebychev Algorithm). *A regret minimization algorithm is said to be sub-Chebychev with parameter $R$ if for all $t > 0$,*

$$\mathbb{P}((\mathbf{Reg}_{\mathsf{DM}})_+ \geq t) \leq R^2/t^2. \tag{16}$$

For sub-Chebychev algorithms, both the mean and (root) second moment of regret are bounded by the parameter $R$ (cf. Appendix F.4), which has the following consequence.

**Corollary 2.1.** *Any regret minimization algorithm with sub-Chebychev parameter $R > 0$ must have*

$$R \geq \widetilde{\Omega}(1) \cdot \sup_{\gamma > \sqrt{2C(T)T}} \mathsf{dec}_{\gamma,\varepsilon_\gamma}(\mathrm{co}(\mathcal{M})) \cdot T - O(T^{1/2}). \tag{17}$$

To interpret this result, suppose for simplicity that $\mathsf{dec}_\gamma(\mathrm{co}(\mathcal{M}))$ and $\mathsf{dec}_{\gamma,\varepsilon_\gamma}(\mathrm{co}(\mathcal{M}))$ are continuous with respect to $\gamma > 0$, and that $\mathsf{dec}_{\gamma,\varepsilon_\gamma}(\mathrm{co}(\mathcal{M})) \gtrsim \gamma^{-1}$, which is satisfied for non-trivial classes.[7] In this case, it follows from Theorem 2.1 (cf. Proposition F.2 for details) that by setting $\delta = 1/T^2$, Algorithm 1 is sub-Chebychev with parameter

$$R = \widetilde{O}\Big( \inf_{\gamma > 0} \{ \mathsf{dec}_\gamma(\mathrm{co}(\mathcal{M})) \cdot T + \gamma \cdot \log(|\Pi|) \} \Big) = \widetilde{O}(\mathsf{dec}_{\gamma_u}(\mathrm{co}(\mathcal{M})) \cdot T), \tag{18}$$

where $\gamma_u$ satisfies the balance $\mathsf{dec}_{\gamma_u}(\mathrm{co}(\mathcal{M})) \propto \frac{\gamma_u}{T} \log|\Pi|$. On the other hand, the lower bound in (17) can be shown to scale with

$$R \geq \widetilde{\Omega}\Big( \mathsf{dec}_{\gamma_\ell,\varepsilon_{\gamma_\ell}}(\mathrm{co}(\mathcal{M})) \cdot T \Big), \tag{19}$$

where $\gamma_\ell$ satisfies the balance $\mathsf{dec}_{\gamma_\ell,\varepsilon_{\gamma_\ell}}(\mathrm{co}(\mathcal{M})) \propto \frac{\gamma_\ell}{T}$. We conclude that the upper bound from Theorem 2.1 cannot be improved beyond (i) localization and (ii) dependence on $\log|\Pi|$.

As an example, we show in Appendix D.3 that for the multi-armed bandit problem with $\Pi = \{1, \ldots, A\}$, the upper bound in (18) yields $R = \widetilde{O}(\sqrt{AT \log A})$, while the lower bound in (19) yields $R = \Omega(\sqrt{AT})$. See Appendix D for additional examples which further illustrate the scaling in the upper and lower bounds.

---

[6]Recall (Appendix B) that $M(\cdot, \cdot \mid \pi)$ is the conditional distribution given $\pi$.

[7]The dominant term $\mathsf{dec}_{\gamma,\varepsilon_\gamma}(\mathrm{co}(\mathcal{M})) \cdot T$ in (15) scales with $T^{1/2}$ for any class that is non-trivial in the sense that it embeds the two-armed bandit problem, so that the $-O(T^{1/2})$ term can be discarded.

The dependence on $\log|\Pi|$ cannot be removed from the upper bound or made to appear in the lower bound in general (cf. Section 3.5 of Foster et al. [17]). As shown in Foster et al. [17], localization is inconsequential for most model classes commonly studied in the literature. The same is true for the examples we consider here (Appendix D), where Theorem 2.2 leads to the correct rate up to small polynomial factors. However, improving the upper bound to achieve localization, which Foster et al. [17] show is possible in the stochastic setting, is an interesting future direction.

See Appendix A for further discussion and for comparison to a related lower bound in Lattimore [31].

**Why convexity?** At this point, a natural question is *why* the convex hull $\mathrm{co}(\mathcal{M})$ plays a fundamental role in the adversarial setting. For the lower bound, the intuition is simple: Given a model class $\mathcal{M}$, the adversary can pick any mixture distribution $\mu \in \Delta(\mathcal{M})$, then choose the sequence of models $M^{(1)}, \ldots, M^{(T)}$ by sampling $M^{(t)} \sim \mu$ independently at each round. This is equivalent to playing a static mixture model $M^\star = \mathbb{E}_{M \sim \mu}[M] \in \mathrm{co}(\mathcal{M})$, which is what allows us to prove a lower bound based on the DEC for the set $\mathrm{co}(\mathcal{M})$ of all such models. In view of the fact that the lower bound is obtained through this static (and stochastic) adversary, we believe the more surprising result here is that good behavior of the convexified DEC is also *sufficient* for low regret for fully adversarial decision making.

## 2.3 Learnability and Comparison to Stochastic Setting

Building on the upper and lower bounds in the prequel, we give a characterization for *learnability* (i.e., when non-trivial regret is possible) for adversarial decision making. This extends the learnability characterization for the stochastic setting in Foster et al. [17], and follows a long tradition in learning theory [52, 2, 49, 44, 11]. To state the result, we define the minimax regret for model class $\mathcal{M}$ as

$$\mathfrak{M}(\mathcal{M}, T) = \inf_{\boldsymbol{p}^{(1)}, \ldots, \boldsymbol{p}^{(T)}} \sup_{\boldsymbol{M}^{(1)}, \ldots, \boldsymbol{M}^{(T)}} \mathbb{E}[\mathbf{Reg}_{\mathsf{DM}}],$$

where $\boldsymbol{p}^{(t)} : (\Pi \times \mathcal{Z})^{t-1} \to \Delta(\Pi)$ and $\boldsymbol{M}^{(t)} : (\Pi \times \mathcal{Z})^{t-1} \to \mathcal{M}$ are policies for the learner and adversary, respectively. Our characterization is as follows.

**Theorem 2.3.** *Suppose there exists $M_0 \in \mathcal{M}$ such that $f^{M_0}$ is a constant function, and that $|\Pi| < \infty$.*

1. *If there exists $\rho > 0$ such that $\lim_{\gamma \to \infty} \mathsf{dec}_\gamma(\mathrm{co}(\mathcal{M})) \cdot \gamma^\rho = 0$, then $\lim_{T \to \infty} \frac{\mathfrak{M}(\mathcal{M}, T)}{T^p} = 0$ for $p < 1$.*

2. *If $\lim_{\gamma \to \infty} \mathsf{dec}_\gamma(\mathrm{co}(\mathcal{M})) \cdot \gamma^\rho > 0$ for all $\rho > 0$, then $\lim_{T \to \infty} \frac{\mathfrak{M}(\mathcal{M}, T)}{T^p} = \infty$ for all $p < 1$.*

*The same conclusion holds when $\Pi = \Pi_T$ grows with $T$, but has $\log|\Pi_T| = O(T^q)$ for any $q < 1$.[8]*

Theorem 2.3 shows that polynomial decay of the convexified DEC is necessary and sufficient for low regret. We emphasize that this result is complementary to Theorem 2.2, and does not require localization or any assumption on the tail behavior of the algorithm. This is a consequence of the coarse, asymptotic nature of the result, which allows us the use of rescaling arguments to remove these conditions.

**Comparison to stochastic setting.** Having shown that the convexified Decision-Estimation Coefficient leads to upper and lower bounds on the optimal regret for the adversarial DMSO setting, we now contrast with the stochastic setting. There, Foster et al. [17] obtain upper bounds on regret that have the same form as (11), but scale with the weaker quantity $\max_{\overline{M} \in \mathrm{co}(\mathcal{M})} \mathsf{dec}_\gamma(\mathcal{M}, \overline{M})$.[9] For classes that are not convex, but where "proper" estimators are available (including finite-state/action MDPs), the upper bounds in Foster et al. [17] can further be improved to scale with $\mathsf{dec}_\gamma(\mathcal{M})$. Hence, our results show that in general, the price of adversarial outcomes can be as large as $\mathsf{dec}_\gamma(\mathrm{co}(\mathcal{M}))/\mathsf{dec}_\gamma(\mathcal{M})$. Examples (see Appendix D for details and more) include:

- For tabular (finite-state/action) MDPs with horizon $H$, $S$ states, and $A$ actions, Foster et al. [17] show that $\mathsf{dec}_\gamma(\mathcal{M}) \leq \mathrm{poly}(H, S, A)/\gamma$, and use this to obtain regret

---

[8]Allowing $\Pi$ to grow with $T$ is useful when considering infinite decision spaces, because it facilitates covering arguments.

[9]Theorem 3.1 of Foster et al. [17] attains $\mathbf{Reg}_{\mathsf{DM}} \lesssim \inf_{\gamma > 0} \{ \max_{\overline{M} \in \mathrm{co}(\mathcal{M})} \mathsf{dec}_\gamma(\mathcal{M}, \overline{M}) + \gamma \cdot \log|\mathcal{M}| \}$ with high probability.

$\sqrt{\text{poly}(H, S, A) \cdot T}$. Tabular MDPs are *not* a convex class, and $\text{co}(\mathcal{M})$ is equivalent to the class of so-called *latent MDPs*, which are known to be intractable [28, 37]. Indeed, we show (Appendix D) that $\text{dec}_\gamma(\text{co}(\mathcal{M})) \geq \Omega(A^{\min\{S,H\}})$, which implies an exponential lower bound on regret through Theorem 2.2. This highlights that in general, the gap between stochastic and adversarial outcomes can be quite large.

- For many common bandit problems, one has $\text{co}(\mathcal{M}) = \mathcal{M}$, leading to polynomial bounds on regret in the adversarial setting. For example, the multi-armed bandit problem with $A$ actions has $\text{dec}_\gamma(\text{co}(\mathcal{M})) \leq O(A/\gamma)$, leading to $\sqrt{AT \log A}$ regret (via Theorem 2.1), and the linear bandit problem in $d$ dimensions has $\text{dec}_\gamma(\text{co}(\mathcal{M})) \leq O(d/\gamma)$, leading to regret $\sqrt{dT \log|\Pi|}$.

## 3   Connections Between Complexity Measures

The Decision-Estimation Coefficient bears a resemblance to the *generalized Information Ratio* introduced by Lattimore and György [32], Lattimore [30] which extends the original Information Ratio of Russo and Van Roy [46, 47]. In what follows, we establish deeper connections between these complexity measures. All of the results in this section are proven in Appendix E.

Let us call any function $D(\cdot \parallel \cdot) : \Delta(\Pi) \times \Delta(\Pi) \to \mathbb{R}^+$ a *divergence-like function*. We restate the generalized Information Ratio from Lattimore [31]. For a distribution $\mu \in \Delta(\mathcal{M} \times \Pi)$ and decision distribution $p \in \Delta(\Pi)$, let $\mathbb{P}$ be the law of the process $(M, \pi^\star) \sim \mu, \pi \sim p, z \sim M(\pi)$, and define $\mu_{\text{pr}}(\pi') := \mathbb{P}(\pi^\star = \pi')$ and $\mu_{\text{po}}(\pi'; \pi, z) := \mathbb{P}(\pi^\star = \pi' \mid (\pi, z))$. The distribution $\mu_{\text{pr}}$ should be thought of as the prior over $\pi^\star$, and $\mu_{\text{po}}$ should be thought of as the posterior over $\pi^\star$ after observing $(z, \pi)$; note that the law $\mu_{\text{po}}$ does not depend on the distribution $p$. For parameter $\lambda > 1$, Lattimore [31] defines the generalized Information Ratio for a class $\mathcal{M}$ via[10]

$$\Psi_\lambda(\mathcal{M}) = \sup_{\mu \in \Delta(\mathcal{M} \times \Pi)} \inf_{p \in \Delta(\Pi)} \left\{ \frac{(\mathbb{E}_{(M,\pi^\star)\sim\mu} \mathbb{E}_{\pi\sim p}[f^M(\pi^\star) - f^M(\pi)])^\lambda}{\mathbb{E}_{\pi\sim p} \mathbb{E}_{z|\pi}[D(\mu_{\text{po}}(\cdot; \pi, z) \parallel \mu_{\text{pr}})]} \right\}. \tag{20}$$

Here, we have slightly generalized the original definition in Lattimore [31] by incorporating models in $\mathcal{M}$ rather than placing an arbitrary prior over observations $z$ directly. We also use a general divergence-like function, while Lattimore [31] uses KL divergence and Lattimore and György [32] use Bregman divergences.

To understand the connection to the Decision-Estimation Coefficient, it will be helpful to introduce another variant of the Information Ratio, which we call the *parameterized Information Ratio*.

**Definition 3.1.** *For a divergence-like function $D(\cdot \parallel \cdot) : \Delta(\Pi) \times \Delta(\Pi) \to \mathbb{R}_+$, the* parameterized Information Ratio *is given by*

$$\inf{}_\gamma^D(\mathcal{M}) \tag{21}$$
$$= \sup_{\mu \in \Delta(\mathcal{M} \times \Pi)} \inf_{p \in \Delta(\Pi)} \mathbb{E}_{\pi\sim p} \big[ \mathbb{E}_{(M,\pi^\star)\sim\mu}[f^M(\pi^\star) - f^M(\pi)] - \gamma \cdot \mathbb{E}_{\pi\sim p} \mathbb{E}_{z|\pi}[D(\mu_{\text{po}}(\cdot; \pi, z) \parallel \mu_{\text{pr}})] \big].$$

The parameterized Information Ratio is always bounded by the generalized Information Ratio in (20); in particular, we have $\inf_\gamma^D(\mathcal{M}) \leq (\Psi_\lambda(\mathcal{M})/\gamma)^{\frac{1}{\lambda-1}} \ \forall \gamma > 0$. All regret bounds based on the generalized Information Ratio that we are aware of [32, 31] implicitly bound regret by the parameterized Information Ratio, and then invoke the inequality above to move to the generalized Information Ratio. In general though, it does not appear that these notions are equivalent. Informally, this is because the notion in (20) is equivalent to requiring that a single distribution $p$ certify a certain bound on the value in (21) for all values of the parameter $\gamma$ simultaneously, while the parameterized Information Ratio allows the distribution $p$ to vary as a function of $\gamma > 0$ (hence the name); see also Appendix E.

Letting $\inf_\gamma^H(\mathcal{M})$ denote the parameterized Information Ratio with $D = D_H^2(\cdot, \cdot)$, we show that this notion is equivalent to the convexified Decision-Estimation Coefficient.

**Theorem 3.1.** *For all $\gamma > 0$, $\inf_\gamma^H(\mathcal{M}) \leq \text{dec}_\gamma(\text{co}(\mathcal{M})) \leq \inf_{\gamma/4}^H(\mathcal{M})$.*

---

[10]Lattimore and György [32] give a slightly different but essentially equivalent definition; cf. Appendix E.

This result is a special case of Theorem E.1 in Appendix E, which shows that a similar equivalence holds for a class of "well-behaved" $f$-divergences that includes KL divergence, but not necessarily for general Bregman divergences. The idea is to use Bayes' law to move from the Decision-Estimation Coefficient, which considers distance between *distributions over observations*, to the Information Ratio, which considers distance between *distributions over decisions*.

In light of this characterization, the results in this paper could have equivalently been presented in terms of the parameterized Information Ratio. We chose to present them in terms of the Decision-Estimation Coefficient in order to draw parallels to the stochastic setting, where guarantees that scale with $\mathsf{dec}_\gamma(\mathcal{M})$ (without convexification) are available. It is unclear whether the Information Ratio can accurately reflect the complexity for both stochastic and adversarial settings in the same fashion, because—unlike the DEC—it is invariant under convexification, as the following proposition shows.[11]

**Proposition 3.1.** *For any divergence-like function $D(\cdot \parallel \cdot)$, we have*

$$\mathsf{inf}_\gamma^D(\mathcal{M}) = \mathsf{inf}_\gamma^D(\mathrm{co}(\mathcal{M})), \quad \forall \gamma > 0.$$

For a final structural result, we show that up to constants, the parameterized Information Ratio is equivalent to the high-probability Exploration-by-Optimization objective.

**Theorem 3.2.** *For all $\eta > 0$,*

$$\mathsf{inf}_{\eta^{-1}}^{\mathsf{H}}(\mathcal{M}) \leq \mathsf{exo}_\eta(\mathcal{M}) \leq \mathsf{inf}_{(8\eta)^{-1}}^{\mathsf{H}}(\mathcal{M})$$

This result is proven through a direct argument (cf. Section 2.1), and the equivalence of the DEC and Exploration-by-Optimization in (13) is proven by combining with Theorem 3.1. Summarizing the equivalence:

**Corollary 3.1.** For all $\eta > 0$,

$$\mathsf{dec}_{(4\eta)^{-1}}(\mathrm{co}(\mathcal{M})) \leq \mathsf{inf}_{\eta^{-1}}^{\mathsf{H}}(\mathcal{M}) \leq \mathsf{exo}_\eta(\mathcal{M}) \leq \mathsf{inf}_{(8\eta)^{-1}}^{\mathsf{H}}(\mathcal{M}) \leq \mathsf{dec}_{(8\eta)^{-1}}(\mathrm{co}(\mathcal{M})).$$

Since this equivalence depends of the value of the parameter $\gamma > 0$ in the parameterized Information Ratio, it is not clear whether a similar equivalence can be established using the generalized Information Ratio in (20). We note in passing that one can use similar arguments to lower bound the Bregman divergence-based Exploration-by-Optimization objective in Lattimore and György [32] by the parameterized Information Ratio for the Bregman divergence of interest, complementing their upper bound.

## 4 Discussion

We have shown that the convexified Decision-Estimation Coefficient is necessary and sufficient to achieve low regret for adversarial interactive decision making, establishing that convexity governs the price of adversarial outcomes. Our results elucidate the relationship between the DEC, Exploration-by-Optimization, and the Information Ratio, and we hope they will find broader use.

This work adds to a growing body of research which shows that online reinforcement learning with agnostic or adversarial outcomes can be statistically intractable [48, 37]. A promising future direction is to extend our techniques to natural semi-adversarial models in which reinforcement learning is tractable. Another interesting direction is to address the issue of computational efficiency for large decision spaces.

#### Acknowledgements

We thank Zak Mhammedi for useful comments and feedback. AR acknowledges support from the ONR through awards N00014-20-1-2336 and N00014-20-1-2394, from the ARO through award W911NF-21-1-0328, and from the DOE through award DE-SC0022199. Part of this work was completed while DF was visiting the Simons Institute for the Theory of Computing. KS acknowledges support from NSF CAREER Award 1750575.

---

[11]The variants in Lattimore and György [32], Lattimore [31] are also invariant under convexification.

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
