# On the Complexity of Adversarial Decision Making

## Abstract

A central problem in online learning and decision making—from bandits to reinforcement learning—is to understand what modeling assumptions lead to sample-efficient learning guarantees. With a focus on stochastic environments, a recent line of research provides general structural conditions under which sample-efficient learning is possible, but robust learning guarantees for agnostic or adversarial settings have remained elusive. We consider a general *adversarial decision making* framework that encompasses (structured) bandit problems with adversarial rewards and reinforcement learning problems with adversarial dynamics. Our main result is to show—via new upper and lower bounds—that the Decision-Estimation Coefficient, a complexity measure introduced by Foster et al. [18] in the stochastic counterpart to our setting, is both necessary and sufficient for low regret in the adversarial setting. However, compared to the stochastic setting, one must apply the Decision-Estimation Coefficient to the *convex hull* of the class of models (or, hypotheses) under consideration. This establishes that the price of accommodating adversarial rewards or dynamics is governed by the behavior of the model class under convexification, and recovers a number of existing results—both positive and negative. En route to obtaining these guarantees, we provide new structural results that connect the Decision-Estimation Coefficient to variants of other well-known complexity measures, including the Information Ratio of Russo and Van Roy [52] and the Exploration-by-Optimization objective of Lattimore and György [34].

## 1 Introduction

We consider the problem of robust data-driven decision making in bandits, reinforcement learning, and beyond. The last decade has seen development of data-driven decision algorithms with strong empirical performance in domains including robotics [28, 40], dialogue systems [38], and personalization [2, 57]. Reliably deploying data-driven decision making methods in safety-critical systems requires principled algorithms with

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

 \mathscr{R} \otimes \mathscr{O}} \left\{ \frac{M(A|\pi)}{M'(A|\pi)} \right\} \vee e$.[5]

**Theorem 2.2** (Main lower bound). *Let $C(T) := c \cdot \log(T \wedge V(\mathcal{M}))$ for a sufficiently large numerical constant $c > 0$. Set $\varepsilon_\gamma := \frac{\gamma}{4C(T)T}$. For any algorithm, there exists an oblivious adversary for which*

$$\mathbb{E}[\mathbf{Reg}_{\mathsf{DM}}] + \sqrt{\mathbb{E}(\mathbf{Reg}_{\mathsf{DM}})_+^2} \ge \Omega(1) \cdot \sup_{\gamma > \sqrt{2C(T)T}} \mathsf{dec}_{\gamma, \varepsilon_\gamma}(\mathrm{co}(\mathcal{M})) \cdot T - O(T^{1/2}). \tag{13}$$

Theorem 2.2 implies that for any algorithm with "reasonable" tail behavior beyond what is granted by control of the first moment (such as Algorithm 1), the regret in Theorem 2.1 cannot be substantially improved. In more detail, consider the notion of a *sub-Chebychev* algorithm.

**Definition 2.1** (Sub-Chebychev Algorithm). *We say that a regret minimization algorithm is sub-Chebychev with parameter $R$ if for all $t > 0$,*

$$\mathbb{P}((\mathbf{Reg}_{\mathsf{DM}})_+ \ge t) \le R^2/t^2. \tag{14}$$

For sub-Chebychev algorithms, both the mean and (root) second moment of regret are bounded by the parameter $R$ (cf. Appendix F.4), which has the following consequence.

---

[5]Recall (Appendix B) that $M(\cdot, \cdot \mid \pi)$ is the conditional distribution given $\pi$; finiteness of $V(\mathcal{M})$ is not necessary, but removes a $\log(T)$ factor from Theorem 2.2.

**Corollary 2.1.** Any regret minimization algorithm with sub-Chebychev parameter $R > 0$ must have

$$R \geq \widetilde{\Omega}(1) \cdot \sup_{\gamma > \sqrt{2C(T)T}} \mathsf{dec}_{\gamma, \varepsilon_\gamma}(\mathrm{co}(\mathcal{M})) \cdot T - O(T^{1/2}). \tag{15}$$

To interpret this result, suppose for simplicity that $\mathsf{dec}_\gamma(\mathrm{co}(\mathcal{M}))$ and $\mathsf{dec}_{\gamma, \varepsilon_\gamma}(\mathrm{co}(\mathcal{M}))$ are continuous with respect to $\gamma > 0$, and that $\mathsf{dec}_{\gamma, \varepsilon_\gamma}(\mathrm{co}(\mathcal{M})) \gtrsim \gamma^{-1}$, which is satisfied for all non-trivial classes.[6] In this case, one can show (cf. Proposition F.2 for a proof) that by setting $\delta = 1/T^2$, Theorem 2.1 implies that Algorithm 1 is sub-Chebychev with parameter

$$R = \widetilde{O}\Big( \inf_{\gamma > 0} \{ \mathsf{dec}_\gamma(\mathrm{co}(\mathcal{M})) \cdot T + \gamma \cdot \log(|\Pi|) \} \Big) = \widetilde{O}(\mathsf{dec}_{\gamma_u}(\mathrm{co}(\mathcal{M})) \cdot T), \tag{16}$$

where $\gamma_u$ satisfies the balance $\mathsf{dec}_{\gamma_u}(\mathrm{co}(\mathcal{M})) \propto \frac{\gamma_u}{T} \log|\Pi|$. On the other hand, the lower bound in (15) can be shown to scale with

$$R \geq \widetilde{\Omega}\Big( \mathsf{dec}_{\gamma_\ell, \varepsilon_{\gamma_\ell}}(\mathrm{co}(\mathcal{M})) \cdot T \Big), \tag{17}$$

where $\gamma_\ell$ satisfies the balance $\mathsf{dec}_{\gamma_\ell, \varepsilon_{\gamma_\ell}}(\mathrm{co}(\mathcal{M})) \propto \frac{\gamma_\ell}{T}$. We conclude that the upper bound from Theorem 2.1 cannot be improved beyond (i) localization and (ii) dependence on $\log|\Pi|$.

As an example, we show in Appendix D.3 that for the multi-armed bandit problem with $\Pi = \{1, \ldots, A\}$, the upper bound in (16) yields $R = O(\sqrt{AT \log A})$, while the lower bound in (17) yields $R = \Omega(\sqrt{AT})$. See Appendix D for additional examples which further illustrate the scaling above.

The dependence on $\log|\Pi|$ cannot be removed from the upper bound or made to appear in the lower bound in general (cf. Section 3.5 of Foster et al. [18]). As shown in Foster et al. [18], localization is inconsequential for essentially all model classes commonly studied in the literature, and the same is true for the examples we consider here (Appendix D), where Theorem 2.2 leads to the correct rate up to small polynomial factors. However, improving the upper bound to achieve localization (which Foster et al. [18] show is possible in the stochastic setting) is an interesting future direction.

See Appendix A for further discussion and for comparison to a related lower bound in Lattimore [33].

**Why convexity?** At this point, a natural question is *why* the convex hull $\mathrm{co}(\mathcal{M})$ plays a fundamental role in the adversarial setting. For the lower bound, the intuition is simple: Given a model class $\mathcal{M}$, the adversary can pick any mixture distribution $\mu \in \Delta(\mathcal{M})$, then choose the sequence of models $M^{(1)}, \ldots, M^{(T)}$ by sampling $M^{(t)} \sim \mu$ independently at each round. This is equivalent to playing a static mixture model $M^\star = \mathbb{E}_{M \sim \mu}[M] \in \mathrm{co}(\mathcal{M})$, which is what allows us to prove a lower bound based on the DEC for the set $\mathrm{co}(\mathcal{M})$ of all such models. In view of the fact that the lower bound is obtained through this static, stochastic adversary, we believe the more surprising result here is that good behavior of the convexified DEC is also *sufficient* for low regret.

## 2.3 Learnability and Comparison to Stochastic Setting

Building on the upper and lower bounds in the prequel, we give a characterization for *learnability* (i.e., when non-trivial regret is possible) in the adversarial setting. This extends the learnability result for the stochastic setting in Foster et al. [18], and follows a long tradition of such characterizations in learning theory [58, 3, 54, 49, 12]. To state the result, we define the minimax regret as

$$\mathfrak{M}(\mathcal{M}, T) = \inf_{p^{(1)}, \ldots, p^{(T)}} \sup_{M^{(1)}, \ldots, M^{(T)}} \mathbb{E}[\mathbf{Reg}_{\mathrm{DM}}],$$

where $p^{(t)} : (\Pi \times \mathcal{Z})^{t-1} \to \Delta(\Pi)$ and $M^{(t)} : (\Pi \times \mathcal{Z})^{t-1} \to \mathcal{M}$ are policies for the learner and adversary, respectively. Our characterization is as follows.

**Theorem 2.3.** *Suppose there exists $M_0 \in \mathcal{M}$ such that $f^{M_0}$ is a constant function, and that $|\Pi| < \infty$.*

1. *If there exists $\rho > 0$ s.t. $\lim_{\gamma \to \infty} \mathsf{dec}_\gamma(\mathrm{co}(\mathcal{M})) \cdot \gamma^\rho = 0$, then $\lim_{T \to \infty} \frac{\mathfrak{M}(\mathcal{M}, T)}{T^p} = 0$ for $p < 1$.*

2. *If $\lim_{\gamma \to \infty} \mathsf{dec}_\gamma(\mathrm{co}(\mathcal{M})) \cdot \gamma^\rho > 0$ for all $\rho > 0$, then $\lim_{T \to \infty} \frac{\mathfrak{M}(\mathcal{M}, T)}{T^p} = \infty$ for all $p < 1$.*

---

[6]Note that the dominant term $\mathsf{dec}_{\gamma, \varepsilon_\gamma}(\mathrm{co}(\mathcal{M})) \cdot T$ in (13) scales with $\sqrt{T}$ any "non-trivial" class that embeds the two-armed bandit problem, so that the $-O(T^{1/2})$ term can be discarded.

 *The same conclusion holds when $\Pi = \Pi_T$ grows with $T$, but has $\log|\Pi_T| = O(T^q)$ for any $q < 1$.*[7]

Theorem 2.3 shows that polynomial decay of the convexified DEC is necessary and sufficient for low regret. We emphasize that this result is complementary to Theorem 2.2, and does not require localization or any assumption on the tail behavior of the algorithm. This is a consequence of the coarse, asymptotic nature of the result, which allows us to perform rescaling tricks to remove these conditions.

**Comparison to stochastic setting.** Having shown that the convexified Decision-Estimation Coefficient plays a fundamental role in determining the optimal regret for the adversarial DMSO setting, now is a good time to make comparisons to the stochastic setting. There, Foster et al. [18] obtain upper bounds on regret that have the same form as (9), but scale with the weaker quantity $\max_{\overline{M} \in \mathrm{co}(\mathcal{M})} \mathsf{dec}_\gamma(\mathcal{M}, \overline{M})$.[8] For classes that are not convex, but where "proper" estimators are available (e.g., tabular MDPs), the upper bounds in Foster et al. [18] can further be improved to scale with $\mathsf{dec}_\gamma(\mathcal{M})$. Hence, our results show that in general, the price of adversarial outcomes can be as large as $\mathsf{dec}_\gamma(\mathrm{co}(\mathcal{M}))/\mathsf{dec}_\gamma(\mathcal{M})$. Examples (see Appendix D for details and more) include:

- For tabular MDPs with horizon $H$, $S$ states, and $A$ actions, Foster et al. [18] show that $\mathsf{dec}_\gamma(\mathcal{M}) = \mathrm{poly}(H, S, A)/\gamma$, and use this to obtain regret $\sqrt{\mathrm{poly}(H, S, A) \cdot T}$. Tabular MDPs are *not* a convex class, and $\mathrm{co}(\mathcal{M})$ is equivalent to the class of so-called *latent MDPs*, which are known to be intractable [30, 41]. Indeed, we show (Appendix D) that $\mathsf{dec}_\gamma(\mathrm{co}(\mathcal{M})) \geq \Omega(A^{\min\{S,H\}})$. This example highlights that in general, the gap between stochastic and adversarial can be quite large.

- For many common bandit problems, one has $\mathrm{co}(\mathcal{M}) = \mathcal{M}$, leading to polynomial bounds on regret in the adversarial setting. For example the multi-armed bandit problem with $A$ actions has $\mathsf{dec}_\gamma(\mathrm{co}(\mathcal{M})) \leq O(A/\gamma)$, leading to $\sqrt{AT \log A}$ regret from Theorem 2.1, and the linear bandit problem in $d$ dimensions has $\mathsf{dec}_\gamma(\mathrm{co}(\mathcal{M})) \leq O(d/\gamma)$, leading to regret $\sqrt{dT \log|\Pi|}$.

# 3 Connections Between Complexity Measures

The Decision-Estimation Coefficient bears a resemblance to the notion of *generalized information ratio* introduced by Lattimore and György [34], Lattimore [32] which extends the original information ratio of Russo and Van Roy [51, 52]. In what follows, we establish deeper connections between these complexity measures. All of the results in this section are proven in Appendix E.

Let us recall the definition of the generalized information ratio from Lattimore [32], which we state here for a general divergence-like function $D(\cdot \parallel \cdot) \to \mathbb{R}^+$ (typically, KL divergence or another Bregman divergence). For a distribution $\mu \in \Delta(\mathcal{M} \times \Pi)$ and decision distribution $p \in \Delta(\Pi)$, define $\mu_{\mathrm{pr}}(\pi') := \mathbb{P}(\pi^\star = \pi')$ and $\mu_{\mathrm{po}}(\pi'; \pi, z) := \mathbb{P}(\pi^\star = \pi' \mid (\pi, z))$, where $\mathbb{P}$ is the law of the process $(M, \pi^\star) \sim \mu, \pi \sim p, z \sim M(\pi)$. $\mu_{\mathrm{pr}}$ should be thought of as the prior over $\pi^\star$, and $\mu_{\mathrm{po}}$ as the posterior having observed $(z, \pi)$; note that the law $\mu_{\mathrm{po}}$ does not depend on the distribution $p$. For parameter $\lambda > 1$, Lattimore [33] defines the generalized information ratio for a class $\mathcal{M}$ via[9]

$$\Psi_\lambda(\mathcal{M}) = \sup_{\mu \in \Delta(\mathcal{M} \times \Pi)} \inf_{p \in \Delta(\Pi)} \left\{ \frac{(\mathbb{E}_{(M,\pi^\star) \sim \mu} \mathbb{E}_{\pi \sim p}[f^M(\pi^\star) - f^M(\pi)])^\lambda}{\mathbb{E}_{\pi \sim p} \mathbb{E}_{z|\pi}[D(\mu_{\mathrm{po}}(\cdot; \pi, z) \parallel \mu_{\mathrm{pr}})]} \right\}. \tag{18}$$

Here, we have slightly generalized the original definition in Lattimore [33] by incorporating models in $\mathcal{M}$ rather than placing an arbitrary prior over observations $z$ directly. We also use a general divergence, while Lattimore [33] uses KL divergence and Lattimore and György [34] use Bregman divergences.

To understand the connection to the Decision-Estimation Coefficient, it will be helpful introduce another variant of the information ratio that we call the *parameterized information ratio*.

**Definition 3.1.** *For a divergence $D(\cdot \parallel \cdot)$, the* parameterized information ratio *is given by*

$$\mathsf{inf}_\gamma^D(\mathcal{M}) \tag{19}$$
$$= \sup_{\mu \in \Delta(\mathcal{M} \times \Pi)} \inf_{p \in \Delta(\Pi)} \mathbb{E}_{\pi \sim p}\left[\mathbb{E}_{(M,\pi^\star) \sim \mu}[f^M(\pi^\star) - f^M(\pi)] - \gamma \cdot \mathbb{E}_{\pi \sim p} \mathbb{E}_{z|\pi}[D(\mu_{\mathrm{po}}(\cdot; \pi, z) \parallel \mu_{\mathrm{pr}})]\right].$$

---

[7]Allowing $\Pi$ to grow with $T$ can be used to handle infinite decision spaces using covering arguments.

[8]Theorem 3.1 of Foster et al. [18] attains $\mathbf{Reg}_{\mathsf{DM}} \lesssim \inf_{\gamma>0}\{\max_{\overline{M} \in \mathrm{co}(\mathcal{M})} \mathsf{dec}_\gamma(\mathcal{M}, \overline{M}) + \gamma \cdot \log|\mathcal{M}|\}$.

[9]Lattimore and György [34] give a slightly different but essentially equivalent definition; cf. Appendix E.

The parameterized information ratio is always bounded by the generalized information ratio in (18); in particular, we have $\inf_\gamma^D(\mathcal{M}) \leq (\Psi_\lambda(\mathcal{M})/\gamma)^{\frac{1}{\lambda-1}} \; \forall \gamma > 0$. All of the regret bounds based on the generalized information ratio that we are aware of [34, 33] implicitly bound regret by the parameterized information ratio, and then invoke the inequality above to move to the generalized information ratio. In general though, it does not appear that these notions are equivalent. Informally, this is because the notion in (18) is equivalent to requiring that a single distribution $p$ certify a certain bound on the value in (19) for all values of the parameter $\gamma$ simultaneously, while the parameterized information ratio allows the distribution $p$ to vary as a function of $\gamma > 0$ (hence the name); see also Appendix E.

Letting $\inf_\gamma^H(\mathcal{M})$ denote the parameterized information ratio with $D = D_H^2(\cdot, \cdot)$, we show that this notion is equivalent to the convexified Decision-Estimation Coefficient.

**Theorem 3.1.** *For all $\gamma > 0$, $\inf_\gamma^H(\mathcal{M}) \leq \mathsf{dec}_\gamma(\mathrm{co}(\mathcal{M})) \leq \inf_{\gamma/4}^H(\mathcal{M})$.*

This result is a special case of Theorem E.1 in Appendix E, which shows that a similar equivalence holds for a class of "well-behaved" $f$-divergences that includes KL divergence (but not necessarily for general Bregman divergences). The basic idea is to use Bayes' rule to move from the Decision-Estimation Coefficient, which considers distance between distributions over *observations*, to the information ratio, which considers distance between distributions over *decisions*.

In light of this characterization, the results in this paper could have equivalently been presented in terms of the parameterized information ratio. We chose to present them in terms of the Decision-Estimation Coefficient in order to draw parallels to the stochastic setting, where guarantees that scale with $\mathsf{dec}_\gamma(\mathcal{M})$ (without convexification) are available. It is unclear whether the information ratio can accurately reflect the complexity for both stochastic and adversarial settings in the same fashion, because—unlike the DEC—it is invariant under convexification.[10]

**Proposition 3.1.** *For any divergence-like function $D(\cdot \parallel \cdot) : \Delta(\Pi) \times \Delta(\Pi) \to \mathbb{R}_+$, we have*

$$\inf_\gamma^D(\mathcal{M}) = \inf_\gamma^D(\mathrm{co}(\mathcal{M})), \quad \forall \gamma > 0.$$

For a final str uctural result, we show that up to constants, the parameterized information ratio is equivalent to the high-probability Exploration-by-Optimization objective.

**Theorem 3.2.** *For all $\eta > 0$, $\inf_{\eta^{-1}}^H(\mathcal{M}) \leq \mathsf{exo}_\eta(\mathcal{M}) \leq \inf_{(8\eta)^{-1}}^H(\mathcal{M})$.*

This result is proven through a direct argument, and the equivalence of the DEC and Exploration-by-Optimization in (11) is proven by combining with Theorem 3.1. Summarizing the equivalence:

**Corollary 3.1.** *For all $\eta > 0$,*

$$\mathsf{dec}_{(4\eta)^{-1}}(\mathrm{co}(\mathcal{M})) \leq \inf_{\eta^{-1}}^H(\mathcal{M}) \leq \mathsf{exo}_\eta(\mathcal{M}) \leq \inf_{(8\eta)^{-1}}^H(\mathcal{M}) \leq \mathsf{dec}_{(8\eta)^{-1}}(\mathrm{co}(\mathcal{M})).$$

Since this equivalence depends of the value of the parameter $\gamma > 0$ in the parameterized information ratio, it seems unlikely that a similar equivalence can be established using the generalized information ratio in (18). We note in passing that one can use similar techniques to lower bound the Bregman divergence-based Exploration-by-Optimization objective in Lattimore and György [34] by the parameterized information ratio for the Bregman divergence of interest, complementing their upper bound.

# 4  Discussion

We have shown that the convexified Decision-Estimation Coefficient is necessary and sufficient to achieve low regret for adversarial interactive decision making, establishing that convexity governs the price of adversarial outcomes. Our results elucidate the relationship between the DEC, Exploration-by-Optimization, and the information ratio, and we hope they will find broader use.

Our results add to a growing body of research which shows that online reinforcement learning with agnostic or adversarial outcomes can be statistically intractable [53, 41]. A promising future direction is to extend our techniques to natural semi-adversarial models in which reinforcement learning is tractable (for example, the so-called *adversarially corrupted* setting [42, 19]). Other interesting questions include (i) extending our lower bounds beyond the observable-reward setting and to directly handle expected regret, and (ii) developing computationally efficient algorithms for large decision spaces.

---

[10]The variants in Lattimore and György [34], Lattimore [33] are also invariant under convexification.

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

# Contents of Appendix

## A  Detailed Discussion of Related Work

Beyond Foster et al. [18], which was the starting point for this work, our results build on a long line of research on partial monitoring and the information ratio [51, 52, 35, 31, 32, 34, 33, 25, 26]; most closely related are the works the works of Lattimore and György [34] and Lattimore [33]. Below we discuss and compare to these results in greater detail.

**Comparison to partial monitoring setting.**  Lattimore and György [34], Lattimore [33] and other works in this sequence consider a general partial monitoring setting in which each outcome $z^{(t)}$ is directly chosen by an adversary, and need not contain a reward signal.

- In terms of reward signal, our setting is more restrictive because we assume that $r^{(t)}$ is observed. Our upper bounds in fact paper encompass the more general setting where rewards are not observed by the learner, thus subsuming the partial monitoring problem, but our lower bounds that require that rewards are observed.

- In terms of data generation process, our setting is more general because we restrict to models in a known class in $\mathcal{M}$. This setup recovers the case where $z^{(t)}$ is fully adversarial because we can take $\mathcal{M}$ to consist of point masses over $\mathcal{Z}$ as a special case. However, the model also allows for semi-stochastic adversaries, and for settings like (structured) adversarial MDPs. For example, if all models in $\mathcal{M}$ place $\varepsilon$ probability mass on a particular outcome $z$, any adversary in our model must place $\varepsilon$ mass on this outcome as well.

**Upper bounds.**  On the upper bound side, our results build on the Exploration-by-Optimization technique, which was introduced in Lattimore and Szepesvári [37] and generalized significantly in Lattimore and György [34]. The latter result shows that for a general family of mirror descent-based Exploration-by-Optimization algorithms parameterized by Bregman divergences, the regret can be bounded by a certain generalized information ratio based on the associated Bregman divergence (cf. Appendix E). This approach yields bounds on expected regret with a similar form to Theorem 2.1 (with $\mathrm{dec}_\gamma(\mathrm{co}(\mathcal{M}))$ replaced by the generalized information ratio), but does not appear to yield high-probability bounds (in general, in-expectation regret bounds do not imply high-probability regret bounds; for example, even for multi-armed bandits, the EXP3 algorithm can experience linear regret with constant probability [36]). To develop high-probability regret bounds which complement our lower bounds, we depart from the Bregman divergence-based framework and exploit refined properties of Hellinger distance. We note that the work of Lattimore and Szepesvári [37] also proposes a high-probability Exploration-by-Optimization objective, but it is unclear whether this objective (which precedes the information ratio-based results of Lattimore and György [34]) can be related to the information ratio or Decision-Estimation Coefficient for general models.[11]

**Lower bounds.**  On the lower bound side, we build on the proof strategy from Foster et al. [18]. Our most important technical result is Theorem F.1, which improves upon Theorem 3.1 from Foster et al. [18] even in the stochastic setting, by using a more refined change of measure argument. In particular, Theorem 3.1 of Foster et al. [18] gives a lower bound based on the DEC that holds with low probability, and therefore only provides a meaningful converse to algorithms with sub-Gaussian or sub-exponential tail behavior. Our result provides a meaningful converse to any upper bound with sub-Chebychev tail behavior, which is a significantly weaker assumption. We note that while Theorem 3.2 of Foster et al. [18] provides lower bounds on expected regret without algorithmic assumptions, this result requires a stronger notion of localization than the one we consider here, and it is not clear whether this notion can be achieved algorithmically in general. Of course, proving a lower bound on expected regret that matches our lower bound remains an interesting open problem.

Lastly, we mention recent work of Lattimore [33], which provides lower bounds on regret in a general partial monitoring setting based on a generalized information ratio (cf. Appendix E). This result is somewhat complementary to our lower bound (Theorem 2.2):

- On the positive side, it leads to lower bounds on *expected regret* that are always tight in terms of dependence on $T$, while our result only leads to tight dependence on $T$ if one restricts to sub-Chebychev algorithms.

---

[11]In particular, this objective is based on a Bernstein-type tail bound, which leads to a requirement of boundedness for the estimation functions. We avoid explicitly requiring boundedness using a more specialized tail bound based on Lemma C.1.

- On the negative side, the lower bound is loose in $\mathrm{poly}(|\Pi|)$ factors, while our lower bound is essentially only loose in $\mathrm{poly}(\log|\Pi|)$ factors. As a result, only our lower bound leads to meaningful dependence on problem-dependence parameters such as dimension for models with large action spaces.

In addition, the lower bound in Lattimore [33] applies to the general partial monitoring setting, while our lower bound requires that rewards are observed. An interesting question for future work is to investigate whether the techniques of Lattimore [33] can be combined with our own to get the best of both worlds.

Finally, we mention in passing that the results of Lattimore [33] also imply a learnability characterization similar to Theorem 2.3. However, because these results are polynomially loose in $|\Pi|$, they cannot handle the case in which $\log|\Pi|$ grows polynomially in $T$.

# B Preliminaries

**Basic notation.** For a set $\mathcal{X}$, we let $\Delta(\mathcal{X})$ denote the set of all Radon probability measures over $\mathcal{X}$. We let $\mathrm{co}(\mathcal{X})$ denote the set of all finitely supported convex combinations of elements in $\mathcal{X}$. We use the shorthand $x \vee y = \max\{x, y\}$ and $x \wedge y = \min\{x, y\}$.

We adopt non-asymptotic big-oh notation: For functions $f, g : \mathcal{X} \to \mathbb{R}_+$, we write $f = O(g)$ (resp. $f = \Omega(g)$) if there exists a constant $C > 0$ such that $f(x) \leq Cg(x)$ (resp. $f(x) \geq Cg(x)$) for all $x \in \mathcal{X}$. We write $f = \widetilde{O}(g)$ if $f = O(g \cdot \mathrm{polylog}(T))$, $f = \widetilde{\Omega}(g)$ if $f = \Omega(g/\mathrm{polylog}(T))$, and $f = \widetilde{\Theta}(g)$ if $f = \widetilde{O}(g)$ and $f = \widetilde{\Omega}(g)$. We write $f \propto g$ if $f = \widetilde{\Theta}(g)$.

**Probability spaces.** We formalize the probability spaces for the DMSO framework in the same fashion as Foster et al. [18], which we briefly summarize here. decisions are associated with a measurable space $(\Pi, \mathscr{P})$, rewards are associated with the space $(\mathcal{R}, \mathscr{R})$, and observations are associated with the space $(\mathcal{O}, \mathscr{O})$. The history up to time $t$ is denoted by $\mathcal{H}^{(t)} = (\pi^{(1)}, r^{(1)}, o^{(1)}), \ldots, (\pi^{(t)}, r^{(t)}, o^{(t)})$. We define

$$\Omega^{(t)} = \prod_{i=1}^{t} (\Pi \times \mathcal{R} \times \mathcal{O}), \quad \text{and} \quad \mathscr{F}^{(t)} = \bigotimes_{i=1}^{t} (\mathscr{P} \otimes \mathscr{R} \otimes \mathscr{O})$$

so that $\mathcal{H}^{(t)}$ is associated with the space $(\Omega^{(t)}, \mathscr{F}^{(t)})$.

Formally, a model $M = M(\cdot, \cdot \mid \cdot) \in \mathcal{M}$ is a probability kernel from $(\Pi, \mathscr{P})$ to $(\mathcal{R} \times \mathcal{O}, \mathscr{R} \otimes \mathscr{O})$; we use the convention $M(\pi) = M(\cdot, \cdot \mid \pi)$ throughout the paper.[12] An *algorithm* for horizon $T$ is a sequence $p^{(1)}, \ldots, p^{(T)}$, where $p^{(t)}(\cdot \mid \cdot)$ is a probability kernel from $(\Omega^{(t-1)}, \mathscr{F}^{(t-1)})$ to $(\Pi, \mathscr{P})$.

**Divergences.**

For probability distributions $\mathbb{P}$ and $\mathbb{Q}$ over a measurable space $(\Omega, \mathscr{F})$ with a common dominating measure, we define the total variation distance as

$$D_{\mathsf{TV}}(\mathbb{P}, \mathbb{Q}) = \sup_{A \in \mathscr{F}} |\mathbb{P}(A) - \mathbb{Q}(A)| = \frac{1}{2} \int |d\mathbb{P} - d\mathbb{Q}|.$$

Hellinger distance is defined as

$$D_{\mathsf{H}}^2(\mathbb{P}, \mathbb{Q}) = \int \left( \sqrt{d\mathbb{P}} - \sqrt{d\mathbb{Q}} \right)^2,$$

and Kullback-Leibler divergence is defined as

$$D_{\mathsf{KL}}(\mathbb{P} \| \mathbb{Q}) = \left\{ \begin{array}{ll} \int \log\left(\frac{d\mathbb{P}}{d\mathbb{Q}}\right) d\mathbb{P}, & \mathbb{P} \ll \mathbb{Q}, \\ +\infty, & \text{otherwise.} \end{array} \right.$$

For a convex function $f : (0, \infty) \to \mathbb{R}$, the associated $f$-divergence for measures $\mathbb{P}$ and $\mathbb{Q}$ with $\mathbb{P} \ll \mathbb{Q}$ is given by

$$D_f(\mathbb{P} \| \mathbb{Q}) := \mathbb{E}_{\mathbb{Q}}\left[ f\left(\frac{d\mathbb{P}}{d\mathbb{Q}}\right) \right] \tag{20}$$

---

[12]For measurable spaces $(\mathcal{X}, \mathscr{X})$ and $(\mathcal{Y}, \mathscr{Y})$ a probability kernel $P(\cdot \mid \cdot)$ from $(\mathcal{X}, \mathscr{X})$ to $(\mathcal{Y}, \mathscr{Y})$ has the property that (i) For all $x \in \mathcal{X}$, $P(\cdot \mid x)$ is a probability measure, (ii) for all $Y \in \mathscr{Y}$, $x \mapsto P(Y \mid x)$ is measurable.

whenever $\mathbb{P} \ll \mathbb{Q}$. More generally, defining $p = \frac{d\mathbb{P}}{d\nu}$ and $q = \frac{d\mathbb{Q}}{d\nu}$ for a common dominating measure $\nu$, we have

$$D_f(\mathbb{P} \| \mathbb{Q}) := \int_{q>0} q f\left(\frac{p}{q}\right) d\nu + \mathbb{P}(q=0) \cdot f'(\infty), \tag{21}$$

where $f'(\infty) := \lim_{x \to 0^+} x f(1/x)$.

## C  Technical Tools

### C.1  Tail Bounds

**Lemma C.1** (e.g., Lemma A.4 of Foster et al. [18]). *For any sequence of real-valued random variables $(X_t)_{t \leq T}$ adapted to a filtration $(\mathscr{F}_t)_{t \leq T}$, we have that with probability at least $1 - \delta$,*

$$\sum_{t=1}^{T} X_t \leq \sum_{t=1}^{T} \log\left(\mathbb{E}\left[e^{X_t} \mid \mathscr{F}_{t-1}\right]\right) + \log(\delta^{-1}). \tag{22}$$

### C.2  Minimax Theorem

**Lemma C.2** (Sion's Minimax Theorem [55]). *Let $\mathcal{X}$ and $\mathcal{Y}$ be convex sets in linear topological spaces, and assume $\mathcal{X}$ is compact. Let $F : \mathcal{X} \times \mathcal{Y} \to \mathbb{R}$ be such that (i) $F(x, \cdot)$ is concave and upper semicontinuous over $\mathcal{Y}$ for all $x \in \mathcal{X}$ and (ii) $F(\cdot, y)$ is convex and lower semicontinuous over $\mathcal{X}$ for all $y \in \mathcal{Y}$. Then*

$$\inf_{x \in \mathcal{X}} \sup_{y \in \mathcal{Y}} F(x, y) = \sup_{y \in \mathcal{Y}} \inf_{x \in \mathcal{X}} F(x, y). \tag{23}$$

### C.3  Information Theory

#### C.3.1  Basic Results

**Proposition C.1.** *For any $f$-divergence $D_f(\cdot \| \cdot)$, one has that for any pair of random variables $(X, Y)$ with joint law $\mathbb{P}_{X,Y}$,*

$$\mathbb{E}_{X \sim \mathbb{P}_X}\left[D_f\left(\mathbb{P}_{Y|X} \| \mathbb{P}_Y\right)\right] = \mathbb{E}_{Y \sim \mathbb{P}_Y}\left[D_f\left(\mathbb{P}_{X|Y} \| \mathbb{P}_X\right)\right].$$

**Proof of Proposition C.1.** Recalling that $D_f(\mathbb{P} \| \mathbb{Q}) = \mathbb{E}_{\mathbb{Q}}\left[f\left(\frac{d\mathbb{P}}{d\mathbb{Q}}\right)\right]$ for $\mathbb{P} \ll \mathbb{Q}$, we have

$$\begin{aligned}
\mathbb{E}_{X \sim \mathbb{P}_X}\left[D_f\left(\mathbb{P}_{Y|X} \| \mathbb{P}_Y\right)\right] &= \mathbb{E}_{X \sim \mathbb{P}_X} \mathbb{E}_{Y \sim \mathbb{P}_Y}\left[f\left(\frac{d\mathbb{P}_{Y|X}}{d\mathbb{P}_Y}\right)\right] \\
&= \mathbb{E}_{X \sim \mathbb{P}_X} \mathbb{E}_{Y \sim \mathbb{P}_Y}\left[f\left(\frac{d\mathbb{P}_{X,Y}}{d(\mathbb{P}_X \otimes \mathbb{P}_Y)}\right)\right] \\
&= \mathbb{E}_{Y \sim \mathbb{P}_Y} \mathbb{E}_{X \sim \mathbb{P}_X}\left[f\left(\frac{d\mathbb{P}_{X|Y}}{d\mathbb{P}_X}\right)\right] = \mathbb{E}_{Y \sim \mathbb{P}_Y}\left[D_f\left(\mathbb{P}_{X|Y} \| \mathbb{P}_X\right)\right],
\end{aligned}$$

where we have used that $\mathbb{P}_{Y|X} \ll \mathbb{P}_Y$, $\mathbb{P}_{X|Y} \ll \mathbb{P}_X$, and $\mathbb{P}_{X,Y} \ll \mathbb{P}_X \otimes \mathbb{P}_Y$. $\qquad \square$

#### C.3.2  Change of Measure

**Lemma C.3** (Donsker-Varadhan (e.g., Polyanskiy and Wu [48])). *Let $\mathbb{P}$ and $\mathbb{Q}$ be probability measures on $(\mathcal{X}, \mathscr{F})$. Then*

$$D_{\mathsf{KL}}(\mathbb{P} \| \mathbb{Q}) = \sup_{h:\mathcal{X} \to \mathbb{R}} \{\mathbb{E}_{\mathbb{P}}[h(X)] - \log(\mathbb{E}_{\mathbb{Q}}[\exp(h(X))])\}. \tag{24}$$

**Lemma C.4.** *Let $\mathbb{P}$ and $\mathbb{Q}$ be probability distributions over a measurable space $(\mathcal{X}, \mathscr{F})$. Then for all functions $h : \mathcal{X} \to \mathbb{R}$,*

$$|\mathbb{E}_{\mathbb{P}}[h(X)] - \mathbb{E}_{\mathbb{Q}}[h(X)]| \leq \sqrt{2^{-1}(\mathbb{E}_{\mathbb{P}}[h^2(X)] + \mathbb{E}_{\mathbb{Q}}[h^2(X)]) \cdot D_{\mathsf{H}}^2(\mathbb{P}, \mathbb{Q})}. \tag{25}$$

**Proof of Lemma C.4.** From Polyanskiy and Wu [48], we have that for all functions $h : \mathcal{X} \to \mathbb{R}$, if $\mathbb{P} \ll \mathbb{Q}$,

$$|\mathbb{E}_{\mathbb{P}}[h(X)] - \mathbb{E}_{\mathbb{Q}}[h(X)]| \leq \sqrt{\mathbb{V}_{\mathbb{Q}}[h(X)] \cdot D_{\chi^2}(\mathbb{P} \| \mathbb{Q})} \leq \sqrt{\mathbb{E}_{\mathbb{Q}}[h^2(X)] \cdot D_{\chi^2}(\mathbb{P} \| \mathbb{Q})}, \quad (26)$$

where $D_{\chi^2}(\mathbb{P} \| \mathbb{Q}) := \int \frac{(d\mathbb{P} - d\mathbb{Q})^2}{d\mathbb{Q}}$ and $\mathbb{V}_{\mathbb{Q}}$ denotes the variance under $\mathbb{Q}$. The result follows by using that $D_{\chi^2}\left(\mathbb{P} \| \frac{\mathbb{P}+\mathbb{Q}}{2}\right) \leq D_{\mathsf{H}}^2(\mathbb{P}, \mathbb{Q})$. $\qquad\square$

**Lemma 2.1.** *Let $\mathbb{P}$ and $\mathbb{Q}$ be probability distributions over a measurable space $(\mathcal{X}, \mathcal{F})$. Then*

$$\frac{1}{2} D_{\mathsf{H}}^2(\mathbb{P}, \mathbb{Q}) \leq \sup_{g:\mathcal{X}\to\mathbb{R}} \left\{ 1 - \mathbb{E}_{\mathbb{P}}\left[e^g\right] \cdot \mathbb{E}_{\mathbb{Q}}\left[e^{-g}\right] \right\} \leq D_{\mathsf{H}}^2(\mathbb{P}, \mathbb{Q}). \quad (12)$$

**Proof of Lemma 2.1.** We first show that Hellinger distance is lower bounded by the quantity in (12). Recall that Hellinger distance is the $f$-divergence associated with $f(x) = (1 - \sqrt{x})^2$ (cf. (21)). Let $f^\star(y) := \sup_{x \geq 0}\{xy - f(x)\}$ be the Fenchel dual of $f$, which has the form

$$f^\star(y) = \begin{cases} \frac{y}{1-y}, & y < 1, \\ \infty, & y \geq 1. \end{cases}$$

Using Theorem 7.14 of Polyanskiy [47], we express Hellinger distance as a following variational problem based on the dual:

$$D_{\mathsf{H}}^2(\mathbb{P}, \mathbb{Q}) = \sup_{h:\mathcal{X}\to(-\infty,1)} \{\mathbb{E}_{\mathbb{P}}[h(X)] - \mathbb{E}_{\mathbb{Q}}[f^\star(h(X))]\} = \sup_{h:\mathcal{X}\to(-\infty,1)} \left\{ \mathbb{E}_{\mathbb{P}}[h(X)] - \mathbb{E}_{\mathbb{Q}}\left[\frac{h(X)}{1-h(X)}\right] \right\}.$$

Reparameterizing via $h(X) = 1 - h'(X)$ for $h' : \mathcal{X} \to (0, \infty)$, this gives

$$D_{\mathsf{H}}^2(\mathbb{P}, \mathbb{Q}) = \sup_{h:\mathcal{X}\to(0,\infty)} \left\{ 2 - \mathbb{E}_{\mathbb{P}}[h(X)] - \mathbb{E}_{\mathbb{Q}}\left[\frac{1}{h(X)}\right] \right\}.$$

To conclude, we observe that for any test function $g : \mathcal{X} \to \mathbb{R}$, by setting $h(x) = e^{g(x)} \cdot \mathbb{E}_{\mathbb{Q}}[e^{-g}]$, we have

$$2 - \mathbb{E}_{\mathbb{P}}[h(X)] - \mathbb{E}_{\mathbb{Q}}\left[\frac{1}{h(X)}\right] = 2 - \mathbb{E}_{\mathbb{P}}\left[e^g\right] \cdot \mathbb{E}_{\mathbb{Q}}\left[e^{-g}\right] - \mathbb{E}_{\mathbb{Q}}\left[e^{-g}\right]/\mathbb{E}_{\mathbb{Q}}\left[e^{-g}\right]$$
$$= 1 - \mathbb{E}_{\mathbb{P}}\left[e^g\right] \cdot \mathbb{E}_{\mathbb{Q}}\left[e^{-g}\right],$$

so that

$$D_{\mathsf{H}}^2(\mathbb{P}, \mathbb{Q}) \geq \sup_{g:\mathcal{X}\to\mathbb{R}} \left\{ 1 - \mathbb{E}_{\mathbb{P}}\left[e^g\right] \cdot \mathbb{E}_{\mathbb{Q}}\left[e^{-g}\right] \right\}.$$

We now prove the other direction of the inequality in (12). Let $\nu$ be a common dominating measure for $\mathbb{P}$ and $\mathbb{Q}$, and set $p = \frac{d\mathbb{P}}{d\nu}$ and $q = \frac{d\mathbb{Q}}{d\nu}$. We first consider the case where $p, q > 0$ everywhere. Set $g(x) = \frac{1}{2}\log(q(x)/p(x))$. Then we have $\mathbb{E}_{\mathbb{P}}\left[e^g\right] = \int \sqrt{pq}\,d\nu = 1 - \frac{1}{2}D_{\mathsf{H}}^2(\mathbb{P}, \mathbb{Q})$, and likewise, $\mathbb{E}_{\mathbb{Q}}\left[e^{-g}\right] = \int \sqrt{pq}\,d\nu = 1 - \frac{1}{2}D_{\mathsf{H}}^2(\mathbb{P}, \mathbb{Q})$. As a result,

$$\sup_{g:\mathcal{X}\to\mathbb{R}} \left\{ 1 - \mathbb{E}_{\mathbb{P}}\left[e^g\right] \cdot \mathbb{E}_{\mathbb{Q}}\left[e^{-g}\right] \right\} \geq 1 - (1 - \tfrac{1}{2}D_{\mathsf{H}}^2(\mathbb{P}, \mathbb{Q}))^2 \geq \frac{1}{2}D_{\mathsf{H}}^2(\mathbb{P}, \mathbb{Q}),$$

where we have used that $D_{\mathsf{H}}^2(\mathbb{P}, \mathbb{Q}) \in [0, 2]$. For the general case, one can appeal to Lemma C.5 below and take $\varepsilon \to 0$. $\qquad\square$

The following result is generalization of Lemma 2.1 which shows that up to small approximation error, the lower bound in (12) can be obtained using test functions with small magnitude.

**Lemma C.5.** *Let $\mathbb{P}$ and $\mathbb{Q}$ be probability distributions over a measurable space $(\mathcal{X}, \mathcal{F})$. Then for any $\alpha \geq 1$, we have*

$$\frac{1}{2} D_{\mathsf{H}}^2(\mathbb{P}, \mathbb{Q}) \leq \sup_{g\in\mathcal{G}_\varepsilon} \left\{ 1 - \mathbb{E}_{\mathbb{P}}[e^g] \cdot \mathbb{E}_{\mathbb{Q}}\left[e^{-g}\right] \right\} + 4e^{-\alpha}, \quad (27)$$

*where $\mathcal{G}_\alpha := \{g : \mathcal{X} \to \mathbb{R} \mid \|g\|_\infty \leq \alpha\}$.*

**Proof of Lemma C.5.** Fix $\alpha \geq 1$ and let $\varepsilon := e^{-2\alpha}$. Note that $\varepsilon \in (0, e^{-2})$. Given measures $\mathbb{P}$ and $\mathbb{Q}$, set $\mathbb{P}_\varepsilon = (1-\varepsilon)\mathbb{P} + \varepsilon\mathbb{Q}$ and $\mathbb{Q}_\varepsilon = (1-\varepsilon)\mathbb{Q} + \varepsilon\mathbb{P}$. Consider the test function $g = \frac{1}{2}\log(\frac{d\mathbb{Q}_\varepsilon}{d\mathbb{P}_\varepsilon})$, which has the following properties:

- $\|g\|_\infty \leq \frac{1}{2}\log\left(\frac{1-\varepsilon}{\varepsilon} + \frac{\varepsilon}{1-\varepsilon}\right) \leq \frac{1}{2}\log(\varepsilon^{-1})$, where we have used that $\varepsilon \leq 1/2$. This establishes that $g \in \mathcal{G}_\alpha$.

- $\mathbb{E}_\mathbb{P}\left[e^g\right] \leq (1-\varepsilon)^{-1/2} \int \sqrt{d\mathbb{P}d\mathbb{Q}_\varepsilon} = (1-\varepsilon)^{-1/2}\left(1 - \frac{1}{2}D_\mathsf{H}^2(\mathbb{P}, \mathbb{Q}_\varepsilon)\right)$.

- $\mathbb{E}_\mathbb{Q}\left[e^{-g}\right] \leq (1-\varepsilon)^{-1/2} \int \sqrt{d\mathbb{P}_\varepsilon d\mathbb{Q}} = (1-\varepsilon)^{-1/2}\left(1 - \frac{1}{2}D_\mathsf{H}^2(\mathbb{P}_\varepsilon, \mathbb{Q})\right)$.

Using these bounds, we have

$$\sup_{g:\mathcal{X}\to\mathbb{R}}\left\{1 - \mathbb{E}_\mathbb{P}\left[e^g\right] \cdot \mathbb{E}_\mathbb{Q}\left[e^{-g}\right]\right\} \geq 1 - (1-\varepsilon)^{-1}(1 - \tfrac{1}{2}D_\mathsf{H}^2(\mathbb{P}_\varepsilon, \mathbb{Q}))(1 - \tfrac{1}{2}D_\mathsf{H}^2(\mathbb{P}, \mathbb{Q}_\varepsilon))$$

$$\geq 1 - (1-\varepsilon)^{-1}(1 - \tfrac{1}{2}D_\mathsf{H}^2(\mathbb{P}_\varepsilon, \mathbb{Q}))$$

$$\geq (1-\varepsilon)^{-1} \cdot \frac{1}{2}D_\mathsf{H}^2(\mathbb{P}_\varepsilon, \mathbb{Q}) - 2\varepsilon.$$

Finally, we note that by the triangle inequality for Hellinger distance and convexity of squared Hellinger distance,

$$D_\mathsf{H}(\mathbb{P}, \mathbb{Q}) \leq D_\mathsf{H}(\mathbb{P}_\varepsilon, \mathbb{Q}) + D_\mathsf{H}(\mathbb{P}, \mathbb{P}_\varepsilon) \leq D_\mathsf{H}(\mathbb{P}_\varepsilon, \mathbb{Q}) + \varepsilon^{1/2}D_\mathsf{H}(\mathbb{P}, \mathbb{Q}),$$

so that $D_\mathsf{H}^2(\mathbb{P}_\varepsilon, \mathbb{Q}) \geq (1 - \varepsilon^{1/2})^2 D_\mathsf{H}^2(\mathbb{P}, \mathbb{Q})$, and

$$\sup_{g:\mathcal{X}\to\mathbb{R}}\left\{1 - \mathbb{E}_\mathbb{P}\left[e^g\right] \cdot \mathbb{E}_\mathbb{Q}\left[e^{-g}\right]\right\} \geq \frac{(1 - \varepsilon^{1/2})^2}{1 - \varepsilon}\frac{1}{2}D_\mathsf{H}^2(\mathbb{P}, \mathbb{Q}) - 2\varepsilon \geq \frac{1}{2}D_\mathsf{H}^2(\mathbb{P}, \mathbb{Q}) - 4\varepsilon^{1/2},$$

where we have used that $\varepsilon \in (0, 1)$ and $D_\mathsf{H}^2(\mathbb{P}, \mathbb{Q}) \in [0, 2]$.

$\square$

## C.4 Online Learning

**Lemma C.6** (e.g., Cesa-Bianchi and Lugosi [10]). *Let $\Pi$ be a finite set. Consider the exponential weights method with learning rate $\eta > 0$ and initial point $q^{(1)} = \mathrm{unif}(\Pi)$, which has the update:*

$$q^{(t+1)}(\pi) = \frac{\exp(\eta \sum_{i\leq t} f^{(i)}(\pi))}{\sum_{\pi'} \exp(\eta \sum_{i\leq t} f^{(t)}(\pi'))},$$

*for an arbitrary (potentially adaptively selected) sequence of reward vectors $f^{(1)}, \ldots, f^{(T)}$ in $\mathbb{R}^\Pi$. This strategy ensures that with probability 1,*

$$\sum_{t=1}^T \langle q - q^{(t)}, f^{(t)}\rangle \leq \sum_{t=1}^T \langle q^{(t+1)} - q^{(t)}, f^{(t)}\rangle - \frac{1}{\eta}\sum_{t=1}^T D_\mathsf{KL}(q^{(t+1)} \| q^{(t)}) + \frac{D_\mathsf{KL}(q \| q^{(1)})}{\eta},$$

*for all $q \in \Delta(\Pi)$.*

# D  Examples

## D.1  Structured Bandits

In this section we consider adversarial (structured) bandit problems, which correspond to the special case of the adversarial DMSO setting in which there are no observations (i.e., $\mathcal{O} = \{\varnothing\}$). We consider three examples: finite-armed bandits, linear bandits, and convex bandits. For each example, we take $\mathcal{R} = [0, 1]$, fix a *reward function class* $\mathcal{F} \subseteq (\Pi \to [0, 1])$, and take $\mathcal{M}_\mathcal{F} = \{M \mid f^M \in \mathcal{F}\}$ to be the induced model class. Conceptually, $\mathcal{M}_\mathcal{F}$ should be thought of as the set of all reward distributions over $[0, 1]$ with mean rewards in $\mathcal{F}$.

**Example D.1** (Finite-armed bandit). In the finite-armed bandit problem, we take $\Pi = \{1, \ldots, A\}$ as the decision space, where $A \in \mathbb{N}$, then let $\mathcal{F} = [0,1]^A$ and take $\mathcal{M} = \mathcal{M}_{\mathcal{F}}$ as the induced model class. For this setting, whenever $A \geq 2$, it holds that

$$\mathsf{dec}_{\gamma}(\mathrm{co}(\mathcal{M})) \leq \frac{A}{\gamma} \quad \forall \gamma > 0, \quad \text{and} \quad \mathsf{dec}_{\gamma, \varepsilon_{\gamma}}(\mathrm{co}(\mathcal{M})) \geq 2^{-6} \cdot \frac{A}{\gamma} \quad \forall \gamma \geq \frac{A}{3}, \qquad (28)$$

where $\varepsilon_{\gamma} = \frac{A}{12\gamma}$. ◁

This result follows from Foster et al. [18, Proposition 5.2 and 5.3], noting that $\mathrm{co}(\mathcal{M}) = \mathcal{M}$. Plugging (28) into Theorem 2.1 yields a $O(\sqrt{AT \log A})$ upper bound on regret, and plugging into Theorem 2.2 gives a $\widetilde{\Omega}(\sqrt{AT})$ lower bound for sub-Chebychev algorithms.[13]

**Example D.2** (Linear bandit). In the linear bandit problem, we have $\Pi \subseteq \mathbb{R}^d$. We take

$$\mathcal{F} = \{f : \Pi \to [0,1] \mid f \text{ is linear}\},$$

and take $\mathcal{M} = \mathcal{M}_{\mathcal{F}}$ as the induced model class. For this setting, it holds that[14]

$$\mathsf{dec}_{\gamma}(\mathrm{co}(\mathcal{M})) \leq \frac{d}{4\gamma} \quad \forall \gamma > 0, \quad \text{and} \quad \mathsf{dec}_{\gamma, \varepsilon_{\gamma}}(\mathrm{co}(\mathcal{M})) \geq \frac{d}{12\gamma} \quad \forall \gamma \geq \frac{2d}{3}, \qquad (29)$$

where $\varepsilon_{\gamma} := \frac{d}{3\gamma}$. ◁

This result follows from Foster et al. [18, Proposition 6.1 and 6.2], again noting that $\mathrm{co}(\mathcal{M}) = \mathcal{M}$. Plugging (28) into Theorem 2.1 yields a $O(\sqrt{dT \log |\Pi|})$ upper bound on regret, and plugging into Theorem 2.2 gives a $\widetilde{\Omega}(\sqrt{dT})$ lower bound for sub-Chebychev algorithms.

**Example D.3** (Convex bandit). In the convex bandit problem, we have $\Pi \subseteq \mathbb{R}^d$. We take

$$\mathcal{F} = \{f : \Pi \to [0,1] \mid f \text{ is convex}\},$$

and take $\mathcal{M} = \mathcal{M}_{\mathcal{F}}$ as the induced model class. For this setting, it holds that for all $\gamma > 0$,

$$\mathsf{dec}_{\gamma}(\mathrm{co}(\mathcal{M})) \leq O\left(\frac{d^4}{\gamma} \cdot \mathrm{polylog}(d, \mathrm{diam}(\Pi), \gamma)\right). \qquad (30)$$

◁

This result follows from Foster et al. [18, Proposition 6.3] (which itself is a restatement of Lattimore and Szepesvári [36, Theorem 3]), and by noting once more that $\mathrm{co}(\mathcal{M}) = \mathcal{M}$.

**Remark D.1.** The adversarial bandit literature [5, 4, 20, 11, 1, 8, 27, 17, 9, 31, 27, 7] typically considers a slightly different formulation in which the adversary selects a deterministic reward function. This can be captured by restricting $\mathcal{M}$ to deterministic models. It is clear that the upper bounds on $\mathsf{dec}_{\gamma}(\mathrm{co}(\mathcal{M}))$ in the examples above lead to upper bounds for this model. The lower bounds in Examples D.1 and D.2 easily extend as well.

## D.2 Reinforcement Learning

We now consider examples in reinforcement learning. We begin by recalling how to view the episodic reinforcement learning problem under the DMSO framework.

**Model class.** For episodic reinforcement learning, we fix a *horizon $H$* and let the model class $\mathcal{M}$ consist of a set of non-stationary Markov Decision Processes (MDP). Each model $M \in \mathcal{M}$ is specified by

$$M = \left\{\{\mathcal{S}_h\}_{h=1}^{H+1}, \mathcal{A}, \{P_h^M\}_{h=1}^{H}, \{R_h^M\}_{h=1}^{H}, d_1\right\},$$

where $\mathcal{S}_h$ is the state space for layer $h$, $\mathcal{A}$ is the action space, $P_h^M : \mathcal{S}_h \times \mathcal{A} \mapsto \Delta(\mathcal{S}_{h+1})$ is the probability transition kernel for layer $h$, $R_h^M : \mathcal{S}_h \times \mathcal{A} \mapsto \Delta([0,1])$ is the reward distribution for layer $h$ and $d_1 \in \Delta(\mathcal{S}_1)$ is the initial state distribution. This formulation allows reward distribution

---

[13]For this example and Example D.2, the lower bound on $\mathsf{dec}_{\gamma, \varepsilon_{\gamma}}(\mathrm{co}(\mathcal{M}))$ in Foster et al. [18] is witnessed by a subfamily $\mathcal{M}' \subseteq \mathcal{M}$ with $V(\mathcal{M}') = O(1)$. As a result, we can take $C(T) = O(1)$ in Theorem 2.2.

[14]The upper bound here holds for all $\Pi$, while the lower bound holds for a specific choice for $\Pi$.

and transition kernel to vary across models in $\mathcal{M}$, but keeps the initial state distribution is fixed. We adopt the convention that $S_{H+1} = \{s_{H+1}\}$ where $s_{H+1}$ is a deterministic terminal state.

Before an episode, the learner selects a non-stationary policy, $\pi = (\pi_1, \ldots, \pi_H)$ where $\pi_h : \mathcal{S}_h \mapsto \mathcal{A}$; we let $\Pi_{\mathrm{NS}}$ denote the set of all such policies. For a given MDP $M \in \mathcal{M}$, an episode proceeds by first sampling $s_1 \sim d_1$, then for $h = 1, \ldots, H$:

- $a_h = \pi_h(s_h)$.
- $r_h \sim R_h^M(s_h, a_h)$ and $s_{h+1} \sim P_h^M(\cdot \mid s_h, a_h)$.

The value of the policy $\pi$ under $M$ is given by $f^M(\pi) := \mathbb{E}^{M,\pi}[\sum_{h=1}^{H} r_h]$, where $\mathbb{E}^{M,\pi}[\cdot]$ denotes expectation under the process above.

**Adversarial protocol.** Within the adversarial DMSO framework, model classes above lead to the following adversarial reinforcement learning protocol. At each time $t$, the learner plays selects a policy $\pi \in \Pi_{\mathrm{NS}}$ and the adversary chooses an MDP $M^{(t)} \in \mathcal{M}$. The policy $\pi^{(t)}$ is then executed in the MDP $M^{(t)}$, resulting in a trajectory $\tau^{(t)} = (s_1^{(t)}, r_1^{(t)}, r_1^{(t)}), \ldots, (s_H^{(t)}, r_H^{(t)}, r_H^{(t)})$. The learner then observes feedback $(r^{(t)}, o^{(t)})$, where $r^{(t)} := \sum_{h=1}^{H} r_H^{(t)}$ is the cumulative reward of the episode, and $o^{(t)} = \tau^{(t)}$ is the trajectory.

With this setting in mind, we give our main example.

**Example D.4** (Tabular MDP). Let $\mathcal{M}$ be the class of finite-state/action (tabular) MDPs with horizon $H$, $S \geq 2$ states, $A \geq 2$ actions, and $\sum_{h=1}^{H} r_h \in [0, 1]$. Then, for any $\gamma \geq A^{\min\{S-1, H\}}/6$,

$$\mathsf{dec}_{\gamma, \varepsilon_\gamma}(\mathrm{co}(\mathcal{M})) \geq \frac{A^{\min\{S-1, H\}}}{24\gamma},$$

where $\varepsilon_\gamma := A^{\min\{S-1, H\}}/24\gamma$. ◁

Using this result with Theorem 2.2 leads to a lower bound on regret that scales with $\Omega(A^{\min S-1, H})$, which recovers existing intractability results for this setting [30, 41]. Note that we have $\mathsf{dec}_\gamma(\mathcal{M}) = \mathrm{poly}(S, A, H)/\gamma$ for this setting [18], so this is a case where there is a separation between the stochastic and adversarial setting.

We briefly mention that the set $\mathrm{co}(\mathcal{M})$ can be interpreted as the set of *latent MDPs* [30]. In the latent MDP setting, each model is a mixture of MDPs. At the beginning of each episode, the underlying MDP from the mixture (the identity is not observed), and then run the MDP for the duration of the episode. This setting is also known to be intractable.

## D.3 Proofs for Examples

### D.3.1 Preliminaries

Our lower bounds on the Decision-Estimation Coefficient involve a constructing hard sub-family of models. Recall the following definition from [18].

**Definition D.1** $((\alpha, \beta, \delta)$-family). *A reference model $\overline{M} \in \mathcal{M}$ and collection $\{M_1, \ldots, M_N\}$ with $N \geq 2$ are said to be an $(\alpha, \beta, \delta)$-family if the following properties hold:*

1. *Regret property. There exist functions $u^M : \Pi \mapsto [0, 1]$, with $\sum_{M \in \mathcal{M}} u^M(\pi) \leq \frac{N}{2}$ for all $\pi$ such that*

$$f^M(\pi_M) - f^M(\pi) \geq \alpha \cdot (1 - u^M(\pi))$$

*for all $M \in \mathcal{M}$.*

2. *Information property. There exist functions $v^M : \Pi \mapsto [0, 1]$, with $\sum_{M \in \mathcal{M}} v^M(\pi) \leq 1$ for all $\pi$, such that*

$$D_{\mathsf{H}}^2\big(M(\pi), \overline{M}(\pi)\big) \leq \beta \cdot v^M(\pi) + \delta.$$

Any $(\alpha, \beta, \delta)$-family leads to a difficult decision making problem because a given decision can have low regret or large information gain on (roughly) one model in the family. This is formalized through the following lemma.

**Lemma D.1** (Lemma 5.1, [18])**.** *Let $\mathcal{M} = \{M_1, \ldots, M_N\}$ be an $(\alpha, \beta, \delta)$-family with respect to $\overline{M}$.* *Then, for all $\gamma \geq 0$,*

$$\mathsf{dec}_\gamma(\mathcal{M}, \overline{M}) \geq \frac{\alpha}{2} - \gamma\left(\frac{\beta}{N} + \delta\right).$$

The following technical lemma bounds Hellinger distance for Bernoulli distributions.

**Lemma D.2** (Lemma A.7, [18])**.** *For any $\Delta \in (0, 1/2)$,*

$$D_{\mathsf{H}}^2\left(\mathrm{Ber}\left(\frac{1}{2} + \Delta\right), \mathrm{Ber}\left(\frac{1}{2}\right)\right) \leq 3\Delta^2.$$

### D.3.2 Proof for Example D.4 (Tabular MDP)

In this section, we prove the lower bound in Example D.4. We first derive an intermediate result which gives a lower bound on the Decision-Estimation Coefficient when the model class $\mathcal{M}$ consists of *mixtures of $K$ MDPs*; this is equivalent to the subset of $\mathrm{co}(\mathcal{M})$ where we restrict to support size $K$, as well as the so-called latent MDP setting [30].

**Lemma D.3.** *Let $K \geq 1$ be given. Let $\mathcal{M}$ be the class of* mixtures of $K$ MDPs *with horizon $H$, $S \geq 2$ states, $A \geq 2$ actions, and $\sum_{h=1}^{H} r_h \in [0, 1]$. Then there exists $\overline{M} \in \mathcal{M}$ such that for all $\gamma \geq A^{\min\{S-1,H,K\}}/6$,*

$$\mathsf{dec}_\gamma(\mathcal{M}_{\varepsilon_\gamma}(\overline{M}), \overline{M}) \geq \frac{A^{\min\{S-1,H,K\}}}{24\gamma},$$

*where $\varepsilon_\gamma := \frac{A^{\min\{S-1,H,K\}}}{24\gamma}$.*

The proof of this result proceeds by constructing a hard sub-family of models and appealing to Lemma D.1. Our construction is based of the lower bound for latent MDPs in Kwon et al. [30].

**Proof of Lemma D.3.** Let $\mathcal{S}$ and $\mathcal{A}$ be arbitrary sets with $|\mathcal{S}| = S$ and $|\mathcal{A}| = A$. Let $\Delta \in (0, 1/2)$ be a parameter to be chosen later, and define $\overline{K} := \min\{S - 1, K, H\}$. Partition the state space $\mathcal{S}$ into sets $\mathcal{S}'$ and $\mathcal{S} \setminus \mathcal{S}'$ such that $|\mathcal{S}'| = \overline{K} + 1$, and label the states in $\mathcal{S}'$ as $\{s^{(1)}, \ldots, s^{(\overline{K}+1)}\}$. Additionally, define sets via $\mathcal{S}_h = \{s^{(h)}, s^{(\overline{K}+1)}\}$ for $h \leq \overline{K}$ and $\mathcal{S}_h = \{s^{(\overline{K}+1)}\} \cup (S \setminus S')$ for $\overline{K} < h \leq H + 1$. Recall that the decision space $\Pi_{\mathrm{NS}}$ is the set of all deterministic non-stationary policies $\pi = (\pi_1, \ldots, \pi_H)$ where $\pi_h : \mathcal{S}_h \mapsto \mathcal{A}$.

We construct a class $\mathcal{M}' \subseteq \mathcal{M}$ in which each model $M \in \mathcal{M}'$ is specified by

$$M = \left\{\{\mathcal{S}_h\}_{h=1}^{H+1}, \mathcal{A}, \{\mathbb{M}_k^M\}_{k=1}^{\overline{K}}, \{a_k^M\}_{k=1}^{K}\right\},$$

where for each $k \in [\overline{K}]$, $a_k^M \in \mathcal{A}$, and where $\mathbb{M}_k^M$ is a tabular MDP specified by

$$\mathbb{M}_k^M = \left\{\{\mathcal{S}_h\}_{h=1}^{H+1}, \mathcal{A}, \{P_{h,k}^M\}_{h=1}^{H}, \{R_{h,k}^M\}_{h=1}^{H}, \delta_{s^{(1)}}\right\}.$$

Here, $d_1 = \delta_{s^{(1)}}$, so that the initial state $s_1$ is $s^{(1)}$ deterministically. The transitions $P_{h,k}^M$ and rewards $R_{h,k}^M$ are constructed as follows.

- Construction of $\mathbb{M}_1^M$.

  (i) For all $h \leq H$, the dynamics $P_{h,k}^M$ are deterministic. For an action $a_h$ in the state $s_h$, the next state $s_{h+1}$ is

  $$s_{h+1} = \begin{cases} s^{(h+1)}, & \text{if } h \leq \overline{K}, s_h = s^{(h)}, \text{ and } a_h = a_i^M, \\ s^{(\overline{K}+1)}, & \text{if } h \leq \overline{K}, s_h = s^{(h)}, \text{ and } a_h \neq a_i^M, \\ s_h, & \text{otherwise.} \end{cases}$$

  (ii) The reward distribution is given by

  $$R_{h,k}^M(s_h, a_h) = \begin{cases} \mathrm{Ber}\left(\frac{1}{2} + \Delta\right), & \text{if } h = \overline{K}, s_h = s^{(\overline{K})}, \text{ and } a_h = a_{\overline{K}}^M, \\ \mathrm{Ber}\left(\frac{1}{2}\right), & \text{if } h = \overline{K}, s_h = s^{(\overline{K})}, \text{ and } a_h \neq a_{\overline{K}}^M, \\ 0, & \text{otherwise.} \end{cases}$$

- Construction of $\mathbb{M}_j^M$ for $2 \leq j \leq \overline{K}$.

  (i) For each $h \leq H$, the dynamics $P_{h,k}^M$ are deterministic. For action $a_h$ in state $s_h$, the next state $s_{h+1}$ is

$$
s_{h+1} = \begin{cases}
s^{(h+1)} & \text{if} \quad s_h = s^{(h)} \text{ and } h < j \\
s^{(\overline{K}+1)} & \text{if} \quad s_h = s^{(h)}, h = j \text{ and } a_h = a_h^M \\
s^{(h+1)} & \text{if} \quad s_h = s^{(h)}, h = j \text{ and } a_h \neq a_h^M \\
s^{(h+1)} & \text{if} \quad s_h = s^{(h)}, h > j \text{ and } a_h = a_h^M \\
s^{(\overline{K}+1)} & \text{if} \quad s_h = s^{(h)}, h > j \text{ and } a_h \neq a_h^M \\
s^{(\overline{K}+1)} & \text{if} \quad h = \overline{K} - 1 \text{ or } h = \overline{K} \\
s_h & \text{otherwise}
\end{cases}
$$

  (ii) The reward distribution is given by

$$
R_{h,k}^M(s_h, a_h) = \begin{cases}
\mathrm{Ber}\left(\frac{1}{2}\right), & \text{if} \quad h = \overline{K}, \\
0, & \text{otherwise.}
\end{cases}
$$

Each model $M \in \mathcal{M}'$ is a uniform mixture of $\overline{K}$ MDPs $\{\mathbb{M}_1^M, \ldots, \mathbb{M}_{\overline{K}}^M\}$ as described above, parameterized by the action sequence $a_{1:\overline{K}}^M$. The model class $\mathcal{M}'$ is defined as the set of all such mixture models (one for each sequence in $\mathcal{A}^{\overline{K}}$, so that $|\mathcal{M}'| = A^{\overline{K}}$.

At the start of each episode, an MDP $\mathbb{M}_z^M$ is chosen by sampling $z \sim \mathrm{Unif}([\overline{K}])$. The trajectory is then drawn by setting $s_1 = s^{(1)}$, and for $h = 1, \ldots, H$:

- $a_h = \pi_h(s_h)$.
- $r_h \sim R_{h,z}^M(s_h, a_h)$ and $s_{h+1} \sim P_{h,z}^M(\cdot \mid s_h, a_h)$.

Note that rewards can be non-zero only at layer $h = \overline{K}$. We receive a reward from $\mathrm{Ber}\left(\frac{1}{2} + \Delta\right)$ only when $z = 1$ and the first $\overline{K}$ actions match $a_{1:\overline{K}}^M$, i.e. $a_{1:\overline{K}} = a_{1:\overline{K}}^M$. For every other action sequence, the reward is sampled from $\mathrm{Ber}\left(\frac{1}{2}\right)$. Thus, for any policy $\pi$,

$$
f^M(\pi) = \tfrac{1}{2} + \Delta \mathbb{I}\{\pi(s_{1:\overline{K}}) = a_{1:\overline{K}}^M\},
$$

which implies that

$$
f^M(\pi_M) - f^M(\pi) = \Delta(1 - \mathbb{I}\{\pi(s_{1:\overline{K}}) = a_{1:\overline{K}}^M\}). \tag{31}
$$

Finally, we define the reference model $\overline{M}$. The model $\overline{M}$ is specified by $\{\{\mathcal{S}_h\}_{h=1}^{H+1}, \mathcal{A}, \mathbb{M}^{\overline{M}}\}$ where $\mathbb{M}^{\overline{M}}$ is a tabular MDP given by

$$
\mathbb{M}^{\overline{M}} = \{\{\mathcal{S}_h\}_{h=1}^{H+1}, \mathcal{A}, P_h^{\overline{M}}, R_h^{\overline{M}}, \delta_{s^{(1)}}\}.
$$

Here, the initial state $s_1$ is $s^{(1)}$ deterministically, and the transitions $P_{h,k}^{\overline{M}}$ and rewards $R_{h,k}^{\overline{M}}$ are as follows:

  (i) Transitions are stochastic and independent of the chosen action. In particular, for each $h \leq H$, the dynamics $P_h^{\overline{M}}$ are given by

$$
P_h^{\overline{M}}(s_{h+1} \mid s_h, a_h) = \begin{cases}
\frac{\overline{K}-h}{\overline{K}-h+1} & \text{if} \quad h \leq \overline{K}, s_h = s^{(h)} \text{ and } s_{h+1} = s^{(h+1)} \\
\frac{1}{\overline{K}-h+1} & \text{if} \quad h \leq \overline{K}, s_h = s^{(h)} \text{ and } s_{h+1} = s^{(\overline{K}+1)} \\
1 & \text{if} \quad h \leq \overline{K}, s_h \neq s^{(h)} \text{ and } s_h = s_{h+1} \\
1 & \text{if} \quad h > \overline{K} \text{ and } s_h = s_{h+1} \\
0 & \text{otherwise}
\end{cases}
$$

  (ii) The reward distribution is given by

$$
R_h^{\overline{M}}(s_h, a_h) = \begin{cases}
\mathrm{Ber}\left(\frac{1}{2}\right), & \text{if} \quad h = \overline{K}, \\
0, & \text{otherwise.}
\end{cases}
$$

Note that $\overline{M}$ can be thought of as a mixture of $\overline{K}$ identical tabular MDPs each given by $\mathbb{M}^{\overline{M}}$. Note that for any policy $\pi$, the rewards for any trajectory in $\overline{M}$ are sampled from $\mathrm{Ber}\left(\frac{1}{2}\right)$, and thus $f^{\overline{M}}(\pi) = \frac{1}{2}$ which implies that

$$f^{\overline{M}}(\pi_{\overline{M}}) - f^{\overline{M}}(\pi) = 0. \tag{32}$$

We define $\mathcal{M}'' = \mathcal{M}' \cup \{\overline{M}\} \subseteq \mathcal{M}$, and note that for any policy $\pi$, the distribution over the trajectories is identical in all mixture models in $\mathcal{M}''$. However, as mentioned before, the rewards in $\overline{M}$ are sampled from $\mathrm{Ber}\left(\frac{1}{2}\right)$ and for any $M \in \mathcal{M}'$, the rewards in $M$ are sampled from $\mathrm{Ber}\left(\frac{1}{2} + \frac{\Delta}{M}\mathbb{I}\left\{\pi(s_{1:\overline{K}}) = a_{1:\overline{K}}^M\right\}\right)$. Thus, for any policy $\pi$ and $M \in \mathcal{M}'$,

$$D_{\mathsf{H}}^2\big(M(\pi), \overline{M}(\pi)\big) = D_{\mathsf{H}}^2\Big(\mathrm{Ber}\Big(\tfrac{1}{2} + \tfrac{\Delta}{\overline{K}}\mathbb{I}\big\{\pi(s_{1:\overline{K}}) = a_{1:\overline{K}}^M\big\}\Big), \mathrm{Ber}\Big(\tfrac{1}{2}\Big)\Big)$$
$$\leq 3\frac{\Delta^2}{\overline{K}^2} \cdot \mathbb{I}\{\pi(s_{1:\overline{K}}) = a_{1:\overline{K}}^M\}, \tag{33}$$

where the last line uses Lemma D.2.

The bounds in (31), (32) and (33) together imply that the model class $\mathcal{M}''$ is a $\big(\frac{\Delta}{\overline{K}}, 3\frac{\Delta^2}{\overline{K}^2}, 0\big)$-family in the sense of Definition D.1, where for each $\pi \in \Pi$ and $M \in \mathcal{M}''$ we take

$$u^M(\pi) := \mathbb{I}\{\pi(s_{1:\overline{K}}) = a_{1:\overline{K}}^M\} \qquad \text{and} \qquad v^M(\pi) := \mathbb{I}\{\pi(s_{1:\overline{K}}) = a_{1:\overline{K}}^M\},$$

with $u^{\overline{M}}(\pi) := 1$ and $v^{\overline{M}}(\pi) := 0$. As a result, Lemma D.1 implies that

$$\mathsf{dec}_\gamma(\mathcal{M}, \overline{M}) \geq \frac{\Delta}{2\overline{K}} - \frac{3\gamma\Delta^2}{\overline{K}^2 N},$$

for $N := A^{\overline{K}} + 1$. Setting $\Delta = \frac{\overline{K}N}{12\gamma}$ leads to the lower bound $\mathsf{dec}_\gamma(\mathcal{M}, \overline{M}) \geq \frac{N}{24\gamma}$. We conclude by noting that all $M \in \mathcal{M}''$ have $M \in \mathcal{M}_{\varepsilon_\gamma}(\overline{M})$ with $\varepsilon_\gamma = \frac{N}{24\gamma}$, and thus the lower bound on the DEC also applies to the class $\mathcal{M}_{\varepsilon_\gamma}(\overline{M})$. $\qquad\square$

**Proof for Example D.4.** let $\mathcal{M}$ be the class of all tabular MDPs, and let $\mathcal{M}^{(K)}$ denote the set of all mixture models in which each $M \in \mathcal{M}^{(K)}$ is a mixture of $K$ MDPs from $\mathcal{M}$. Additionally, define $\widetilde{\mathcal{M}} = \mathrm{co}(\mathcal{M})$, and note that $\mathcal{M}^{(K)} \subseteq \widetilde{\mathcal{M}}$ for all $K \geq 1$. For any $\varepsilon > 0$ and $\overline{M} \in \mathcal{M}^{(K)}$, we have that $\mathcal{M}_\varepsilon^{(k)}(\overline{M}) \subseteq \widetilde{\mathcal{M}}_\varepsilon(\overline{M})$, which implies that

$$\mathsf{dec}_\gamma(\widetilde{\mathcal{M}}_\varepsilon(\overline{M}), \overline{M}) \geq \mathsf{dec}_\gamma(\mathcal{M}_\varepsilon^{(K)}(\overline{M}), \overline{M}),$$

because $\mathsf{dec}_\gamma(\cdot, \overline{M})$ is a non-decreasing function with respect to inclusion. Using Lemma D.3, we have that for any $K \geq 1$ and $\gamma \geq A^{\min\{S-1,H,K\}}/6$, with $\varepsilon_\gamma := A^{\min\{S-1,H,K\}}/24\gamma$,

$$\mathsf{dec}_\gamma(\widetilde{\mathcal{M}}_\varepsilon(\overline{M}), \overline{M}) \geq \mathsf{dec}_\gamma(\mathcal{M}_\varepsilon^{(K)}(\overline{M}), \overline{M}) \geq \frac{A^{\min\{S-1,H,K\}}}{24\gamma}.$$

Setting $K = S$ above gives the desired lower bound. $\qquad\square$

# E  Structural Results

This section is organized as follows.

- In Appendix E.1, we recall existing variants of the information ratio and state some basic properties.
- In Appendix E.2, we prove equivalence of the Decision-Estimation Coefficient and the parameterized information ratio with Hellinger distance (Theorem 3.1), as well as a generalization of this result (Theorem E.1).
- In Appendix E.3, we prove equivalence of the parameterized information ratio with Hellinger distance and the high-probability exploration-by-optimization objective.

## E.1 Background on Complexity Measures

For a measurable space $(\mathcal{X}, \mathscr{F})$, let us call any function $D : \Delta(\mathcal{X}) \times \Delta(\mathcal{X}) \to \mathbb{R}_+$ a *divergence-like* function.

**Generalized information ratio.** Below we recall two notions of *generalized information ratio* introduced by Lattimore and György [34] and Lattimore [33], which extend the original definition of Russo and Van Roy [51, 52].

For a given prior $\mu \in \Delta(\mathcal{M} \times \Pi)$, define $\mu_{\mathrm{pr}}(\pi') := \mathbb{P}(\pi^\star = \pi')$ and $\mu_{\mathrm{po}}(\pi'; \pi, z) := \mathbb{P}(\pi^\star = \pi' \mid (\pi, z))$ under the process $(M, \pi^\star) \sim \mu, \pi \sim p, z \sim M(\pi)$.

1. Lattimore and György [34] define a class $\mathcal{M}$ to have generalized information ratio $(\alpha, \beta, \lambda)$ (where $\alpha, \beta \geq 0$, $\lambda > 1$) if for each prior $\mu \in \Delta(\mathcal{M} \times \Pi)$, there exists a distribution $p \in \Delta(\Pi)$ such that

$$\mathbb{E}_{(M,\pi^\star)\sim\mu} \mathbb{E}_{\pi\sim p}[f^M(\pi^\star) - f^M(\pi)] \leq \alpha + \beta^{1-1/\lambda} \big( \mathbb{E}_{\pi\sim p} \mathbb{E}_{z|\pi}[D(\mu_{\mathrm{po}}(\cdot; \pi, z) \| \mu_{\mathrm{pr}})] \big)^{1/\lambda}. \tag{34}$$

2. Lattimore [33] define the generalized information ratio for a class $\mathcal{M}$ (for $\lambda > 1$) via

$$\Psi_\lambda(\mathcal{M}) = \sup_{\mu \in \Delta(\mathcal{M} \times \Pi)} \inf_{p \in \Delta(\Pi)} \left\{ \frac{(\mathbb{E}_{(M,\pi^\star)\sim\mu} \mathbb{E}_{\pi\sim p}[f^M(\pi^\star) - f^M(\pi)])^\lambda}{\mathbb{E}_{\pi\sim p} \mathbb{E}_{z|\pi}[D(\mu_{\mathrm{po}}(\cdot; \pi, z) \| \mu_{\mathrm{pr}})]} \right\}. \tag{35}$$

As mentioned in Section 3, the formulations above slightly generalize the original versions in Lattimore and György [34], Lattimore [33] by incorporating models $M \in \mathcal{M}$ and considering general distances.

The following proposition shows that boundedness of the generalized information ratio implies boundedness of the parameterized information ratio (Definition 3.1).

**Proposition E.1.** *Fix $\alpha, \beta \geq 0$ and $\lambda > 1$. If a class $\mathcal{M}$ has generalized information ratio $(\alpha, \beta, \lambda)$ in the sense of (34), then*

$$\inf_\gamma^D(\mathcal{M}) \leq \alpha + \frac{\beta}{\gamma^{\frac{1}{\lambda-1}}} \quad \forall \gamma > 0.$$

*Likewise, the generalized information ratio in (35) satisfies*

$$\inf_\gamma^D(\mathcal{M}) \leq (\Psi_\lambda(\mathcal{M})/\gamma)^{\frac{1}{\lambda-1}} \quad \forall \gamma > 0.$$

**Proof of Proposition E.1.** Suppose $\mathcal{M}$ has generalized information ratio $(\alpha, \beta, \lambda)$. Then there exists $p \in \Delta(\Pi)$ such that for all $\mu \in \Delta(\mathcal{M} \times \Pi)$, we have

$$\mathbb{E}_{(M,\pi^\star)\sim\mu} \mathbb{E}_{\pi\sim p}[f^M(\pi^\star) - f^M(\pi)] \leq \alpha + \beta^{1-1/\lambda} \big( \mathbb{E}_{\pi\sim p} \mathbb{E}_{z|\pi}[D(\mu_{\mathrm{po}}(\cdot; \pi, z) \| \mu_{\mathrm{pr}})] \big)^{1/\lambda}$$

$$\leq \alpha + \frac{\beta}{\gamma^{\frac{1}{\lambda-1}}} + \gamma \cdot \mathbb{E}_{\pi\sim p} \mathbb{E}_{z|\pi}[D(\mu_{\mathrm{po}}(\cdot; \pi, z) \| \mu_{\mathrm{pr}})],$$

where we have applied Young's inequality, which gives that $xy \leq \frac{\lambda-1}{\lambda} x^{\frac{\lambda}{\lambda-1}} + \frac{1}{\lambda} y^\lambda$ for $x, y \geq 0$.

For the second result, we use that the definition of $\Psi_\lambda(\mathcal{M})$ implies generalized information ratio $(0, (\Psi_\lambda(\mathcal{M}))^{\frac{1}{\lambda-1}}, \lambda)$. $\qquad\square$

This results show that an upper bound in terms of the parameterized information ratio in Definition 3.1 implies an upper bound in terms of either version of the generalized information ratio. It is also straightforward to see that generalized information ratio $(0, \beta, \lambda)$ in (34) implies that $\Psi_\lambda(\mathcal{M}) \leq \beta^{\lambda-1}$ and vice-versa. Note that $\alpha = 0$ is the most interesting regime, as the regret bounds in Lattimore and György [34] scale with $\alpha \cdot T$ when $\alpha > 0$.

Another important property of the parameterized information ratio (as well both generalized information ratios) is that it is invariant under convexification.

**Proposition 3.1.** *For any divergence-like function $D(\cdot \| \cdot) : \Delta(\Pi) \times \Delta(\Pi) \to \mathbb{R}_+$, we have*

$$\inf_\gamma^D(\mathcal{M}) = \inf_\gamma^D(\mathrm{co}(\mathcal{M})), \quad \forall \gamma > 0.$$

**Proof of Proposition 3.1.** Fix $\mu \in \Delta(\mathrm{co}(\mathcal{M}) \times \Pi)$. We can represent any $\overline{M} \in \mathrm{co}(\mathcal{M})$ as a mixture $\nu \in \Delta(\mathcal{M})$, so that $\overline{M} = \mathbb{E}_{M \sim \nu}[M]$. Let $\widetilde{\mu} \in \Delta(\Delta(\mathcal{M}) \times \Pi)$ be such that the process $(\nu, \pi^\star) \sim \widetilde{\mu}$, $\overline{M} = \mathbb{E}_{M \sim \nu}[M]$ has the same law as $(\overline{M}, \pi^\star) \sim \widetilde{\mu}$. Finally, let $\mu' \in \Delta(\mathcal{M} \times \Pi)$ be the law of $(M, \pi^\star)$ induced by sampling $(\nu, \pi^\star) \sim \widetilde{\mu}$ and $M \sim \nu$.

We observe that for any distribution $p \in \Delta(\Pi)$,

$$
\begin{aligned}
&\mathbb{E}_{(\overline{M}, \pi^\star) \sim \mu} \, \mathbb{E}_{\pi \sim p} \big[ f^{\overline{M}}(\pi^\star) - f^{\overline{M}}(\pi) \big] \\
&= \mathbb{E}_{(\nu, \pi^\star) \sim \widetilde{\mu}} \, \mathbb{E}_{\pi \sim p} \, \mathbb{E}_{M \sim \nu} [ f^M(\pi^\star) - f^M(\pi) ] \\
&= \mathbb{E}_{(M, \pi^\star) \sim \mu'} \, \mathbb{E}_{\pi \sim p} [ f^M(\pi^\star) - f^M(\pi) ].
\end{aligned}
$$

Next, observe that $(\pi, \pi^\star, z)$ are identically distributed under the processes $\pi \sim p$, $(\overline{M}, \pi^\star) \sim \mu$, $z \sim \overline{M}(\pi)$ and $\pi \sim p$, $(M, \pi^\star) \sim \mu'$, $z \sim M(\pi)$. As a result, we have $\mu_{\mathrm{pr}} = \mu'_{\mathrm{pr}}$ and $\mu_{\mathrm{po}} = \mu'_{\mathrm{po}}$, so

$$
\mathbb{E}_{\pi \sim p} \, \mathbb{E}_{z|\pi} [ D(\mu_{\mathrm{po}}(\cdot; \pi, z) \,\|\, \mu_{\mathrm{pr}}) ] = \mathbb{E}_{\pi \sim p} \, \mathbb{E}_{z|\pi} \big[ D\big( \mu'_{\mathrm{po}}(\cdot; \pi, z) \,\|\, \mu'_{\mathrm{pr}} \big) \big].
$$

This establishes that $\inf_\gamma^D(\mathrm{co}(\mathcal{M})) \le \inf_\gamma^D(\mathcal{M})$; the other direction is trivial. $\qquad\square$

### E.2  Decision-Estimation Coefficient and Information Ratio (Theorem 3.1)

**Theorem 3.1.** *For all $\gamma > 0$, $\inf_\gamma^{\mathsf{H}}(\mathcal{M}) \le \mathsf{dec}_\gamma(\mathrm{co}(\mathcal{M})) \le \inf_{\gamma/4}^{\mathsf{H}}(\mathcal{M})$.*

Theorem 3.1 is a special case of the following theorem, which concerns general divergence-like functions.

**Theorem E.1.** *Let $\Delta(\Pi) \times \Delta(\Pi) \to \mathbb{R}_+$ be any divergence-like function for which there exist constants $c_1, c_2 \ge 1$ such that:*

    *1. For all $\mathbb{Q} \in \Delta(\Pi)$, $\mathbb{P} \mapsto D(\mathbb{P} \,\|\, \mathbb{Q})$ is convex.*

    *2. For all pairs of random variables $(X, Y)$,*

$$
\mathbb{E}_{X \sim \mathbb{P}_X} \big[ D\big( \mathbb{P}_{Y|X} \,\|\, \mathbb{P}_Y \big) \big] \le c_1 \cdot \mathbb{E}_{Y \sim \mathbb{P}_Y} \big[ D\big( \mathbb{P}_{X|Y} \,\|\, \mathbb{P}_X \big) \big]
$$

    *3. For all pairs of random variables $(X, Y)$,*

$$
\mathbb{E}_{X \sim \mathbb{P}_X} \big[ D\big( \mathbb{P}_{Y|X} \,\|\, \mathbb{P}_Y \big) \big] \le c_2 \cdot \inf_{\mathbb{Q}} \mathbb{E}_{X \sim \mathbb{P}_X} \big[ D\big( \mathbb{P}_{Y|X} \,\|\, \mathbb{Q} \big) \big].
$$

    *4. For all $\varepsilon > 0$ sufficiently small, and all $\mathbb{Q} \in \Delta(\Pi)$, there exists $\mathbb{Q}' \in \Delta(\Pi)$ such that $D(\mathbb{P} \,\|\, \mathbb{Q}) \ge D(\mathbb{P} \,\|\, \mathbb{Q}') - \varepsilon$ and $\sup_{\mathbb{P} \in \Delta(\Pi)} D(\mathbb{P} \,\|\, \mathbb{Q}') < \infty$.*

*Then we have*

$$
\inf_{c_1 \gamma}^D(\mathcal{M}) \le \mathsf{dec}_\gamma^D(\mathrm{co}(\mathcal{M})) \le \inf_{(c_1 c_2)^{-1} \gamma}^D(\mathcal{M}). \tag{36}
$$

All $f$-divergences satisfy Property 2 with $c_1 = 1$, but may not satisfy Property 3. On the other hand, Bregman divergences[15] satisfy Property 3 with $c_2 = 1$, but may not satisfy Property 2 (consider squared euclidean distance). KL-divergence, being both an $f$-divergence and a Bregman divergence, satisfies both properties with $c_1 = c_2 = 1$ (this fact has been used tacitly in many prior works). Squared Hellinger distance is an $f$-divergence but not a Bregman divergence, yet satisfies Property 3 with $c_2 = 4$ as a consequence of the triangle inequality.

**Proof of Theorem E.1.** We first bound the DEC by the information ratio, then proceed to bound the information ratio by the DEC.

**Bounding the DEC by the information ratio.** Fix $M' \in \mathcal{M}$, and $\varepsilon > 0$ and let $M''$ be such that $D_{\mathsf{H}}^2(\cdot, M'(\pi)) \ge D_{\mathsf{H}}^2(\cdot, M''(\pi)) - \varepsilon$ and $D_{\mathsf{H}}^2(\cdot, M''(\pi)) < \infty$ (as guaranteed by Property 4). Using

---

[15]Recall that for a convex set $\mathcal{X}$ and regularizer $\mathcal{R} : \mathcal{X} \to \mathbb{R}$, $D_{\mathcal{R}}(x \,\|\, y) := \mathcal{R}(x) - \mathcal{R}(y) - \langle \nabla \mathcal{R}(y), x - y \rangle$ is the associated Bregman divergence.

the minimax theorem (Lemma C.2), we have

$$\mathsf{dec}_\gamma^D(\mathcal{M}, M') \leq \inf_{p \in \Delta(\Pi)} \sup_{M \in \mathcal{M}} \mathbb{E}_{\pi \sim p}[f^M(\pi_M) - f^M(\pi) - \gamma \cdot D(M(\pi) \| M''(\pi))] + \gamma\varepsilon$$

$$= \inf_{p \in \Delta(\Pi)} \sup_{\nu \in \Delta(\mathcal{M})} \mathbb{E}_{\pi \sim p} \mathbb{E}_{M \sim \nu}[f^M(\pi_M) - f^M(\pi) - \gamma \cdot D(M(\pi) \| M''(\pi))] + \gamma\varepsilon$$

$$= \sup_{\nu \in \Delta(\mathcal{M})} \inf_{p \in \Delta(\Pi)} \mathbb{E}_{\pi \sim p} \mathbb{E}_{M \sim \nu}[f^M(\pi_M) - f^M(\pi) - \gamma \cdot D(M(\pi) \| M''(\pi))] + \gamma\varepsilon.$$

Note that the application of the minimax theorem is admissible here, since $\Delta(\Pi)$ is compact (a consequence of finiteness of $\Pi$) and the objective value is bounded (a consequence of the choice of $M''$ and the fact that $f^M \in [0, 1]$).

Fix $\nu \in \Delta(\mathcal{M})$, and let $\mu \in \Delta(\mathcal{M} \times \Pi)$ be the induced law of $(M, \pi_M)$. Let $\overline{M}_{\pi'}(\pi) = \mathbb{E}_{M \sim \nu}[M(\pi) \mid \pi_M = \pi']$ and $\overline{M}(\pi) = \mathbb{E}_{M \sim \mu}[M(\pi)] = \mathbb{E}_{\pi^\star \sim \mu}[M_{\pi^\star}(\pi)]$. Then for any $p \in \Delta(\Pi)$, we have

$$\mathbb{E}_{M \sim \nu} \mathbb{E}_{\pi \sim p}[f^M(\pi_M) - f^M(\pi) - \gamma \cdot D(M(\pi) \| M''(\pi))]$$

$$= \mathbb{E}_{(M, \pi^\star) \sim \mu} \mathbb{E}_{\pi \sim p}[f^M(\pi^\star) - f^M(\pi) - \gamma \cdot D(M(\pi) \| M''(\pi))]$$

$$\leq \mathbb{E}_{(M, \pi^\star) \sim \mu} \mathbb{E}_{\pi \sim p}\big[f^M(\pi^\star) - f^M(\pi) - \gamma \cdot D\big(\overline{M}_{\pi^\star}(\pi) \| M'; (\pi)\big)\big]$$

$$\leq \mathbb{E}_{(M, \pi^\star) \sim \mu} \mathbb{E}_{\pi \sim p}\big[f^M(\pi^\star) - f^M(\pi) - \gamma c_2^{-1} \cdot D\big(\overline{M}_{\pi^\star}(\pi) \| \overline{M}(\pi)\big)\big],$$

where the first inequality uses convexity of $\mathbb{P} \mapsto D(\mathbb{P} \| \mathbb{Q})$ (Property 1), and the second inequality uses Property 3. To proceed, let $\mathbb{P}$ be the law of the process $\pi \sim p$, $(M, \pi^\star) \sim \mu$, $z \sim M(\pi)$. Observe that $\overline{M}_{\pi^\star}(\pi) = \mathbb{P}_{z|\pi,\pi^\star}$ and $\overline{M}(\pi) = \mathbb{P}_{z|\pi}$. Hence, using Property 2, we have that for all $\pi$,

$$\mathbb{E}_{\pi^\star \sim \nu}\big[D\big(\overline{M}_{\pi^\star}(\pi) \| \overline{M}(\pi)\big)\big] \geq c_1^{-1} \mathbb{E}_{z|\pi}\big[D\big(\mathbb{P}_{\pi^\star|\pi,z} \| \mathbb{P}_{\pi^\star|\pi}\big)\big] = c_1^{-1} \mathbb{E}_{z|\pi}\big[D\big(\mathbb{P}_{\pi^\star|\pi,z} \| \mathbb{P}_{\pi^\star}\big)\big],$$

where the last equality uses that $\pi$ and $\pi^\star$ are independent (marginally). Since $D\big(\mathbb{P}_{\pi^\star|\pi,z} \| \mathbb{P}_{\pi^\star}\big) = D(\mu_{\mathrm{po}}(\cdot; \pi, z) \| \mu_{\mathrm{pr}})$, if we choose $p$ to attain the minimum in (19) for $\mu$ we are guaranteed that

$$\mathbb{E}_{M \sim \nu} \mathbb{E}_{\pi \sim p}[f^M(\pi_M) - f^M(\pi) - \gamma \cdot D(M(\pi) \| M'(\pi))]$$

$$\leq \mathbb{E}_{(M, \pi^\star) \sim \mu} \mathbb{E}_{\pi \sim p}[f^M(\pi^\star) - f^M(\pi)] - \gamma(c_1 c_2)^{-1} \cdot \mathbb{E}_{\pi \sim p} \mathbb{E}_{z|\pi}[D(\mu_{\mathrm{po}}(\cdot; \pi, z) \| \mu_{\mathrm{pr}})] + \gamma\varepsilon$$

$$\leq \mathsf{inf}_{(c_1 c_2)^{-1}\gamma}^D(\mathcal{M}) + \gamma\varepsilon.$$

Taking $\varepsilon \to 0$, we conclude that $\mathsf{dec}_\gamma^D(\mathcal{M}) \leq \mathsf{inf}_{(c_1 c_2)^{-1}\gamma}^D(\mathcal{M})$. By Proposition 3.1, $\mathsf{inf}_\gamma^D(\mathcal{M}) = \mathsf{inf}_\gamma^D(\mathrm{co}(\mathcal{M}))$, so applying the result to $\mathrm{co}(\mathcal{M})$ yields

$$\mathsf{dec}_\gamma^D(\mathrm{co}(\mathcal{M})) \leq \mathsf{inf}_{(c_1 c_2)^{-1}\gamma}^D(\mathcal{M}).$$

**Bounding the information ratio by the DEC.** We now consider the opposite direction. Fix a prior $\mu \in \Delta(\mathcal{M} \times \Pi)$ and consider the value for the parameterized information ratio:

$$\mathbb{E}_{(M, \pi^\star) \sim \mu} \mathbb{E}_{\pi \sim p}[f^M(\pi^\star) - f^M(\pi)] - \gamma \cdot \mathbb{E}_{\pi \sim p} \mathbb{E}_{z|\pi}[D(\mu_{\mathrm{po}}(\cdot; \pi, z) \| \mu_{\mathrm{pr}})].$$

Define $\overline{M}_{\pi'}(\pi) := \mathbb{E}_\mu[M(\pi) \mid \pi^\star = \pi']$ and $\overline{M}(\pi) = \mathbb{E}_{M \sim \mu}[M(\pi)]$. Using that $(\pi^\star, \pi)$ are independent, along with Property 3, we have

$$\mathbb{E}_{z|\pi}[D(\mu_{\mathrm{po}}(\cdot; \pi, z) \| \mu_{\mathrm{pr}})] = \mathbb{E}_{z|\pi}\big[D\big(\mathbb{P}_{\pi^\star|\pi,z} \| \mathbb{P}_{\pi^\star}\big)\big]$$

$$= \mathbb{E}_{z|\pi}\big[D\big(\mathbb{P}_{\pi^\star|\pi,z} \| \mathbb{P}_{\pi^\star|\pi}\big)\big] \geq c_1^{-1} \mathbb{E}_{\pi^\star \sim \mu}\big[D\big(\overline{M}_{\pi^\star}(\pi) \| \overline{M}(\pi)\big)\big].$$

Next, observe that we have

$$\mathbb{E}_{(M, \pi^\star) \sim \mu} \mathbb{E}_{\pi \sim p}[f^M(\pi^\star) - f^M(\pi)] = \mathbb{E}_{\pi \sim p} \mathbb{E}_{\pi^\star \sim \mu} \mathbb{E}[f^M(\pi^\star) - f^M(\pi) \mid \pi^\star]$$

$$= \mathbb{E}_{\pi \sim p} \mathbb{E}_{\pi^\star \sim \mu}\big[f^{\overline{M}_{\pi^\star}}(\pi^\star) - f^{\overline{M}_{\pi^\star}}(\pi)\big]$$

$$\leq \mathbb{E}_{\pi^\star \sim \mu} \mathbb{E}_{\pi \sim p}\Big[\max_{\pi'} f^{\overline{M}_{\pi^\star}}(\pi') - f^{\overline{M}_{\pi^\star}}(\pi)\Big].$$

Recall that the definition of $\mathsf{dec}_\gamma(\mathrm{co}(\mathcal{M}))$ implies the following: For any $\kappa \in \Delta(\mathcal{M})$ there exists a distribution $p \in \Delta(\Pi)$ such that for all $\nu \in \Delta(\mathcal{M})$, defining $\overline{M}_\kappa(\pi) := \mathbb{E}_{M \sim \kappa}[M(\pi)]$ and $\overline{M}_\nu(\pi) := \mathbb{E}_{M \sim \nu}[M(\pi)]$, we have

$$\mathbb{E}_{\pi \sim p}\Big[\max_{\pi'} f^{\overline{M}_\nu}(\pi') - f^{\overline{M}_\nu}(\pi) - \gamma \cdot D\big(\overline{M}_\nu(\pi) \,\|\, \overline{M}_\kappa(\pi)\big)\Big] \leq \mathsf{dec}_\gamma(\mathrm{co}(\mathcal{M})). \qquad (37)$$

By invoking (37) with $\overline{M}_\kappa = \overline{M}$ and $\overline{M}_\nu = \overline{M}_{\pi^\star}$, we are guaranteed that for every draw of $\pi^\star$

$$\mathbb{E}_{\pi \sim p}\Big[\max_{\pi'} f^{\overline{M}_{\pi^\star}}(\pi') - f^{\overline{M}_{\pi^\star}}(\pi)\Big] \leq \gamma c_1^{-1} \cdot \mathbb{E}_{\pi \sim p}\big[D\big(\overline{M}_{\pi^\star}(\pi) \,\|\, \overline{M}(\pi)\big)\big] + \mathsf{dec}_{c_1^{-1}\gamma}(\mathrm{co}(\mathcal{M})).$$

Taking the expectation over $\pi^\star \sim \mu$, we conclude that

$$\mathsf{inf}_\gamma^D(\mathcal{M}) \leq \mathsf{dec}_{c_1^{-1}\gamma}(\mathrm{co}(\mathcal{M})).$$

$\qquad \square$

### E.3 High-Probability Exploration-By-Optimization and Information Ratio (Theorem 3.2)

**Theorem 3.2.** *For all $\eta > 0$, $\mathsf{inf}_{\eta^{-1}}^{\mathsf{H}}(\mathcal{M}) \leq \mathsf{exo}_\eta(\mathcal{M}) \leq \mathsf{inf}_{(8\eta)^{-1}}^{\mathsf{H}}(\mathcal{M})$.*

**Proof of Theorem 3.2.** We first state the following basic result, which is proven in the sequel.

**Lemma E.1.** *For any fixed $M \in \mathcal{M}$ and $\pi^\star \in \Pi$, the map $(p, g) \mapsto \Gamma_{q,\eta}(p, g\,; \pi^\star, M)$ is jointly convex with respect to $(p, g) \in \Delta(\Pi) \times \mathcal{G}$, where $\mathcal{G} := (\Pi \times \Pi \times \mathcal{Z} \to \mathbb{R})$.*

**Upper bound: Minimax theorem.** We first use the minimax theorem to move to a Bayesian counterpart to the Exploration-by-Optimization objective. This requires some care to ensure boundedness and compactness, but otherwise is conceptually straightforward. To begin, observe that we can write the Exploration-by-Optimization objective as

$$\mathsf{exo}_\eta(\mathcal{M}) = \sup_{q \in \Delta(\Pi)} \inf_{p \in \Delta(\Pi), g \in \mathcal{G}} \sup_{M \in \mathcal{M}, \pi^\star \in \Pi} [\Gamma_{q,\eta}(p, g\,; \pi^\star, M)]$$
$$= \sup_{q \in \Delta(\Pi)} \inf_{p \in \Delta(\Pi), g \in \mathcal{G}} \sup_{\mu \in \Delta(\mathcal{M} \times \Pi)} \mathbb{E}_{(M, \pi^\star) \sim \mu}[\Gamma_{q,\eta}(p, g\,; \pi^\star, M)].$$

Fix $\alpha \geq 1 \vee \eta^{-1}$ and $\varepsilon \in (0, 1)$, and define

$$\mathcal{G}_\alpha = \{g \in \mathcal{G} \mid \|g\|_\infty \leq \alpha\}, \quad \text{and} \quad \mathcal{P}_\varepsilon = \{p \in \Delta(\Pi) \mid p(\pi) \geq \varepsilon|\Pi|^{-1} \ \forall \pi\}.$$

Then, by restricting to these classes, we have[16]

$$\mathsf{exo}_\eta(\mathcal{M}) \leq \sup_{q \in \Delta(\Pi)} \inf_{p \in \mathcal{P}_\varepsilon, g \in \mathcal{G}_\alpha} \sup_{\mu \in \Delta(\mathcal{M} \times \Pi)} \mathbb{E}_{(M, \pi^\star) \sim \mu}[\Gamma_{q,\eta}(p, g\,; \pi^\star, M)]$$

We verify that the conditions required to apply the minimax theorem are satisfied.

- The map $\mu \mapsto \mathbb{E}_{(M, \pi^\star) \sim \mu}[\Gamma_{q,\eta}(p, g\,; \pi^\star, M)]$ is linear. Furthermore, by Lemma E.1, the map $(p, g) \mapsto \mathbb{E}_{(M, \pi^\star) \sim \mu}[\Gamma_{q,\eta}(p, g\,; \pi^\star, M)]$ is convex.

- Since we have restricted to $p \in \mathcal{P}_\varepsilon$ and $g \in \mathcal{G}_\alpha$, the value $\Gamma_{q,\eta}(p, g; \pi^\star, M)$ is uniformly bounded, as well as continuous with respect to $p$ and $g$ (so long as $\varepsilon > 0$ and $\alpha < \infty$).

- The set $\Delta(\mathcal{M} \times \Pi)$ is convex. Since $|\Pi| < \infty$, the set $\mathcal{P}_\varepsilon \times \mathcal{G}_\alpha$ is convex and compact (for $\mathcal{P}_\varepsilon$ equipped with the usual topology and $\mathcal{G}_\alpha$ equipped with the product topology; see Lattimore and György [34] for details).

Hence, using Lemma C.2 we can bound by the value of the Bayesian game as follows:

$$\mathsf{exo}_\eta(\mathcal{M}) \leq \sup_{q \in \Delta(\Pi)} \sup_{\mu \in \Delta(\mathcal{M} \times \Pi)} \inf_{p \in \mathcal{P}_\varepsilon, g \in \mathcal{G}_\alpha} \mathbb{E}_{(M, \pi^\star) \sim \mu}[\Gamma_{q,\eta}(p, g\,; \pi^\star, M)]. \qquad (38)$$

---

[16]Restricting to these sets allows us to enforce boundedness and continuity of the Exploration-by-Optimization objective, which is necessary to appeal to the minimax theorem. The parameters $\alpha$ and $\varepsilon$ will not enter the final bound quantitatively.

**Upper bound: Moving to Hellinger distance.** For any $q \in \Delta(\Pi)$, $\mu \in \Delta(\mathcal{M} \times \Pi)$, and $p \in \mathcal{P}_\varepsilon$ the value of the game in (38) is

$$\mathbb{E}_{(M,\pi^\star)\sim\mu} \mathbb{E}_{\pi\sim p}[f^M(\pi^\star) - f^M(\pi)]$$
$$+ \eta^{-1} \inf_{g\in\mathcal{G}_\alpha} \mathbb{E}_{(M,\pi^\star)\sim\mu}\left[\mathbb{E}_{\pi\sim p, z\sim M(\pi)} \mathbb{E}_{\pi'\sim q} \exp\left(\frac{\eta}{p(\pi)}(g(\pi';\pi,z) - g(\pi^\star;\pi,z))\right) - 1\right].$$

Using Bayes' rule, we can rewrite the second term above as

$$\inf_{g\in\mathcal{G}_\alpha} \mathbb{E}_{\pi\sim p} \mathbb{E}_{z|\pi}\left[\mathbb{E}_{\pi'\sim q}\left[\exp\left(\eta\frac{g(\pi';\pi,z)}{p(\pi)}\right)\right] \cdot \mathbb{E}_{\pi^\star\sim\mu_{\mathrm{po}}(\cdot\,;\pi,z)}\left[\exp\left(-\eta\frac{g(\pi^\star;\pi,z)}{p(\pi)}\right)\right] - 1\right]$$

By reparameterizing via $g(\pi';\pi,z) \leftarrow \frac{p(\pi)}{\eta} g(\pi';\pi,z)$, the value is upper bounded by

$$\inf_{g\in\mathcal{G}_{\alpha\eta}} \mathbb{E}_{\pi\sim p} \mathbb{E}_{z|\pi}\left[\mathbb{E}_{\pi'\sim q}[\exp(g(\pi';\pi,z))] \cdot \mathbb{E}_{\pi^\star\sim\mu_{\mathrm{po}}(\cdot\,;\pi,z)}[\exp(-g(\pi^\star;\pi,z))] - 1\right].$$

Furthermore, by skolemizing, we can rewrite this as

$$V(p,q,\mu) := \mathbb{E}_{\pi\sim p} \mathbb{E}_{z|\pi} \inf_{g:\Pi\to\mathbb{R}, \|g\|_\infty\leq\alpha\eta}\left\{\mathbb{E}_{\pi'\sim q}[\exp(g(\pi'))] \cdot \mathbb{E}_{\pi^\star\sim\mu_{\mathrm{po}}(\cdot\,;\pi,z)}[\exp(-g(\pi^\star))] - 1\right\}.$$

We now appeal to Lemma C.5, which grants that

$$V(p,q,\mu) \leq -\frac{1}{2} \mathbb{E}_{\pi\sim p} \mathbb{E}_{z|\pi}\left[D_{\mathsf{H}}^2(\mu_{\mathrm{po}}(\cdot\,;\pi,z),q)\right] + 4e^{-\alpha\eta}. \tag{39}$$

Using (39), we have

$\mathsf{exo}_\eta(\mathcal{M})$

$$\leq \sup_{q\in\Delta(\Pi)} \sup_{\mu\in\Delta(\mathcal{M}\times\Pi)} \inf_{p\in\mathcal{P}_\varepsilon} \left\{\mathbb{E}_{(M,\pi^\star)\sim\mu} \mathbb{E}_{\pi\sim p}[f^M(\pi^\star) - f^M(\pi)] - \frac{1}{2\eta} \mathbb{E}_{\pi\sim p} \mathbb{E}_{z|\pi}\left[D_{\mathsf{H}}^2(\mu_{\mathrm{po}}(\cdot\,;\pi,z),q)\right]\right\} + 4\eta^{-1}e^{-\alpha\eta}.$$

In addition, since $f^M \in [0,1]$ and $D_{\mathsf{H}}^2(\cdot,\cdot) \in [0,2]$, we can further upper bound by

$$\sup_{q\in\Delta(\Pi)} \sup_{\mu\in\Delta(\mathcal{M}\times\Pi)} \inf_{p\in\Delta(\Pi)} \left\{\mathbb{E}_{(M,\pi^\star)\sim\mu} \mathbb{E}_{\pi\sim p}[f^M(\pi^\star) - f^M(\pi)] - \frac{1}{2\eta} \mathbb{E}_{\pi\sim p} \mathbb{E}_{z|\pi}\left[D_{\mathsf{H}}^2(\mu_{\mathrm{po}}(\cdot\,;\pi,z),q)\right]\right\}$$
$$+ O(\eta^{-1}e^{-\alpha\eta} + \varepsilon \cdot (1 + \eta^{-1})).$$

Since this expression only depends on $\alpha$ and $\varepsilon$ through the additive approximation terms, taking the limit as $\alpha \to \infty$ and $\varepsilon \to 0$ yields

$$\mathsf{exo}_\eta(\mathcal{M}) \leq \sup_{q\in\Delta(\Pi)} \sup_{\mu\in\Delta(\mathcal{M}\times\Pi)} \inf_{p\in\Delta(\Pi)} \left\{\mathbb{E}_{(M,\pi^\star)\sim\mu} \mathbb{E}_{\pi\sim p}[f^M(\pi^\star) - f^M(\pi)] - \frac{1}{2\eta} \mathbb{E}_{\pi\sim p} \mathbb{E}_{z|\pi}\left[D_{\mathsf{H}}^2(\mu_{\mathrm{po}}(\cdot\,;\pi,z),q)\right]\right\}.$$

Finally, recall that since Hellinger distance satisfies the triangle inequality, we have

$$\mathbb{E}_{\pi\sim p} \mathbb{E}_{z|\pi}\left[D_{\mathsf{H}}^2(\mu_{\mathrm{po}}(\cdot\,;\pi,z),\mu_{\mathrm{pr}})\right] \leq 2\,\mathbb{E}_{\pi\sim p} \mathbb{E}_{z|\pi}\left[D_{\mathsf{H}}^2(\mu_{\mathrm{po}}(\cdot\,;\pi,z),q)\right] + 2D_{\mathsf{H}}^2(\mu_{\mathrm{pr}},q).$$

Using that $\mu_{\mathrm{pr}}(\pi') = \mathbb{E}_{\pi\sim p} \mathbb{E}_{z|\pi}[\mu_{\mathrm{po}}(\pi';\pi,z)]$ and that squared Hellinger distance is convex, we have $D_{\mathsf{H}}^2(\mu_{\mathrm{pr}},q) \leq \mathbb{E}_{\pi\sim p} \mathbb{E}_{z|\pi}\left[D_{\mathsf{H}}^2(\mu_{\mathrm{po}}(\cdot\,;\pi,z),q)\right]$, and so

$$\mathbb{E}_{\pi\sim p} \mathbb{E}_{z|\pi}\left[D_{\mathsf{H}}^2(\mu_{\mathrm{po}}(\cdot\,;\pi,z),\mu_{\mathrm{pr}})\right] \leq 4 \cdot \mathbb{E}_{\pi\sim p} \mathbb{E}_{z|\pi}\left[D_{\mathsf{H}}^2(\mu_{\mathrm{po}}(\cdot\,;\pi,z),q)\right].$$

It follows that

$$\mathsf{exo}_\eta(\mathcal{M}) \leq \sup_{\mu\in\Delta(\mathcal{M}\times\Pi)} \inf_{p\in\Delta(\Pi)} \left\{\mathbb{E}_{(M,\pi^\star)\sim\mu} \mathbb{E}_{\pi\sim p}[f^M(\pi^\star) - f^M(\pi)] - \frac{1}{8\eta} \mathbb{E}_{\pi\sim p} \mathbb{E}_{z|\pi}\left[D_{\mathsf{H}}^2(\mu_{\mathrm{po}}(\cdot\,;\pi,z),\mu_{\mathrm{pr}})\right]\right\}$$
$$= \mathsf{inf}_{(8\eta)^{-1}}^{\mathsf{H}}(\mathcal{M}).$$

**Lower bound.** It is immediate (without having to invoke the minimax theorem) that

$$\mathsf{exo}_\eta(\mathcal{M}) = \sup_{q\in\Delta(\Pi)} \inf_{p\in\Delta(\Pi),g\in\mathcal{G}} \sup_{\mu\in\Delta(\mathcal{M}\times\Pi)} \mathbb{E}_{(M,\pi^\star)\sim\mu}[\Gamma_{q,\eta}(p,g\,;\pi^\star,M)]$$
$$\geq \sup_{q\in\Delta(\Pi)} \sup_{\mu\in\Delta(\mathcal{M}\times\Pi)} \inf_{p\in\Delta(\Pi),g\in\mathcal{G}} \mathbb{E}_{(M,\pi^\star)\sim\mu}[\Gamma_{q,\eta}(p,g\,;\pi^\star,M)].$$

Performing the same sequence of calculations as in the upper bound, we have that for any $q \in \Delta(\Pi)$, $\mu \in \Delta(\mathcal{M} \times \Pi)$, and $p \in \Delta(\Pi)$,

$$\inf_{g \in \mathcal{G}} \mathbb{E}_{(M,\pi^\star) \sim \mu}[\Gamma_{q,\eta}(p, g\,; \pi^\star, M)]$$

$$= \mathbb{E}_{(M,\pi^\star) \sim \mu} \mathbb{E}_{\pi \sim p}[f^M(\pi^\star) - f^M(\pi)]$$

$$+ \eta^{-1} \inf_{g \in \mathcal{G}} \mathbb{E}_{(M,\pi^\star) \sim \mu} \left[ \mathbb{E}_{\pi \sim p, z \sim M(\pi)} \mathbb{E}_{\pi' \sim q} \exp\left( \frac{\eta}{p(\pi)}(g(\pi'\,; \pi, z) - g(\pi^\star\,; \pi, z)) \right) - 1 \right]$$

$$= \mathbb{E}_{(M,\pi^\star) \sim \mu} \mathbb{E}_{\pi \sim p}[f^M(\pi^\star) - f^M(\pi)] + \eta^{-1} \mathbb{E}_{\pi \sim p} \mathbb{E}_{z|\pi} \inf_{g \in \mathcal{G}} \left\{ \mathbb{E}_{\pi' \sim q}[\exp(g(\pi'))] \cdot \mathbb{E}_{\pi^\star \sim \mu_{\mathrm{po}}(\cdot\,; \pi, z)}[\exp(-g(\pi^\star))] - 1 \right\}.$$

Using Lemma 2.1, we have

$$\mathbb{E}_{\pi \sim p} \mathbb{E}_{z|\pi} \inf_{g \in \mathcal{G}} \left\{ \mathbb{E}_{\pi' \sim q}[\exp(g(\pi'))] \cdot \mathbb{E}_{\pi^\star \sim \mu_{\mathrm{po}}(\cdot\,; \pi, z)}[\exp(-g(\pi^\star))] - 1 \right\} \geq - \mathbb{E}_{\pi \sim p} \mathbb{E}_{z|\pi} \left[ D_{\mathsf{H}}^2(\mu_{\mathrm{po}}(\cdot\,; \pi, z), q) \right].$$

We conclude that

$$\mathsf{exo}_\eta(\mathcal{M}) \geq \sup_{q \in \Delta(\Pi)} \sup_{\mu \in \Delta(\mathcal{M} \times \Pi)} \inf_{p \in \Delta(\Pi)} \left\{ \mathbb{E}_{(M,\pi^\star) \sim \mu} \mathbb{E}_{\pi \sim p}[f^M(\pi^\star) - f^M(\pi)] - \frac{1}{\eta} \mathbb{E}_{\pi \sim p} \mathbb{E}_{z|\pi} \left[ D_{\mathsf{H}}^2(\mu_{\mathrm{po}}(\cdot\,; \pi, z), q) \right] \right\}$$

$$\geq \sup_{\mu \in \Delta(\mathcal{M} \times \Pi)} \inf_{p \in \Delta(\Pi)} \left\{ \mathbb{E}_{(M,\pi^\star) \sim \mu} \mathbb{E}_{\pi \sim p}[f^M(\pi^\star) - f^M(\pi)] - \frac{1}{\eta} \mathbb{E}_{\pi \sim p} \mathbb{E}_{z|\pi} \left[ D_{\mathsf{H}}^2(\mu_{\mathrm{po}}(\cdot\,; \pi, z), \mu_{\mathrm{pr}}) \right] \right\}$$

$$= \mathsf{inf}_{\eta^{-1}}^{\mathsf{H}}(\mathcal{M}).$$

$\square$

**Proof of Lemma E.1.** Let $M \in \mathcal{M}$ and $\pi^\star \in \Pi$ be fixed. The map $p \mapsto \mathbb{E}_{\pi \sim p}[f^M(\pi_M) - f^M(\pi)]$ is linear, so our main task is to show that the function

$$(p, g) \mapsto \sum_\pi p(\pi) \mathbb{E}_{z \sim M(\pi)} \left[ \sum_{\pi'} q(\pi') \exp\left( \frac{\eta}{p(\pi)}(g(\pi'\,; \pi, z) - g(\pi^\star\,; \pi, z)) \right) \right]$$

is jointly convex. We can rewrite this as

$$\sum_\pi q(\pi') \sum_\pi p(\pi) \mathbb{E}_{z \sim M(\pi)} \left[ \exp\left( \frac{\eta}{p(\pi)}(g(\pi'\,; \pi, z) - g(\pi^\star\,; \pi, z)) \right) \right].$$

Since convexity is preserved under summation with non-negative weights, it suffices to show that for any fixed $(\pi, \pi')$, the map

$$(p(\pi), g) \mapsto p(\pi) \mathbb{E}_{z \sim M(\pi)} \left[ \exp\left( \frac{\eta}{p(\pi)}(g(\pi'\,; \pi, z) - g(\pi^\star\,; \pi, z)) \right) \right] \tag{40}$$

is convex. Since the function $g \mapsto \mathbb{E}_{z \sim M(\pi)}[\exp(\eta(g(\pi'\,; \pi, z) - g(\pi^\star\,; \pi, z)))]$ is convex over $\mathcal{G}$, convexity for (40) follows from the following standard result.

**Proposition E.2** (Convexity of perspective transformation). *Let $f : \mathbb{R}^d \to (-\infty, \infty)$ be a convex function. Then the function*

$$(x, t) \mapsto t \cdot f(x/t)$$

*is convex over $\mathbb{R}^d \times \mathbb{R}_+$.*

$\square$

# F   Proofs for Main Results (Section 2)

## F.1   Proof of Theorem 2.1

**Theorem 2.1** (Main upper bound). *For any choice of $\eta > 0$, Algorithm 1 ensures that for all $\delta > 0$, with probability at least $1 - \delta$,*

$$\mathbf{Reg}_{\mathsf{DM}} \leq \mathsf{dec}_{(8\eta)^{-1}}(\mathrm{co}(\mathcal{M})) \cdot T + 2\eta^{-1} \cdot \log(|\Pi|/\delta). \tag{8}$$

*In particular, for any $\delta > 0$, with appropriate $\eta$, the algorithm has that with probability at least $1 - \delta$,*

$$\mathbf{Reg}_{\mathsf{DM}} \leq O(1) \cdot \inf_{\gamma > 0} \{ \mathsf{dec}_\gamma(\mathrm{co}(\mathcal{M})) \cdot T + \gamma \cdot \log(|\Pi|/\delta) \}. \tag{9}$$

 **Proof of Theorem 2.1.** Let us adopt convention $\langle p, f \rangle = \sum_\pi p(\pi) \cdot f(\pi)$ and let $e_\pi$ denote the $\pi$th
 standard basis vector in $\mathbb{R}^\Pi$. For each $\pi^\star \in \Pi$, we write regret as

$$\mathbf{Reg}_{\mathsf{DM}}(\pi^\star) = \sum_{t=1}^{T} \mathbb{E}_{\pi \sim p^{(t)}} \left[ f^{M^{(t)}}(\pi^\star) - f^{M^{(t)}}(\pi) \right] = \sum_{t=1}^{T} \langle e_{\pi^\star} - p^{(t)}, f^{M^{(t)}} \rangle.$$

 Adding and subtracting $\sum_{t=1}^{T} \langle e_{\pi^\star} - q^{(t)}, \widehat{f}^{(t)} \rangle$, we rewrite this as

$$\sum_{t=1}^{T} \langle e_{\pi^\star} - p^{(t)}, f^{M^{(t)}} \rangle = \sum_{t=1}^{T} \langle e_{\pi^\star} - p^{(t)}, f^{M^{(t)}} \rangle + \sum_{t=1}^{T} \langle e_{\pi^\star} - q^{(t)}, \widehat{f}^{(t)} \rangle - \sum_{t=1}^{T} \langle e_{\pi^\star} - q^{(t)}, \widehat{f}^{(t)} \rangle.$$
$$(41)$$

 The exponential weights update ensures (Lemma C.6) that with probability 1,

$$\sum_{t=1}^{T} \langle e_{\pi^\star} - q^{(t)}, \widehat{f}^{(t)} \rangle \leq \sum_{t=1}^{T} \langle q^{(t+1)} - q^{(t)}, \widehat{f}^{(t)} \rangle - \frac{1}{\eta} \sum_{t=1}^{T} D_{\mathsf{KL}}(q^{(t+1)} \,\|\, q^{(t)}) + \frac{D_{\mathsf{KL}}(e_{\pi^\star} \,\|\, q^{(1)})}{\eta}$$
$$\leq \sum_{t=1}^{T} \langle q^{(t+1)} - q^{(t)}, \widehat{f}^{(t)} \rangle - \frac{1}{\eta} \sum_{t=1}^{T} D_{\mathsf{KL}}(q^{(t+1)} \,\|\, q^{(t)}) + \frac{\log|\Pi|}{\eta}.$$

 In addition, using Lemma C.3, we have that for all $t$,

$$\langle q^{(t+1)}, \widehat{f}^{(t)} \rangle - \frac{1}{\eta} D_{\mathsf{KL}}(q^{(t+1)} \,\|\, q^{(t)}) \leq \frac{1}{\eta} \log \left( \sum_\pi q^{(t)}(\pi) \exp \left( \eta \cdot \widehat{f}^{(t)}(\pi) \right) \right).$$

 Hence, combining this with (41), we have

$$\mathbf{Reg}_{\mathsf{DM}}(\pi^\star) \leq \sum_{t=1}^{T} \langle e_{\pi^\star} - p^{(t)}, f^{M^{(t)}} \rangle - \langle e_{\pi^\star}, \widehat{f}^{(t)} \rangle + \frac{1}{\eta} \sum_{t=1}^{T} \log \left( \sum_\pi q^{(t)}(\pi) \exp \left( \eta \cdot \widehat{f}^{(t)}(\pi) \right) \right) + \frac{\log|\Pi|}{\eta}.$$

 Let $\mathscr{F}_t := \sigma(\pi^{(1)}, z^{(1)}, \dots, \pi^{(t)}, z^{(t)})$ be a filtration, and let $\mathbb{E}_t[\cdot] := \mathbb{E}[\cdot \mid \mathscr{F}_t]$. For each $\pi \in \Pi$,
 define a sequence of random variables $\{X_t(\pi)\}_{t=1}^{T}$ via

$$X_t(\pi) = \frac{1}{\eta} \log \left( \sum_{\pi'} q^{(t)}(\pi') \exp \left( \eta \cdot \widehat{f}^{(t)}(\pi') \right) \right) - \langle e_\pi, \widehat{f}^{(t)} \rangle.$$

 Using Lemma C.1 and a union bound, we have that for any $\eta > 0$, with probability at least $1 - \delta$, for
 all $\pi \in \Pi$

$$\sum_{t=1}^{T} X_t(\pi) \leq \frac{1}{\eta} \sum_{t=1}^{T} \log(\mathbb{E}_{t-1}[\exp(\eta X_t(\pi))]) + \frac{\log(|\Pi|/\delta)}{\eta}.$$

 Since this bounded holds uniformly for all $\pi$, we have that with probability at least $1 - \delta$, for all
 $\pi^\star \in \Pi$,

$$\mathbf{Reg}_{\mathsf{DM}}(\pi^\star) \leq \sum_{t=1}^{T} \langle e_{\pi^\star} - p^{(t)}, f^{M^{(t)}} \rangle + \frac{1}{\eta} \sum_{t=1}^{T} \log(\mathbb{E}_{t-1}[\exp(\eta X_t(\pi^\star))]) + 2\frac{\log(|\Pi|/\delta)}{\eta}.$$

 We compute that for any $\pi^\star \in \Pi$,

$$\log(\mathbb{E}_{t-1}[\exp(\eta X_t(\pi^\star))])$$
$$= \log \left( \mathbb{E}_{\pi \sim p^{(t)}} \mathbb{E}_{z \sim M^{(t)}(\pi)} \mathbb{E}_{\pi' \sim q^{(t)}} \left[ \exp \left( \frac{\eta}{p^{(t)}(\pi)} \cdot \left( g^{(t)}(\pi'; \pi, z) - g^{(t)}(\pi^\star; \pi, z) \right) \right) \right] \right)$$
$$\leq \mathbb{E}_{\pi \sim p^{(t)}} \mathbb{E}_{z \sim M^{(t)}(\pi)} \mathbb{E}_{\pi' \sim q^{(t)}} \left[ \exp \left( \frac{\eta}{p^{(t)}(\pi)} \cdot \left( g^{(t)}(\pi'; \pi, z) - g^{(t)}(\pi^\star; \pi, z) \right) \right) \right] - 1,$$

where we have used that $\log(x) \le x - 1$ for $x > 0$. Hence, with probability at least $1 - \delta$, for all $\pi^\star \in \Pi$,

$$
\begin{aligned}
\mathbf{Reg}_{\mathsf{DM}}(\pi^\star) \le{}& \sum_{t=1}^{T} \left\langle e_{\pi^\star} - p^{(t)}, f^{M^{(t)}} \right\rangle + 2\frac{\log(|\Pi|/\delta)}{\eta} \\
&+ \frac{1}{\eta} \left( \mathbb{E}_{\pi \sim p^{(t)}} \mathbb{E}_{z \sim M^{(t)}(\pi)} \mathbb{E}_{\pi' \sim q^{(t)}} \left[ \exp\left( \frac{\eta}{p^{(t)}(\pi)} \cdot \left( g^{(t)}(\pi'\,;\pi, z) - g^{(t)}(\pi^\star\,;\pi, z) \right) \right) \right] - 1 \right) \\
={}& \sum_{t=1}^{T} \Gamma_{q^{(t)},\eta}(p^{(t)}, g^{(t)}\,;\pi^\star, M^{(t)}) + 2\frac{\log(|\Pi|/\delta)}{\eta} \\
\le{}& \mathsf{exo}_\eta(\mathcal{M}) \cdot T + 2\frac{\log(|\Pi|/\delta)}{\eta},
\end{aligned}
$$

where the last line uses that $(p^{(t)}, g^{(t)})$ are chosen to minimize the Exploration-By-Optimization objective. Finally, using Corollary 3.1, we have that $\mathsf{exo}_\eta(\mathcal{M}) \le \mathsf{dec}_{(8\eta)^{-1}}(\mathrm{co}(\mathcal{M}))$.

$\qquad\square$

## F.2   Proof of Theorem 2.2

In this section we prove Theorem 2.2. Most of the work consists of proving an improved lower bound for the *stochastic* setting in which $M^{(t)} = M^\star$ is fixed across $t$ (Theorem F.1). We then appeal to this stochastic lower bound with the class $\mathrm{co}(\mathcal{M})$. Since $\mathrm{co}(\mathcal{M})$ is equivalent to the set of mixtures of models in $\mathcal{M}$, this establishes existence of distribution $\mu \in \Delta(\mathcal{M})$ and mixture model $M_\mu = \mathbb{E}_{M \sim \mu}[M]$ for which regret in the stochastic setting must scale with $\mathsf{dec}_{\gamma,\varepsilon_\gamma}(\mathrm{co}(\mathcal{M}))$. The proof concludes by arguing that this yields a lower bound for the adversarial setting when we sample $M^{(t)} \sim \mu$.

Throughout this section, we define the *one-sided variance* for a random variable $Z$ as

$$
\mathbb{V}_+[Z] := \mathbb{E}\big[(Z - \mathbb{E}[Z])_+^2\big].
$$

**Theorem 2.2** (Main lower bound). *Let $C(T) := c \cdot \log(T \wedge V(\mathcal{M}))$ for a sufficiently large numerical constant $c > 0$. Set $\varepsilon_\gamma := \frac{\gamma}{4C(T)T}$. For any algorithm, there exists an oblivious adversary for which*

$$
\mathbb{E}[\mathbf{Reg}_{\mathsf{DM}}] + \sqrt{\mathbb{E}(\mathbf{Reg}_{\mathsf{DM}})_+^2} \ge \Omega(1) \cdot \sup_{\gamma > \sqrt{2C(T)T}} \mathsf{dec}_{\gamma,\varepsilon_\gamma}(\mathrm{co}(\mathcal{M})) \cdot T - O(T^{1/2}). \tag{13}
$$

We also have the following slight variant of Theorem 2.2.

**Theorem 2.2a.** *Let $C(T) := c \cdot \log(T \wedge V(\mathcal{M}))$ for a sufficiently large numerical constant $c > 0$. Set $\varepsilon_\gamma := \frac{\gamma}{4C(T)T}$. For any algorithm, there exists an oblivious adversary for which $\mathbb{E}[\mathbf{Reg}_{\mathsf{DM}}] \ge 0$ and*

$$
\mathbb{E}[\mathbf{Reg}_{\mathsf{DM}}] + \sqrt{\mathbb{E}[\mathbf{Reg}_{\mathsf{DM}}] \cdot T} \ge \Omega(1) \cdot \sup_{\gamma > \sqrt{2C(T)T}} \mathsf{dec}_{\gamma,\varepsilon_\gamma}(\mathrm{co}(\mathcal{M})) \cdot T, \tag{42}
$$

**Proof of Theorem 2.2.** We invoke Theorem F.1 with the model class $\mathrm{co}(\mathcal{M})$, which implies that there exists a distribution $\mu \in \Delta(\mathcal{M})$ for which

$$
\mathbb{E}\big[\widetilde{\mathbf{Reg}}_{\mathsf{DM}}\big] + \sqrt{\mathbb{V}_+\big[\widetilde{\mathbf{Reg}}_{\mathsf{DM}}\big]} \ge L := 8^{-1} \cdot \sup_{\gamma > \sqrt{2C(T)T}} \mathsf{dec}_{\gamma,\varepsilon_\gamma}(\mathrm{co}(\mathcal{M})) \cdot T,
$$

where

$$
\widetilde{\mathbf{Reg}}_{\mathsf{DM}} := \sum_{t=1}^{T} \mathbb{E}_{\pi^{(t)} \sim p^{(t)}} \mathbb{E}_{M \sim \mu}[f^M(\pi_\mu) - f^M(\pi^{(t)})],
$$

and $\pi_\mu := \arg\max_{\pi \in \Pi} \mathbb{E}_{M \sim \mu}[f^M(\pi)]$, with the data generating process is (for each $t = 1, \dots, T$):

-   The learner samples $\pi^{(t)} \sim p^{(t)}$.

1067 • Nature samples $z^{(t)} \sim \mathbb{E}_{M\sim\mu}[M(\pi^{(t)})]$.

1068 Observe that this is equivalent in law to the following data-generating process, which constitutes an
1069 admissible adversary (with $M^{(t)} \in \mathcal{M}$):

1070 • The learner samples $\pi^{(t)} \sim p^{(t)}$.

1071 • Nature samples $M^{(t)} \sim \mu$ and $z^{(t)} \sim M^{(t)}(\pi^{(t)})$.

1072 Likewise, we can equivalently write

$$\widetilde{\mathbf{Reg}}_{\mathsf{DM}} = \sum_{t=1}^{T} \mathbb{E}_{M^{(t)}\sim\mu} \mathbb{E}_{\pi^{(t)}\sim p^{(t)}} \left[ f^{M^{(t)}}(\pi_\mu) - f^{M^{(t)}}(\pi^{(t)}) \right].$$

1073 Hence, all that remains is to relate the quantity $\widetilde{\mathbf{Reg}}_{\mathsf{DM}}$ to the realized regret $\mathbf{Reg}_{\mathsf{DM}}$ for the sequence
1074 $M^{(1)}, \ldots M^{(T)}$, which entails removing the conditional expectation over $M^{(t)} \sim \mu$. To this end, we
1075 first observe that

$$\mathbb{E}\left[\widetilde{\mathbf{Reg}}_{\mathsf{DM}}\right] = \mathbb{E}\left[ \sum_{t=1}^{T} \mathbb{E}_{\pi^{(t)}\sim p^{(t)}} \left[ f^{M^{(t)}}(\pi_\mu) - f^{M^{(t)}}(\pi^{(t)}) \right] \right]$$
$$\leq \mathbb{E}\left[ \max_{\pi^\star \in \Pi} \sum_{t=1}^{T} \mathbb{E}_{\pi^{(t)}\sim p^{(t)}} \left[ f^{M^{(t)}}(\pi^\star) - f^{M^{(t)}}(\pi^{(t)}) \right] \right] = \mathbb{E}\left[\mathbf{Reg}_{\mathsf{DM}}\right].$$

1076 Next, note that since $\widetilde{\mathbf{Reg}}_{\mathsf{DM}}$ is non-negative, $\mathbb{V}_+\left[\widetilde{\mathbf{Reg}}_{\mathsf{DM}}\right] \leq \mathbb{E}\left[(\widetilde{\mathbf{Reg}}_{\mathsf{DM}})_+^2\right]$. Define

$$\widehat{\mathbf{Reg}}_{\mathsf{DM}} := \sum_{t=1}^{T} \mathbb{E}_{\pi^{(t)}\sim p^{(t)}} \left[ f^{M^{(t)}}(\pi_\mu) - f^{M^{(t)}}(\pi^{(t)}) \right].$$

1077 Then we have

$$\mathbb{E}\left[(\widetilde{\mathbf{Reg}}_{\mathsf{DM}})_+^2\right] \leq 2\mathbb{E}\left[(\widehat{\mathbf{Reg}}_{\mathsf{DM}})_+^2\right] + 2\mathbb{E}\left[(\widetilde{\mathbf{Reg}}_{\mathsf{DM}} - \widehat{\mathbf{Reg}}_{\mathsf{DM}})^2\right]$$
$$\leq 2\mathbb{E}\left[(\mathbf{Reg}_{\mathsf{DM}})_+^2\right] + 2\mathbb{E}\left[(\widetilde{\mathbf{Reg}}_{\mathsf{DM}} - \widehat{\mathbf{Reg}}_{\mathsf{DM}})^2\right]$$
$$\leq 2\mathbb{E}\left[(\mathbf{Reg}_{\mathsf{DM}})_+^2\right] + 2T,$$

1078 where the first inequality uses that $\widehat{\mathbf{Reg}}_{\mathsf{DM}} \leq \mathbf{Reg}_{\mathsf{DM}}$ almost surely, and the second inequality uses (i)
1079 $f^M \in [0,1]$, and (ii) for any sequence of random variables $(Z_t)_{t=1}^T$ with $\mathbb{E}[Z_t \mid Z_1,\ldots,Z_{t-1}] = 0$,
1080 $\mathbb{E}\left[(\sum_{t=1}^T Z_t)^2\right] = \sum_{t=1}^T \mathbb{E}[Z_t^2]$. Putting everything together, we conclude that

$$\mathbb{E}\left[\mathbf{Reg}_{\mathsf{DM}}\right] + \sqrt{2\mathbb{E}\left[(\mathbf{Reg}_{\mathsf{DM}})_+^2\right]} \geq L - \sqrt{2T}.$$

1081 This proves Theorem 2.2. To prove Theorem 2.2a, we use that since $\widetilde{\mathbf{Reg}}_{\mathsf{DM}} \in [0,T]$,

$$\mathbb{V}_+\left[\widetilde{\mathbf{Reg}}_{\mathsf{DM}}\right] \leq T \cdot \mathbb{E}\left[\widetilde{\mathbf{Reg}}_{\mathsf{DM}}\right] \leq T \cdot \mathbb{E}[\mathbf{Reg}_{\mathsf{DM}}].$$

1082 $\square$

1083 The following result concerns the *stochastic setting* in Foster et al. [18]. Here, there is a (unknown)
1084 underlying model $M^\star \in \mathcal{M}$. For $t = 1,\ldots,T$, data is generated through the process:

1085 • Learner samples $\pi^{(t)} \sim p^{(t)}$.

1086 • Nature samples $z^{(t)} \sim M^\star(\pi^{(t)})$.

1087 In addition, regret simplifies to

$$\mathbf{Reg}_{\mathsf{DM}} = \sum_{t=1}^{T} \mathbb{E}_{\pi^{(t)}\sim p^{(t)}} \left[ f^{M^\star}(\pi_{M^\star}) - f^{M^\star}(\pi^{(t)}) \right] \tag{43}$$

1088 For a fixed algorithm, let $\mathbb{P}^M$ denote the law of $\mathcal{H}^{(T)}$ when $M^\star = M$, and let $\mathbb{E}^M[\cdot]$ and $\mathbb{V}_+ \sup M[\cdot]$
1089 denote the corresponding expectation non-negative variance. Our main lower bound for the stochastic
1090 setting is as follows.

**Theorem F.1.** *Let $C(T) := 2^9 \log(T \wedge V(\mathcal{M}))$, and set $\varepsilon_\gamma = \frac{\gamma}{4C(T)T}$. For any algorithm, there exists a model in $\mathcal{M}$ for which*

$$\mathbb{E}^M[\mathbf{Reg}_{\mathsf{DM}}] + \sqrt{\mathbb{V}_+^M[\mathbf{Reg}_{\mathsf{DM}}]} \geq 8^{-1} \cdot \sup_{\gamma \geq 4\sqrt{C(T)T}} \sup_{\overline{M} \in \mathcal{M}} \mathsf{dec}_\gamma(\mathcal{M}_{\varepsilon_\gamma}(\overline{M}), \overline{M}) \cdot T.$$

The general structure of the lower bound follows that of Theorem 3.1 in Foster et al. [18], with the main difference being that we use a more refined change-of-measure argument to move from a "reference" model $\overline{M} \in \mathcal{M}$ to a worst-case alternative. Specifically, we replace Lemma A.11 in Foster et al. [18], which requires an almost sure bound on the random variables under consideration (in our case, regret), with Lemma C.4, which requires only boundedness of the second moment. Combining this with a self-bounding argument that takes advantage of the localized model class yields the result.

**Proof of Theorem F.1.** Throughout this proof we will use that $\mathbf{Reg}_{\mathsf{DM}}$ is non-negative in the stochastic setting, which can be seen by inspecting (43) (in the general adversarial setting, it is possible for $\mathbf{Reg}_{\mathsf{DM}}$ to be negative).

Let us introduce some additional notation. For $M \in \mathcal{M}$, define $g^M(\pi) = f^M(\pi_M) - f^M(\pi)$, and for $p \in \Delta(\Pi)$, let $g^M(p) = \mathbb{E}_{\pi \sim p}[g^M(\pi)]$. Let $\widehat{p} := \frac{1}{T} \sum_{t=1}^T p^{(t)}$, and $p_M := \mathbb{E}^M\left[\frac{1}{T} \sum_{t=1}^T p^{(t)}\right]$.

To begin, fix $\overline{M} \in \mathcal{M}$, $\gamma > 0$, and $\varepsilon > 0$, and set

$$M = \argmax_{M \in \mathcal{M}_\varepsilon(\overline{M})} \mathbb{E}_{\pi \sim p_{\overline{M}}}\left[f^M(\pi_M) - f^M(\pi) - \gamma \cdot D_{\mathsf{H}}^2(M(\pi), \overline{M}(\pi))\right].$$

Abbreviate $\mathsf{dec}_\gamma \equiv \mathsf{dec}_\gamma(\mathcal{M}_\varepsilon(\overline{M}), \overline{M})$. The definition of the DEC implies that

$$\mathsf{dec}_\gamma \leq \mathbb{E}_{p_{\overline{M}}}[g^M(\pi)] - \gamma \cdot \mathbb{E}_{p_{\overline{M}}}\left[D_{\mathsf{H}}^2(M(\pi), \overline{M}(\pi))\right] = \mathbb{E}^{\overline{M}}[g^M(\widehat{p})] - \gamma \cdot \mathbb{E}_{p_{\overline{M}}}\left[D_{\mathsf{H}}^2(M(\pi), \overline{M}(\pi))\right]. \tag{44}$$

**Change of measure.** To proceed, we write

$$\mathbb{E}^{\overline{M}}[g^M(\widehat{p})] = \mathbb{E}^{\overline{M}}\left[g^M(\widehat{p}) - g^{\overline{M}}(\widehat{p}) - \mathbb{E}^M[g^M(\widehat{p})]\right] + \mathbb{E}^{\overline{M}}\left[g^{\overline{M}}(\widehat{p})\right] + \mathbb{E}^M[g^M(\widehat{p})]$$
$$\leq \mathbb{E}^{\overline{M}}\left[(g^M(\widehat{p}) - g^{\overline{M}}(\widehat{p}) - \mathbb{E}^M[g^M(\widehat{p})])_+\right] + \mathbb{E}^{\overline{M}}\left[g^{\overline{M}}(\widehat{p})\right] + \mathbb{E}^M[g^M(\widehat{p})]. \tag{45}$$

We recall the following technical lemma.

**Lemma C.4.** *Let $\mathbb{P}$ and $\mathbb{Q}$ be probability distributions over a measurable space $(\mathcal{X}, \mathscr{F})$. Then for all functions $h : \mathcal{X} \to \mathbb{R}$,*

$$|\mathbb{E}_{\mathbb{P}}[h(X)] - \mathbb{E}_{\mathbb{Q}}[h(X)]| \leq \sqrt{2^{-1}(\mathbb{E}_{\mathbb{P}}[h^2(X)] + \mathbb{E}_{\mathbb{Q}}[h^2(X)]) \cdot D_{\mathsf{H}}^2(\mathbb{P}, \mathbb{Q})}. \tag{25}$$

Defining $h(\widehat{p}) = (g^M(\widehat{p}) - g^{\overline{M}}(\widehat{p}) - \mathbb{E}^M[g^M(\widehat{p})])_+$, Lemma C.4 implies that

$$\mathbb{E}^{\overline{M}}\left[(g^M(\widehat{p}) - g^{\overline{M}}(\widehat{p}) - \mathbb{E}^M[g^M(\widehat{p})])_+\right]$$
$$\leq \mathbb{E}^M\left[(g^M(\widehat{p}) - g^{\overline{M}}(\widehat{p}) - \mathbb{E}^M[g^M(\widehat{p})])_+\right] + \sqrt{\left(\mathbb{E}^M[h(\widehat{p})^2] + \mathbb{E}^{\overline{M}}[h(\widehat{p})^2]\right) \cdot D_{\mathsf{H}}^2(\mathbb{P}^M, \mathbb{P}^{\overline{M}})}$$
$$\leq \mathbb{E}^M[g^M(\widehat{p})] + \sqrt{\left(\mathbb{E}^M[h(\widehat{p})^2] + \mathbb{E}^{\overline{M}}[h(\widehat{p})^2]\right) \cdot D_{\mathsf{H}}^2(\mathbb{P}^M, \mathbb{P}^{\overline{M}})}, \tag{46}$$

where we have used that $g^M, g^{\overline{M}} \geq 0$. We proceed to bound the second moment terms. First, we have

$$\mathbb{E}^M\left[h(\widehat{p})^2\right] = \mathbb{E}^M\left[(g^M(\widehat{p}) - g^{\overline{M}}(\widehat{p}) - \mathbb{E}^M[g^M(\widehat{p})])_+^2\right]$$
$$\leq \mathbb{E}^M\left[(g^M(\widehat{p}) - \mathbb{E}^M[g^M(\widehat{p})])_+^2\right]$$
$$= \mathbb{V}_+^M[g^M(\widehat{p})]. \tag{47}$$

where the first inequality uses that $g^{\overline{M}} \geq 0$. For the second variance term, we have

$$\mathbb{E}^{\overline{M}}\left[h(\widehat{p})^2\right] = \mathbb{E}^{\overline{M}}\left[(g^M(\widehat{p}) - g^{\overline{M}}(\widehat{p}) - \mathbb{E}^M[g^M(\widehat{p})])_+^2\right] \leq \mathbb{E}^{\overline{M}}\left[(g^M(\widehat{p}) - g^{\overline{M}}(\widehat{p}))_+^2\right].$$

We have
$$\mathbb{E}^{\overline{M}}\big[(g^M(\widehat{p}) - g^{\overline{M}}(\widehat{p}))_+^2\big]$$
$$= \mathbb{E}^{\overline{M}}\big[(g^M(\widehat{p}) - g^{\overline{M}}(\widehat{p}))_+(f^M(\pi_M) - f^{\overline{M}}(\pi_{\overline{M}}) + f^{\overline{M}}(\widehat{p}) - f^M(\widehat{p})_+\big]$$
$$\leq \mathbb{E}^{\overline{M}}\big[(g^M(\widehat{p}) - g^{\overline{M}}(\widehat{p}))_+(f^M(\pi_M) - f^{\overline{M}}(\pi_{\overline{M}})_+\big] + \mathbb{E}^{\overline{M}}\big[(g^M(\widehat{p}) - g^{\overline{M}}(\widehat{p}))_+ f^{\overline{M}}(\widehat{p}) - f^M(\widehat{p})_+\big].$$

For the first term above, we have
$$\mathbb{E}^{\overline{M}}\big[(g^M(\widehat{p}) - g^{\overline{M}}(\widehat{p}))_+(f^M(\pi_M) - f^{\overline{M}}(\pi_{\overline{M}})_+\big] \leq \varepsilon \cdot \mathbb{E}^{\overline{M}}\big[(g^M(\widehat{p}) - g^{\overline{M}}(\widehat{p}))_+\big] \leq \varepsilon \cdot \mathbb{E}^{\overline{M}}[g^M(\widehat{p})],$$

where we have used the localization property and the fact that $g^M, g^{\overline{M}} \geq 0$. For the second term, using the AM-GM inequality, we have
$$\mathbb{E}^{\overline{M}}\big[(g^M(\widehat{p}) - g^{\overline{M}}(\widehat{p}))_+ f^{\overline{M}}(\widehat{p}) - f^M(\widehat{p})_+\big]$$
$$\leq \frac{1}{2}\mathbb{E}^{\overline{M}}\big[(g^M(\widehat{p}) - g^{\overline{M}}(\widehat{p}))_+^2\big] + \frac{1}{2}\mathbb{E}^{\overline{M}}\big[(f^{\overline{M}}(\widehat{p}) - f^M(\widehat{p}))^2\big]$$
$$\leq \frac{1}{2}\mathbb{E}^{\overline{M}}\big[(g^M(\widehat{p}) - g^{\overline{M}}(\widehat{p}))_+^2\big] + \frac{1}{2}\mathbb{E}_{\pi \sim p_{\overline{M}}}\big[(f^M(\pi) - f^{\overline{M}}(\pi))^2\big]$$
$$\leq \frac{1}{2}\mathbb{E}^{\overline{M}}\big[(g^M(\widehat{p}) - g^{\overline{M}}(\widehat{p}))_+^2\big] + \frac{1}{2}\mathbb{E}_{\pi \sim p_{\overline{M}}}\big[D_{\mathsf{H}}^2\big(M(\pi), \overline{M}(\pi)\big)\big],$$

where the last line uses that rewards are observed and bounded in $[0,1]$. After combining these results and rearranging, we have
$$\mathbb{E}^{\overline{M}}\big[h(\widehat{p})^2\big] \leq \mathbb{E}^{\overline{M}}\big[(g^M(\widehat{p}) - g^{\overline{M}}(\widehat{p}))_+^2\big] \leq 2\varepsilon \cdot \mathbb{E}^{\overline{M}}[g^M(\widehat{p})] + \mathbb{E}_{\pi \sim p_{\overline{M}}}\big[D_{\mathsf{H}}^2\big(M(\pi), \overline{M}(\pi)\big)\big]. \quad (48)$$

From Lemma A.13 of Foster et al. [18], we have
$$D_{\mathsf{H}}^2\big(\mathbb{P}^M, \mathbb{P}^{\overline{M}}\big) \leq C(T) \cdot T \cdot \mathbb{E}_{\pi \sim p_{\overline{M}}}\big[D_{\mathsf{H}}^2\big(M(\pi), \overline{M}(\pi)\big)\big], \quad (49)$$

where $C(T) \leq 2^8 \cdot \log(T \wedge V(\mathcal{M}))$.

Combining the variance bounds with (46), we have
$$\mathbb{E}^{\overline{M}}\big[(g^M(\widehat{p}) - g^{\overline{M}}(\widehat{p}) - \mathbb{E}^M[g^M(\widehat{p})])_+\big]$$
$$\leq \mathbb{E}^M[g^M(\widehat{p})] + \sqrt{\big(\mathbb{V}_+^M[g^M(\widehat{p})] + 2\varepsilon \cdot \mathbb{E}^{\overline{M}}[g^M(\widehat{p})] + \mathbb{E}_{\pi \sim p_{\overline{M}}}\big[D_{\mathsf{H}}^2\big(M(\pi), \overline{M}(\pi)\big)\big]\big) \cdot D_{\mathsf{H}}^2(\mathbb{P}^M, \mathbb{P}^{\overline{M}})}$$
$$\leq \mathbb{E}^M[g^M(\widehat{p})] + \sqrt{2\mathbb{V}_+^M[g^M(\widehat{p})]} + \sqrt{\big(2\varepsilon \cdot \mathbb{E}^{\overline{M}}[g^M(\widehat{p})] + \mathbb{E}_{\pi \sim p_{\overline{M}}}\big[D_{\mathsf{H}}^2\big(M(\pi), \overline{M}(\pi)\big)\big]\big) \cdot D_{\mathsf{H}}^2(\mathbb{P}^M, \mathbb{P}^{\overline{M}})}$$
$$\leq \mathbb{E}^M[g^M(\widehat{p})] + \sqrt{2\mathbb{V}_+^M[g^M(\widehat{p})]} + \sqrt{C(T)T} \cdot \mathbb{E}_{\pi \sim p_{\overline{M}}}\big[D_{\mathsf{H}}^2\big(M(\pi), \overline{M}(\pi)\big)\big]$$
$$+ \sqrt{2\varepsilon\,\mathbb{E}^{\overline{M}}[g^M(\widehat{p})] \cdot C(T)T\,\mathbb{E}_{\pi \sim p_{\overline{M}}}\big[D_{\mathsf{H}}^2\big(M(\pi), \overline{M}(\pi)\big)\big]},$$

where the second inequality uses that $D_{\mathsf{H}}^2(\cdot, \cdot) \leq 2$ and the last inequality uses (49). where the second inequality uses that $D_{\mathsf{H}}^2\big(\mathbb{P}^M, \mathbb{P}^{\overline{M}}\big) \leq 2$.

Now, suppose we restrict to $\varepsilon \leq \frac{\gamma}{4TC(T)}$. Then we have
$$\sqrt{2\varepsilon \cdot \mathbb{E}^{\overline{M}}[g^M(\widehat{p})] \cdot C(T)T\,\mathbb{E}_{\pi \sim p_{\overline{M}}}\big[D_{\mathsf{H}}^2\big(M(\pi), \overline{M}(\pi)\big)\big]} \leq \sqrt{\mathbb{E}^{\overline{M}}[g^M(\widehat{p})] \cdot \frac{\gamma}{2} \cdot \mathbb{E}_{\pi \sim p_{\overline{M}}}\big[D_{\mathsf{H}}^2\big(M(\pi), \overline{M}(\pi)\big)\big]}$$
$$\leq \frac{1}{2}\mathbb{E}^{\overline{M}}[g^M(\widehat{p})] + \frac{\gamma}{4} \cdot \mathbb{E}_{\pi \sim p_{\overline{M}}}\big[D_{\mathsf{H}}^2\big(M(\pi), \overline{M}(\pi)\big)\big].$$

Altogether, we have
$$\mathbb{E}^{\overline{M}}\big[(g^M(\widehat{p}) - g^{\overline{M}}(\widehat{p}) - \mathbb{E}^M[g^M(\widehat{p})])_+\big]$$
$$\leq \mathbb{E}^M[g^M(\widehat{p})] + \sqrt{2\mathbb{V}_+^M[g^M(\widehat{p})]} + (\sqrt{C(T)T} + \gamma/4) \cdot \mathbb{E}_{\pi \sim p_{\overline{M}}}\big[D_{\mathsf{H}}^2\big(M(\pi), \overline{M}(\pi)\big)\big] + \frac{1}{2}\mathbb{E}^{\overline{M}}[g^M(\widehat{p})]$$

and, using (45),
$$\mathbb{E}^{\overline{M}}[g^M(\widehat{p})] \leq 2\,\mathbb{E}^M[g^M(\widehat{p})] + \mathbb{E}^{\overline{M}}\big[g^{\overline{M}}(\widehat{p})\big] + \sqrt{2\mathbb{V}_+^M[g^M(\widehat{p})]}$$
$$+ (\sqrt{C(T)T} + \gamma/4) \cdot \mathbb{E}_{\pi \sim p_{\overline{M}}}\big[D_{\mathsf{H}}^2\big(M(\pi), \overline{M}(\pi)\big)\big] + \frac{1}{2}\mathbb{E}^{\overline{M}}[g^M(\widehat{p})].$$

1128 After rearranging, this implies that

$$\mathbb{E}^{\overline{M}}[g^M(\widehat{p})] \le 4\,\mathbb{E}^M[g^M(\widehat{p})] + 2\,\mathbb{E}^{\overline{M}}\big[g^{\overline{M}}(\widehat{p})\big] + \sqrt{8\mathbb{V}_+^M[g^M(\widehat{p})]} + 2(\sqrt{C(T)T} + \gamma/4)\cdot \mathbb{E}_{\pi\sim p_{\overline{M}}}\big[D_{\mathsf{H}}^2\big(M(\pi),\overline{M}(\pi)\big)\big].$$
(50)

1129 **Completing the proof.** Combining (50) with (44), we have

$$\mathsf{dec}_\gamma \le 4\,\mathbb{E}^M[g^M(\widehat{p})] + 2\,\mathbb{E}^{\overline{M}}\big[g^{\overline{M}}(\widehat{p})\big] + \sqrt{8\mathbb{V}_+^M[g^M(\widehat{p})]} + \Big(2(\sqrt{C(T)T} + \gamma/4) - \gamma\Big)\cdot \mathbb{E}_{\pi\sim p_{\overline{M}}}\big[D_{\mathsf{H}}^2\big(M(\pi),\overline{M}(\pi)\big)\big].$$

1130 In particular, whenever $\gamma \ge 4\sqrt{C(T)T}$, this implies that there exists an instance $M' \in \{M, \overline{M}\}$ for
1131 which

$$\mathbb{E}^{M'}\big[g^{M'}(\widehat{p})\big] + \sqrt{\mathbb{V}_+^{M'}[g^{M'}(\widehat{p})]} \ge 8^{-1}\cdot \mathsf{dec}_\gamma.$$

1132 Finally, we observe that $g^{M'}(\widehat{p})$ is identical in law to $\mathbf{Reg}_{\mathsf{DM}}$ under $\mathbb{P}^{M'}$.

1133 $\qquad\qquad\qquad\qquad\qquad\qquad\qquad\qquad\qquad\qquad\qquad\qquad\qquad\qquad\qquad\qquad\qquad\qquad\qquad\qquad\qquad\qquad\qquad$ $\square$

## 1134 F.3 Proof of Theorem 2.3

1135 **Theorem 2.3.** *Suppose there exists $M_0 \in \mathcal{M}$ such that $f^{M_0}$ is a constant function, and that $|\Pi| < \infty$.*

1136 *1. If there exists $\rho > 0$ s.t. $\lim_{\gamma\to\infty} \mathsf{dec}_\gamma(\mathrm{co}(\mathcal{M}))\cdot \gamma^\rho = 0$, then $\lim_{T\to\infty} \frac{\mathfrak{M}(\mathcal{M},T)}{T^p} = 0$ for $p < 1$.*

1137 *2. If $\lim_{\gamma\to\infty} \mathsf{dec}_\gamma(\mathrm{co}(\mathcal{M}))\cdot \gamma^\rho > 0$ for all $\rho > 0$, then $\lim_{T\to\infty} \frac{\mathfrak{M}(\mathcal{M},T)}{T^p} = \infty$ for all $p < 1$.*

1138 *The same conclusion holds when $\Pi = \Pi_T$ grows with $T$, but has $\log|\Pi_T| = O(T^q)$ for any $q < 1$.*

1139 **Proof of Theorem 2.3.** This proof closely follows that of Theorem 3.5 in Foster et al. [18].

1140 **Upper bound.** Assume that $\lim_{\gamma\to\infty} \mathsf{dec}_\gamma(\mathrm{co}(\mathcal{M}))\cdot \gamma^\rho = 0$ for some $\rho > 0$, and that $\log|\Pi_T| =$
1141 $\widetilde{O}(T^q)$ for some $q < 1$. Using Theorem 2.1 with $\delta = 1/T$, we have that for each $T$, for all
1142 adversaries,

$$En[\mathbf{Reg}_{\mathsf{DM}}(T)] \le \widetilde{O}(\mathsf{dec}_\gamma(\mathrm{co}(\mathcal{M}))\cdot T + \gamma\cdot \log|\Pi_T|) \le \widetilde{O}(\mathsf{dec}_\gamma(\mathrm{co}(\mathcal{M}))\cdot T + \gamma\cdot T^q),$$

1143 with $\widetilde{O}(\cdot)$ hiding factors logarithmic in $T$. For each $T$, we set $\gamma = \gamma_T := T^{\frac{1-q}{1+\rho}}$; recall that $1 - q > 0$.
1144 The assumption that $\lim_{\gamma\to\infty} \mathsf{dec}_\gamma(\mathrm{co}(\mathcal{M}))\cdot \gamma^\rho = 0$, implies that for all $\varepsilon > 0$, there exists $\gamma' > 0$
1145 such that $\mathsf{dec}_\gamma(\mathrm{co}(\mathcal{M})) \le \varepsilon/\gamma^\rho$ for all $\gamma \ge \gamma'$. For $T$ sufficiently large, this implies that for all
1146 adversaries

$$\mathbb{E}[\mathbf{Reg}_{\mathsf{DM}}] \le \widetilde{O}\left(\frac{T}{\gamma_T^\rho} + \gamma_T\cdot T^q\right) = \widetilde{O}(T^{\frac{1+\rho q}{1+\rho}}).$$

1147 Defining $p' := \frac{1}{2}(p + 1) < 1$, this establishes that

$$\lim_{T\to\infty} \frac{\mathfrak{M}(\mathcal{M},T)}{T^{p'}} = 0.$$

1148 **Lower bound.** Assume that $\lim_{\gamma\to\infty} \mathsf{dec}_\gamma(\mathrm{co}(\mathcal{M}))\cdot \gamma^\rho = \infty$ for all $\rho > 0$ (this is equivalent
1149 to assuming that $\lim_{\gamma\to\infty} \mathsf{dec}_\gamma(\mathrm{co}(\mathcal{M}))\cdot \gamma^\rho > 0$ for all $\rho > 0$, as in the theorem statement). Let
1150 $\rho \in (0, 1/2)$ be fixed. Using Theorem 2.2a, we are guaranteed that for any algorithm, there exists an
1151 adversary for which $\mathbb{E}[\mathbf{Reg}_{\mathsf{DM}}] \ge 0$ and

$$\mathbb{E}[\mathbf{Reg}_{\mathsf{DM}}] + \sqrt{\mathbb{E}[\overline{\mathbf{Reg}_{\mathsf{DM}}}]\cdot T} = \widetilde{\Omega}\big(\mathsf{dec}_{\gamma,\varepsilon(\gamma,T)}(\mathrm{co}(\mathcal{M}))\cdot T\big),$$

1152 for all $\gamma = \omega(\sqrt{T\log(T)})$, where $\varepsilon(\gamma,T) := c\cdot \frac{\gamma}{T\log(T)}$ for a sufficiently small numerical constant
1153 $c \le 1$. Since there exists $M_0 \in \mathcal{M}$ such that the function $f^{M_0}$ is constant, Lemma B.1 of Foster et al.
1154 [18] further implies that

$$\mathbb{E}[\mathbf{Reg}_{\mathsf{DM}}] + \sqrt{\mathbb{E}[\overline{\mathbf{Reg}_{\mathsf{DM}}}]\cdot T} = \widetilde{\Omega}(\varepsilon(\gamma,T)\cdot \mathsf{dec}_\gamma(\mathrm{co}(\mathcal{M}))\cdot T).$$

For each $T$, set $\gamma = \gamma_T := T$. By the assumption that $\lim_{\gamma \to \infty} \mathsf{dec}_\gamma(\mathrm{co}(\mathcal{M})) \cdot \gamma^\rho = \infty$, we have that for $T$ sufficiently large, $\mathsf{dec}_{\gamma_T}(\mathrm{co}(\mathcal{M})) \geq \gamma_T^{-\rho}$, which implies that and

$$\mathbb{E}[\mathbf{Reg_{DM}}] + \sqrt{\mathbb{E}[\mathbf{Reg_{DM}}] \cdot T} = \widetilde{\Omega}\left(\frac{T}{\gamma_T^\rho}\right),$$

where we have used that $\varepsilon(\gamma_T, T) \propto \frac{1}{\log(T)}$. Rearranging, this implies that

$$\mathbb{E}[\mathbf{Reg_{DM}}] = \widetilde{\Omega}\left(T^{1-2\rho}\right).$$

Hence, for any $p \in (0, 1)$, by setting $\rho = \frac{1-p}{2} \in (0, 1/2)$, we have

$$\mathbb{E}[\mathbf{Reg_{DM}}] = \widetilde{\Omega}(T^p).$$

Applying this argument with $p' = \frac{1}{2}(p+1) \in (1/2, 1)$ yields

$$\lim_{T \to \infty} \frac{\mathfrak{M}(\mathcal{M}, T)}{T^p} = \infty.$$

$\square$

## F.4 Sub-Chebychev Algorithms

**Proposition F.1.** *Any random variable with* $\mathbb{E}[X_+^2] \leq R$ *has*

$$\mathbb{P}(X_+ > t) \leq \frac{R^2}{t^2}, \quad \forall t > 0.$$

*Conversely, if* $X \in (-\infty, B)$ *and has* $\mathbb{P}(X_+ > t) \leq \frac{R^2}{t^2}$ $\forall t > 0$, *then*

$$\mathbb{E}[X_+^2] \leq R^2(\log(B/R) + 1).$$

**Proof of Proposition F.1.** For the first direction, note that if $\mathbb{E}[X_+^2] \leq R$, Chebychev's inequality implies that for all $t > 0$,

$$\mathbb{P}(X_+^2 > t) \leq \frac{R^2}{t^2}. \tag{51}$$

For the other direction, since $X_+ \in [0, B]$ almost surely, we have

$$\mathbb{E}[X_+^2] = \int_0^B \mathbb{P}(X_+ > t)t\,dt \leq R^2 + \int_R^B \mathbb{P}(X_+ > t)t\,dt \leq R^2 + R^2 \int_R^B \frac{1}{t}\,dt \leq R^2 + R^2 \log(B/R).$$

$\square$

**Proposition F.2.** *Suppose that for any* $\delta > 0$, *an algorithm (with* $\delta$ *as a parameter) ensures that with probability at least* $1 - \delta$,

$$\mathbf{Reg_{DM}} \leq R \log^\rho(\delta^{-1})$$

*for some* $R \geq 1$ *and* $\rho > 0$. *Then the algorithm, when invoked with parameter* $\delta = 1/T^2$, *is sub-Chebychev with parameter* $5^{1/2} R \log^\rho(T)$.

**Proof of Proposition F.2.** Set $\delta = 1/T^2$. Then, since $|\mathbf{Reg_{DM}}| \leq T$, the law of total expectation implies that

$$\mathbb{E}[(\mathbf{Reg_{DM}})_+^2] \leq R^2 \log^{2\rho}(T^2) + T^2/T^2 \leq 5R^2 \log^2(T),$$

where we have used that $R \geq 1$. Chebychev's inequality now implies that for all $t > 0$

$$\mathbb{P}((\mathbf{Reg_{DM}})_+ \geq t) \leq \frac{\mathbb{E}[(\mathbf{Reg_{DM}})_+^2]}{t^2} \leq \frac{5R^2 \log^{2\rho}(T)}{t}.$$

$\square$

**Corollary 2.1.** Any regret minimization algorithm with sub-Chebychev parameter $R > 0$ must have

$$R \geq \widetilde{\Omega}(1) \cdot \sup_{\gamma > \sqrt{2C(T)T}} \mathsf{dec}_{\gamma, \varepsilon_\gamma}(\mathrm{co}(\mathcal{M})) \cdot T - O(T^{1/2}). \tag{15}$$

**Proof of Corollary 2.1.** This result immediately follows from Proposition F.1, Proposition F.2, and Theorem 2.2. $\square$