# OpenReview forum: "On the Complexity of Adversarial Decision Making"
_NeurIPS.cc/2022/Conference — NeurIPS 2022 Accept_

### Official Review · Reviewer_tHnk · 2022-07-11

**Rating:** 7
**Confidence:** 3
**Soundness:** 4 excellent
**Presentation:** 3 good
**Contribution:** 3 good

**Summary:**

The paper studies a general adversarial decision-making framework and proposes a new complexity measure, the decision estimation coefficient. It also presents a general algorithm, for which algorithm-specific upper bounds are given with respect to the decision estimation coefficient. An algorithm agnostic lower bound is also given.

**Questions:**

1. It seems the first step of algorithm 1 is fixed to be exponential weights and theorem 2.1 is algorithm-specific. Is it possible to extend eq(4) to other forms of updates (e.g. OMD/FTRL)?

2. The main step of the algorithm involves solving the high probability exploration by optimization objective eq(5). It is not very clear to me if this minimax objective is convex-concave and I am wondering if this can be solved efficiently.

3. for the reward estimator eq(6), I suppose that one would still need some information about the transition (for p^t(\pi^t), not required for bandits, but needed for online RL)? However, the transition information is often assumed to be unknown (e.g. see [1]).

[1] Learning Adversarial Markov Decision Processes with Bandit Feedback and Unknown Transition by Chi Jin, Tiancheng Jin, Haipeng Luo, Suvrit Sra, Tiancheng Yu, ICML 2020

=== === === === ===
The authors have cleared my concerns after their responses. I do enjoy reading the paper and I've raised my score accordingly. Good luck!

**Limitations:**

This work is theoretical, thus have no potential negative social impact.

**Strengths And Weaknesses:**

Strengths:
1. The paper gives one of the first characterizations of complexity in adversarial decision making, which may facilitate further study in bandits/RL etc. The upper and lower bound is new and non-trivial.
2. The paper is also well-written. The relationships between complexities in adversarial/stochastic settings, and connections between different complexity notions are well explained.

Weaknesses:
There is a few places that I am not very clear about. Please see the questions section.

---

> ### Author Response · Authors · 2022-08-02
> **Response for Reviewer tHnk**
>
> We appreciate your positive comments and feedback!
>
> > *It seems the first step of algorithm 1 is fixed to be exponential weights and theorem 2.1 is algorithm-specific. Is it possible to extend eq(4) to other forms of updates (e.g. OMD/FTRL)?*
>
>   While prior works (e.g. [34, 35]) based on
>   exploration-by-optimization can incorporate general OMD/FTRL-style
>   updates, our novel *high-probability* variant leverages
>   specific structural properties of the exponential weights update
>   (specifically, connections between the log-sum-exponential function,
>   which arises as the Fenchel dual of the negative
>   entropy regularizer, and Hellinger distance). Understanding
>   how to extend our results to general OMD/FTRL-style updates (which
>   will like require additional algorithmic changes) is a fascinating
>   direction for future research.
>
> > *The main step of the algorithm involves solving the high probability exploration by optimization objective eq(5). It is not very clear to me if this minimax objective is convex-concave and I am wondering if this can be solved efficiently?*
>
>   Thank you for the question. We show in Lemma E.1 that the exploration-by-optimization
>   objective $(p, g) \mapsto \Gamma_{q, \eta}\left(p, g; \pi^*, M\right)$
>   is always jointly convex in the learner's variables $(p,g)$. This
>   implies that Eq. (5) is a (Lipschitz, but non-smooth) convex
>   optimization problem, which can be solved by standard first-order
>   methods. The only caveat is that calculating the subgradient for the
>   convex function $(p, g) \mapsto \max_{M\in\mathcal{M},\pi^{\star}\in\Pi}\Gamma_{q,
>     \eta}\left(p, g; \pi^*, M\right)$ will generally require enumerating
>   over $\mathcal{M}$ and $\Pi$, so the final algorithm will have runtime
>   scaling with $|\Pi|$ and $|\mathcal{M}|$. This is consistent with
>   previous exploration-by-optimization variants [34, 35], and
>   obtaining sublinear runtime is an interesting question for future research.
>
>   Lastly, let us mention that Eq. (5) equivalent to a convex-concave
>   optimization problem in which we expand to distributions over the
>   pair $(M,\pi^{\star})$ via
> $\min_{(p,g)}\max\_{\mu\in\Delta(\mathcal{M}\times\Pi)}\mathbb{E}\_{(M,\pi^{\star})\sim\mu}\left[\Gamma\_{q,
>     \eta}\left(p, g; \pi^*, M\right)\right]$. As mentioned in lines 985-991 in our submission, the map $\mu
>   \mapsto \mathbb{E}\_{(M, \pi^*) \sim \mu} \left\[\Gamma\_{q, \eta}\left(p, g;
>       \pi^*, M\right)\right]$ is linear in $\mu$ (and is thus a concave
>   function of $\mu$). Furthermore, the map $(p, g) \mapsto \mathbb{E}\_{(M,
>     \pi^*) \sim \mu} \left[\Gamma_{q, \eta}\left(p, g; \pi^*, M\right)\right]$ is
>   convex in $(p, g)$ by Lemma E.1. Thus, this optimization problem
>   is a convex-concave saddle point problem which can be solved using standard first-order min-max optimization
>   algorithms (e.g., [LJJ20]). As before, the running time for solving (5) will scale linearly with the size of the policy class $\Pi$ and the model class $\mathcal{M}$.
>
> [LJJ20] "Near-Optimal Algorithms for Minimax Optimization", Tianyi Lin, Chi Jin, Michael. I. Jordan, 2020
>
> > *for the reward estimator eq(6), I suppose that one
>      would still need some information about the transition (for
>      $p^t(\pi^t)$, not required for bandits, but needed for online
>      RL)? However, the transition information is often assumed to be
>      unknown (e.g. see [1]).*
>
>    When applied to MDPs (i.e., when $\mathcal{M}$ is a class of MDPs), our results do not assume that the transition structure is
>    known. Note, however, that the estimator in Eq. (6) incorporates the
>    observations $z^{(t)}$ observed by the learner after playing a
>    policy $\pi^{(t)}$, and that for MDPs, $z^{(t)} = \left(s^{(t)}\_h,
>    a^{(t)}\_h, r^{(t)}\_h\right)\_{h=1}^H$ is the trajectory for the episode,
>    which allows the objective Eq. (6) to (implicitly) incorporate
>    transition information.
>
>    We emphasize  that while our algorithm *does* incorporate
>    observed trajectories, directly estimating transition information
>    is impossible in the adversarial setting we consider, which is what
>    necessitates our implicit approach. This is because the MDP $M^{(t)}$
>    can change arbitrarily from round to round, and we only get to
>    interact with it for a single episode (via policy
>    $\pi^{(t)}$). In fact, this hardness of estimating transition information is
>    why the optimal regret when $\mathcal{M}$ is a class
>    of tabular MDPs is *exponential* in our adversarial setting (see
>    Example D.4), and contrasts with the stochastic setting, where
>    polynomial regret is possible. In light of this lower bound, the
>    reward estimator in Eq. (6) should be thought of as doing the
>    best one can hope for given the limited trajectory information available.

---

### Official Review · Reviewer_d2Qa · 2022-07-11

**Rating:** 6
**Confidence:** 3
**Soundness:** 4 excellent
**Presentation:** 3 good
**Contribution:** 4 excellent

**Summary:**

The paper considers a general adversarial decision making framework that encompasses (structured) bandit problems with adversarial rewards and reinforcement learning problems with adversarial dynamics. The main result shows that the Decision-Estimation Coefficient is both necessary and sufficient for low regret in the adversarial setting. The paper also provides other interesting results, such as the connection of the Decision-Estimation Coefficient to variants of other well-known complexity measures.

**Questions:**

* What are the implications of the work on the practical side as informed by the theoretical development?
* Please provide some intuitive explanations of what V(\mathcal{M}) represents, and comment on the conditions for which to be finite.
* While the authors have explained the role of convexity in lower bound, it is not clear what is the cost we need to pay by taking the convex hull of model class for the upper bound.

**Limitations:**

The authors have discussed the limitations of the work.

**Strengths And Weaknesses:**

Strengths:
* The paper is a nontrivial extension of the study by Foster et al. [18] in the stochastic setting to the adversarial setting.
* The results are theoretically sound and novel.

Weakness:
* While the focus of the work is primarily on sample complexity, Algorithm 1 may not be practical for large decision spaces. Thus, the contributions are limited to theoretical understandings.

---

> ### Author Response · Authors · 2022-08-02
> **Response for Reviewer d2Qa**
>
> We appreciate your positive comments!
>
> > *Algorithm 1 may not be practical for large decision spaces*
>
> Thank you for raising this question. We emphasize that the focus of this work is not on
> computational efficiency, but on understanding statistical complexity
> and fundamental limits (i.e., optimal regret, as determined by the
> class $\mathcal{M}$). The ExO+ algorithm is computationally efficient for
> classes where $|\Pi|$ and $|\mathcal{M}|$ are small , which is consistent with
> previous exploration-by-optimization variants [34, 35], but is
> inefficient for
> classes like MDPs where $|\Pi|$ is large. This is somewhat
> expected for the extremely high level of generality we consider (for
> example, [41] show *computational* lower bounds for achieving low regret
> with adversarial MDPs), and
> understanding what additional structural assumptions lead to more
> efficient algorithms (e.g., with sublinear runtime) is an interesting
> question for future research.
>
> > *What are the implications of the work on the practical side as informed by the theoretical development?*
>
> From a practical viewpoint, the main takeaways from our paper are:
> - We provide a precise characterization of the minimax regret for
>   adversarial interactive learning, and we believe that our work can serve as a
> valuable benchmark for design of efficient algorithms going
> forward: For any new problem of
>   interest, one can use our characterization to understand if the
>   problem is tractable, and if it is, then the practitioner can use
>   our work as a starting point from which to develop efficient
>   algorithms tailored to the problem of interest.
>
> - We provide a frequentist algorithm that is implementable for small policy and model classes.
>
> Lastly, let us mention an implication of our results for solving
> two-player (or $n$-player) games
> with limited (e.g., bandit) feedback. A standard approach to computing
> Nash equilibria for two-player zero-sum games is to take an algorithm
> with a regret guarantee for adversarial outcomes, and have both
> players select actions using individual copies of the
> algorithm. [41] show that for adversarial MDPs, the optimal regret is
> exponential, so that for Markov games, computing equilibria by
> reducing to regret minimization in this fashion is not viable. Our
> results show that this phenomenon is generic: naively applying to regret minimization will lead to poor results for classes where the convexified DEC is large.
>
> > *Please provide some intuitive explanations of what
>     $V(\mathcal{M})$ represents, and comment on the conditions for
>     which to be finite.*
>
> We are happy to include further discussion in the final version! We emphasize that finiteness of $V(\mathcal{M})$ is not required by our
> results---the purpose of including this parameter is that for
> classes where $V(\mathcal{M})=O(1)$, a $1/\log(T)$ factor can be removed from
> the lower bound in Theorem 2.2, which leads to tighter results for
> standard classes. Briefly, $V(\mathcal{M})$ is a measure of absolute continuity for
> pairs of models in $\mathcal{M}$. When this parameter is bounded, this means
> that if a model $M\in\mathcal{M}$ assigns low probability to an event $\mathcal{E}$,
> any other model $M'\in\mathcal{M}$ must assign relatively low probability as
> well. Typically, $V(\mathcal{M})$ is not bounded globally, but is bounded for a
> well-behaved sub-class $\mathcal{M}'\subseteq\mathcal{M}$, which suffices to derive tight
> lower bounds for standard classes. For example, $V(\mathcal{M})=O(1)$ when
> $\mathcal{M}$ consists of Bernoulli bandit problems with means in
> $[\epsilon,1-\epsilon]$ for constant $\epsilon$, which suffices to derive
> a $\sqrt{AT}$ minimax lower bound. Please see further
> examples within [18].
>
> > *While the authors have explained the role of convexity in lower bound, it is not clear what is the cost we need to pay by taking the convex hull of model class for the upper bound.*
>
> For many cases, (e.g. finite-armed, linear bandits, and convex
> bandits), $\mathrm{co}(\mathcal{M}) = \mathcal{M}$, and thus there is no overhead from
> convexification. However, for other cases, e.g. MDPs, $\mathrm{co}(\mathcal{M})
> \neq \mathcal{M}$ and the price of convexification can make the problem
> intractable (e.g., for MDPs, $\mathsf{dec}\_{\gamma}(\mathcal{M})$ is
> polynomial, while $\mathsf{dec}\_{\gamma}(\mathrm{co}(\mathcal{M}))$ is
> exponential). We summarize these differences in Section 4 of the
> paper. In general, it seems unlikely that there exists a succinct/generic
> characterization for how large
> $\mathsf{dec}\_{\gamma}(\mathrm{co}(\mathcal{M}))$ can be relative to
> $\mathsf{dec}\_{\gamma}(\mathcal{M})$, but this is an interesting direction for future research. We emphasize, however,
> that our results provide matching upper and *lower* bounds on the regret for
> adversarial decision making, and our lower bounds scale with
> $\mathsf{dec}\_{\gamma}(\mathrm{co}(\mathcal{M}))$, which shows that the
> convexified class $\mathrm{co}(\mathcal{M})$ indeed plays a fundamental role.

---

### Official Review · Reviewer_iALf · 2022-07-12

**Rating:** 7
**Confidence:** 4
**Soundness:** 4 excellent
**Presentation:** 3 good
**Contribution:** 3 good

**Summary:**

The paper proposes a new measure of complexity for sequential decision making in the adversarial setting, called (convexified) Decision-Estimation Coefficient (DEC in the following), in term of which matching upper and lower bounds on the regret can be established. This follows directly from the work by Foster et al. (2021) that did the same thing for the stochastic setting.
In the stochastic setting, intuitively, the DEC captures the tradeoff between playing high-reward actions (exploitation) and revealing information about the reward model by playing potentially suboptimal actions (exploration). It is then a function of the model class $\mathcal{M}$ and a trade-off parameter $\gamma$. The main intuition from the present paper is that the DEC can be used for the adversarial setting by replacing $\mathcal{M}$ with the convex hull of $\mathcal{M}$. Intuitively, this is because the adversary could fix a distribution over models and sample a model from it at each round.
Following the same outline as Foster et al. (2021), first a general regret upper bound is established for an abstract algorithm. The algorithm is inspired by the Exploration-by-Optimization framework (Lattimore and Gyorgy, 2021), but allows to establish high-probability regret bounds rather than in expectation. Then, a matching lower bound is established for a meaningful class of algorithms. This is established by first refining the lower bound for the stochastic setting from Foster et al. (2021) by using a novel change-of-measure result. Both are of independent interest. Finally, some concrete examples are given, recovering known results in bandits and RL. Interestingly, convexification provides a very clear picture of what problems are easy and which are not in the adversarial setting: problems that are known to be tractable have $\mathcal{M}=co(\mathcal{M})$.

**Questions:**

1. How practical is the ExO+ algorithm once instantiated to the bandit and MDP examples? How hard is the optimization problem from line 4? Is the linear time complexity from the exponential weights update from line 3? I think more discussion on the algorithm itself would be helpful.
2. I checked the main proofs and they seem sound, but I am still a bit puzzled by the fact that convexification is enough to capture the hardness of the adversarial problem. An adversary that commits to a model distribution just seems too easy. Do you have any further intuition about why small convexified DEC is sufficient for small regret?
3. Do you know of any concrete example where $co(\mathcal{M})$ is strictly larger than $\mathcal{M}$ but the adversarial problem is still tractable?

Minor:
- line 338: structural
- line 579: "in fact paper" ?
- line 681: where is this Chi-squared-Hellinger inequality from?

**Limitations:**

Limitations are sufficiently discussed, I just suggest to expand the discussion on the algorithm.

**Strengths And Weaknesses:**

The paper builds heavily on Foster et al. (2021) and is incremental in nature, but has several original and significant contributions, such as the high probability variant of exploration-by-optimization and the refined change-of-measure arguments.
Mostly, this work inherits the strengths and weaknesses of its stochastic predecessor: the main strength being its generality and unification power, the main weakness being the abstractness and highly technical nature of the central result. Specifically, the DEC encodes the exploration-exploitation trade-off in such a literal way that it looks more like an intermediate proof step than a measure of complexity.
However, the effort to establish a general theory of adversarial sequential decision making is remarkable, and a lot of insight can be gained from reading this paper.
The paper is also very well written. The technical results are clearly stated and backed up by intuition, the notation is consistent, the separation between existing results and original contributions is clearly established, and the proofs of the main theorems are also well commented and easy to follow.

---

> ### Author Response · Authors · 2022-08-02
> **Response for Reviewer iALf**
>
> > How practical is the ExO+ algorithm once instantiated to the bandit and MDP examples?
>
> We will expand the discussion of computational efficiency in the final version of the paper. In more detail:
> * The exponential weights update step in line (3) has runtime (as
>   well as memory) linear in the size of the decision space $\Pi$,
>   since the algorithm maintains a separate weight for each decision $\pi\in\Pi$.
> *  We show in Lemma E.1 that the exploration-by-optimization
>   objective $(p, g) \mapsto \Gamma_{q, \eta}(p, g; \pi^*, M)$  is always jointly convex in the learner's variables $(p,g)$. This implies that Eq. (5) is a (Lipschitz, but non-smooth) convex optimization problem, which can be solved by standard first-order methods. The only caveat is that calculating the subgradient for the  convex function $(p, g) \mapsto \max_{M\in\mathcal{M},\pi^{\star}\in\Pi}\Gamma_{q, \eta}(p, g; \pi^*, M)$ will generally require enumerating over $\mathcal{M}$ and $\Pi$, so the final algorithm will have runtime scaling with $|\Pi|$ and $|\mathcal{M}|$. See the response to Reviewer tHnk for further details.
>
> Summarizing, the ExO+ algorithm is computationally efficient for
> classes where $|\Pi||$ and $|\mathcal{M}|$ are small, which is consistent with previous exploration-by-optimization variants [34, 35], but is inefficient for classes like MDPs where $|\Pi|$ is large. This is somewhat expected for the high level of generality we consider (for example, [41] show *computational* lower bounds for achieving low regret with adversarial MDPs), and understanding what additional structural assumptions lead to more efficient algorithms (e.g., with sublinear runtime) is an interesting question for future research.
>
> > Do you have any further intuition about why small convexified DEC is sufficient for small regret?
>
> On the upper bound side, the adversarial nature of our setting
> (rewards and outcomes) is taken care of by the fact that we are
> running a *online learning* algorithm (exponential weights), and
> the regret of this algorithm is controlled by the quality of the
> estimators produced by the exploration-by-optimization objective (in
> particular, regret is bounded by the parameter
> ${\sf exo}\_{\eta}(\mathcal{M})$ introduced in Eq. (10)). Hence, your
> question can be boiled down to: "why is ${\sf exo}\_{\eta}(\mathcal{M})$
> bounded by $\mathsf{dec}\_{\gamma}(\mathrm{co}(\mathcal{M}))$?". The answer here comes
> from minimax duality: Even though the $\mathsf{exo}\_{\eta}(\mathcal{M})$ objective is defined as an adversarial problem in which an adversary chooses a model $M\in\mathcal{M}$ (a proxy for the unknown true model $M^{(t)}$ at round $t$) given the learner's choice of distribution/estimator $(p,g)$, the value is equivalent to a stochastic (Bayesian) problem in which the adversary first chooses a prior over models, and the learner chooses $(p,g)$ with knowledge of this distribution (see proof of Theorem 3.2). The reason why the convexified DEC bounds the value of this Bayesian problem (but the regular DEC does not), is that it contains certain posterior distributions that naturally arise.
>
> Finally, our work only considers finite decision
> spaces $\Pi$, and our upper bounds scale with $\log|\Pi|$. Based
> on existing results in (full-information) online learning, we expect to see further differences between the stochastic and adversarial
> setting for infinite classes. For the stochastic
> setting, one might hope to replace $\log|\Pi|$ with a classical
> complexity measure like the VC-dimension, but for the adversarial
> setting, the best one should hope for is an analogous
> *sequential* complexity measure such as the Littlestone dimension
> (see [Ref 2]).
>
> [Ref 1] Statistical Learning and Sequential Prediction, Rakhlin and  Sridharan
>
> [Ref 2] Bandit Algorithms, Lattimore Csaba Szepes\'vari
>
> > Do you know of any concrete example where adversarial is harder but still tractable?
>
> As a basic example, consider the case where $\mathcal{M}$ is  the class of finite-armed bandit problems with $\Pi=[A]$ and *noiseless rewards* in $[0,1]$ (i.e., $r=f^{M}(\pi)$  a.s. under $r\sim{}M(\pi)$). In this case it is straightforward to show that  $\mathsf{dec}\_{\gamma}(\mathcal{M})\propto\mathbf{1}\{\gamma\leq{}A/2\}$,  while  $\mathsf{dec}\_{\gamma}(\mathrm{conv}(\mathcal{M}))\propto\frac{A}{\gamma}$. This  leads to $\mathrm{Reg}=\tilde{\Theta}(A)$ in the stochastic setting  and $\mathrm{Reg}=\tilde{\Theta}(\sqrt{AT})$ in the adversarial setting.
>
> The intuition here is that in the stochastic setting, because $\mathcal{M}$  only contains models with noiseless rewards, the learner can pull each arm a single time to learn the reward exactly, so that only $A$ rounds are required to identify the optimal action. However,  models in $\mathrm{co}(\mathcal{M})$ can ``simulate'' noise, which forces the learner to pay the usual $\sqrt{AT}$ minimax rate in the adversarial setting.
>
> > Line 681:
>
> We will add a reference in the final version.

---

> > ### Comment · Reviewer_iALf · 2022-08-08
> > **Thanks**
> >
> > Thank you for your answer, it improved my understanding of your work, and I confirm my score.
> > I think the noiseless MAB example is also worth including in the final version.

---

### Meta-Review · Area_Chair_S9vr · 2022-08-27

**Recommendation:** Accept
**Confidence:** Certain

**Metareview:**

The submission studies the important problem of quantifying the complexity of learning in adversarial sequential decision problems with partial feedback. Although the problem is well-studied in the full-information setting, the same problems in the bandit and reinforcement learning settings are largely open. It is still not clear how the optimal regret depends on the shape of the action and parameter spaces.

This paper makes a significant contribution in this area. It shows that the Decision-Estimation Coefficient (introduced by Foster et al. 2021) quantifies the complexity of learning in many adversarial sequential decision problems. This result would be of interest to the online learning community in NeurIPS. Although the paper does not really resolve any open questions or show substantial improvements that wouldn't be possible with other techniques, the paper provides new important insights, and I believe the introduced tools and techniques in this paper will ultimately lead to such improvements.

**Award:**

Yes

---

### Decision · Program_Chairs · 2022-09-14

Accept